# FNDC1 is a myokine that promotes myogenesis and muscle regeneration

Rui Xin Zhang [1,2], Yuan Yuan Zhai[1,2], Rong Rong Ding[1,2], Jia He Huang[1], Xiao Chen Shi[1], Huan Liu[1], Xiao Peng Liu[1], Jian Feng Zhang[1], Jun Feng Lu[1], Zhe Zhang [1], Xiang Kai Leng[1], De Fu Li[1], Jun Ying Xiao[1], Bo Xia [1] & Jiang Wei Wu [1 ✉]

## Abstract

**Myogenesis is essential for skeletal muscle formation and regeneration after injury, yet its regulators are largely unknown. Here we identified fibronectin type III domain containing 1 (FNDC1) as a previously uncharacterized myokine. In vitro studies showed that knockdown of *Fndc1* in myoblasts reduces myotube formation, while overexpression of *Fndc1* promotes myogenic differentiation. We further generated recombinant truncated mouse FNDC1 (mFNDC1), which retains reliable activity in promoting myoblast differentiation in vitro. Gain- and loss-of-function studies collectively showed that FNDC1 promotes cardiotoxin (CTX)-induced muscle regeneration in adult mice. Furthermore, recombinant FNDC1 treatment ameliorated pathological muscle phenotypes in the *mdx* mouse model of Duchenne muscular dystrophy. Mechanistically, FNDC1 bound to the integrin α5β1 and activated the downstream FAK/PI3K/AKT/mTOR pathway to promote myogenic differentiation. Pharmacological inhibition of integrin α5β1 or of the downstream FAK/PI3K/AKT/mTOR pathway abolished the pro-myogenic effect of FNDC1. Collectively, these results suggested that myokine FNDC1 might be used as a therapeutic agent to regulate myogenic differentiation and muscle regeneration for the treatment of acute and chronic muscle disease.**

**Keywords** FNDC1; Integrin α5β1; Myogenic Differentiation; Muscle Regeneration
**Subject Categories** Development; Signal Transduction; Stem Cells & Regenerative Medicine

## Introduction

Skeletal muscle is a highly plastic tissue that is capable of making up for turnover caused by daily wear-and-tear as well as regenerating upon damage (Furrer and Handschin, 2019). Myogenesis is required for formation of skeletal muscle and the regeneration after injury, and its dysfunction leads to muscle disorder (Chamberlain and Chamberlain, 2017). Myogenesis follows an ordered series of cellular events including satellite cell activation, migration, alignment, adhesion, and membrane fusion, generating new multinucleated myofibers or fusing with existing ones, which is critical for the repair of damaged myofibres (Park et al, 2016; Tierney and Sacco, 2016). Nonetheless, regulators of myogenesis remain largely unknown.

As an endocrine organ, skeletal muscle secretes a variety of factors to regulate muscle function under different circumstances such as during exercise or muscle repair upon injury (Pedersen and Febbraio, 2012). Exercise-induced myokines, such as FNDC5/irisin (Fang et al, 2023; Kurdiova et al, 2014) and interleukin 6 (IL6) (Pedersen and Febbraio, 2008), increase skeletal muscle mass and strength. Another class of myokines such as brain-derived neurotrophic factor (BDNF) (Wang et al, 2021; Waldemer-Streyer et al, 2022), is abundantly generated after muscle injury or atrophy, stimulating regeneration and reversing muscle atrophy by directly or indirectly regulating satellite cells. Apart from extensive efforts in developing the inhibitors for the well-recognized myokine myostatin, most of the myokines have not been sufficiently characterized (Lo et al, 2020; Markworth et al, 2020). Identification of novel myokines in myogenesis and skeletal muscle regeneration is also valuable for exploring innovative therapeutic targets.

To explore potential regulators of myogenesis, we performed a bioinformatics analysis on four myogenesis-related microarray datasets and identified fibronectin type III domain containing 1 (*Fndc1*) as a candidate gene. FNDC1 contains four conserved fibronectin type III (FN3) domains and one large tegument protein UL36 (PHA03247) domain. It was first identified in rat ischemic hearts and was involved in hypoxia-induced cardiomyocyte apoptosis (Sato et al, 2006; Sato et al, 2009). FNDC1 has also been identified as the causative gene for otitis media (van Ingen et al, 2016) and Kawasaki disease (Lin et al, 2021). It is closely related to the occurrence and poor prognosis of various malignant tumors (lymphoma (Sung et al, 2011), breast cancer (Cordeiro et al, 2021), gastric cancer (Jiang et al, 2020), colorectal cancer (Chen et al, 2022), and lung cancer (Ma et al, 2023)). The biological function of FNDC1 in skeletal muscle has not been documented yet. Mutations in human *FNDC1* lead to impaired muscle function with ageing (Franco et al, 2018), implying that FNDC1 may play a key role in

---

[1]Key Laboratory of Animal Genetics, Breeding and Reproduction of Shaanxi Province, College of Animal Science and Technology, Northwest A&F University, Yangling, Shaanxi 712100, China. [2]These authors contributed equally: Rui Xin Zhang, Yuan Yuan Zhai, Rong Rong Ding. ✉E-mail: wujiangwei@nwafu.edu.cn

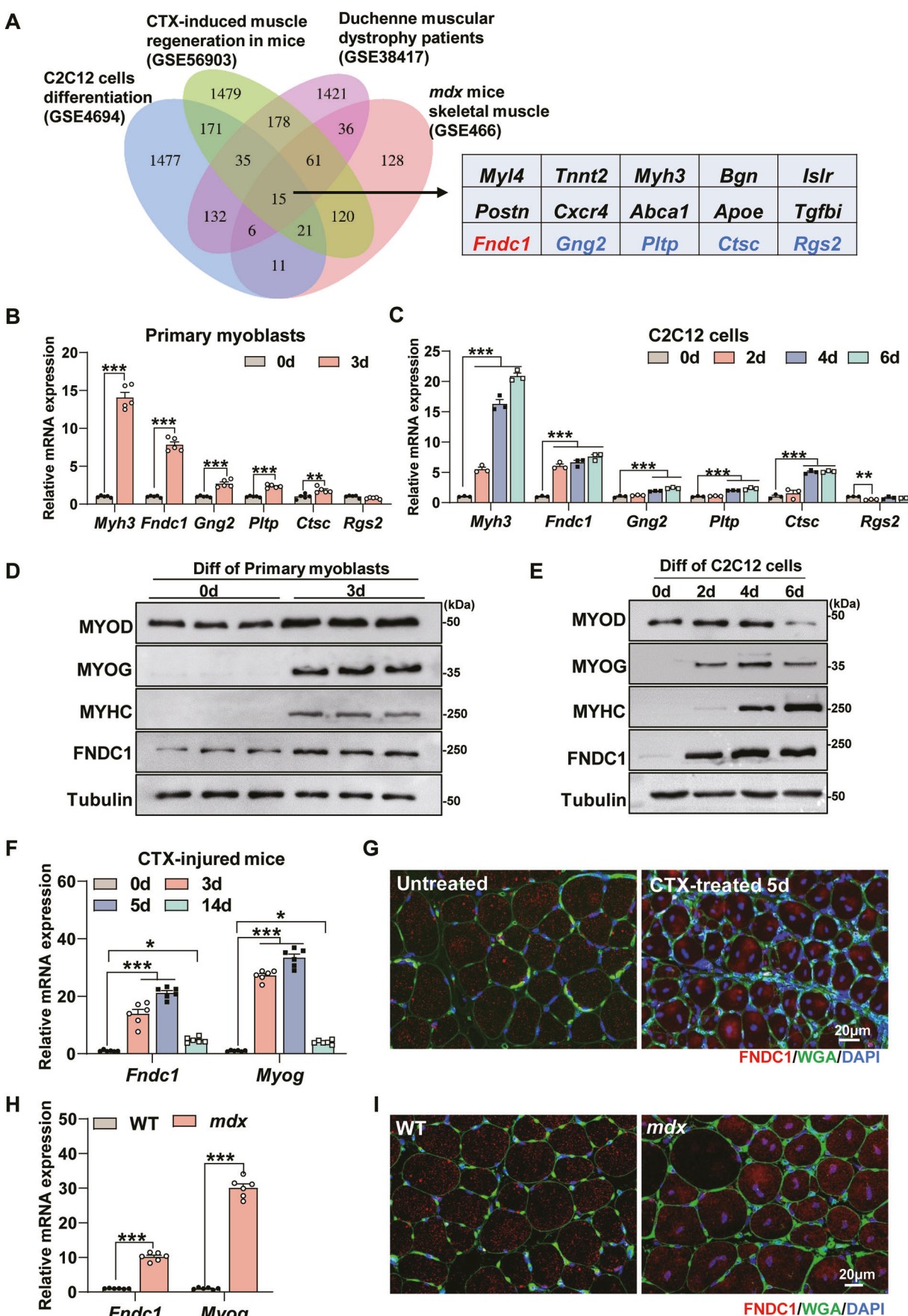

**Figure 1.  FNDC1 is closely associated with myogenesis.**

(A) Venn diagram showing fifteen overlapping differentially expressed genes among four independent microarray datasets related to myogenesis. (B) mRNA expression of *Myh3, Fndc1, Gng2, Pltp, Ctsc,* and *Rgs2* in primary myoblasts during myogenic differentiation ($n = 5$ independent experiments). Two-tailed t-test, $P = 6.5 \times 10^{-8}$, $P = 1.23 \times 10^{-7}$, $P = 5.95 \times 10^{-5}$, $P = 1.54 \times 10^{-6}$, $P = 5.97 \times 10^{-3}$, and $P = 0.0477$ for *Myh3, Fndc1, Gng2, Pltp, Ctsc,* and *Rgs2,* respectively. (C) mRNA expression of *Myh3, Fndc1, Gng2, Pltp, Ctsc,* and *Rgs2* in C2C12 cells during myogenic differentiation ($n = 3$ independent experiments). One-way ANOVA, 0 d *vs.* 2 d, $P = 6.63 \times 10^{-4}$, 0 d *vs.* 4 d, $P = 4.80 \times 10^{-8}$, 0 d *vs.* 6 d and $P = 1.12 \times 10^{-8}$ for *Myh3*; 0 d *vs.* 2 d, $P = 8.19 \times 10^{-6}$, 0 d *vs.* 4 d, $P = 3.59 \times 10^{-6}$, 0 d *vs.* 6 d, $P = 1.05 \times 10^{-6}$ for *Fndc1*; 0 d *vs.* 2 d, $P = 0.2487$, 0 d *vs.* 4 d, $P = 7.03 \times 10^{-4}$ and 0 d *vs.* 6 d and $P = 3.34 \times 10^{-5}$ for *Gng2*; 0 d *vs.* 2 d, $P = 0.8222$, 0 d *vs.* 4 d, $P = 5.06 \times 10^{-5}$ and 0 d *vs.* 6 d, $P = 7.90 \times 10^{-6}$ for *Pltp*; 0 d *vs.* 2 d, $P = 0.5834$, 0 d *vs.* 4 d, $P = 3.78 \times 10^{-5}$ and 0 d *vs.* 6 d, $P = 3.20 \times 10^{-5}$ for *Ctsc;* and 0 d *vs.* 2 d, $P = 3.94 \times 10^{-3}$, 0 d *vs.* 4 d, $P = 0.1819$ and 0 d *vs.* 6 d, $P = 0.1222$ for *Rgs2*. (D, E) Representative immunoblotting of MYOD, MYOG, MYHC, and FNDC1 in primary myoblasts or C2C12 cells during myogenic differentiation. (F) mRNA expression of *Fndc1* and *Myog* in the tibialis anterior (TA) muscle of cardiotoxin (CTX) injured WT mice ($n = 6$ mice). One-way ANOVA, 0 d *vs.* 3 d, $P = 1.69 \times 10^{-8}$, 0 d *vs.* 5 d, $P = 6.04 \times 10^{-12}$ and 0 d *vs.* 14 d, $P = 0.0204$ for *Fndc1*; 0 d *vs.* 3 d, $P = 2.70 \times 10^{-14}$, 0 d *vs.* 5 d, $P = 2.3 \times 10^{-14}$ and 0 d *vs.* 14 d, $P = 0.0454$ for *Myog*. (G) Representative immunofluorescence staining of FNDC1$^+$ fibers (in red) in untreated or CTX-treated TA muscle at day 5 post-injury. Cell membrane was stained with WGA (in green) and nuclei were counterstained with DAPI (in blue) ($n = 6$ mice). Scale bar $= 20\ \mu m$. (H) mRNA expression of *Fndc1* and *Myog* in TA muscle of WT or *mdx* mice ($n = 6$ mice). Two-tailed t-test, $P = 3.27 \times 10^{-9}$ and $P = 2.16 \times 10^{-7}$ for *Fndc1* and *Myog*, respectively. (I) Representative immunofluorescence staining of FNDC1$^+$ fibers (in red) in TA muscle of WT or *mdx* mice. Cell membrane was stained with WGA (in green) and nuclei were counterstained with DAPI (in blue) ($n = 6$ mice). Scale bar $= 20\ \mu m$. Data are represented as mean ± SEM. *$p < 0.05$, **$p < 0.01$, and ***$p < 0.001$. Source data are available online for this figure.

skeletal muscle homeostasis. Furthermore, its family member, FNDC5, is a well-known pro-muscle factor that induces skeletal muscle hypertrophy and rescues atrophy (Reza et al, 2017). FNDC4 was also reported to promote skeletal muscle cell differentiation (Li et al, 2020; Wang et al, 2020). Based on this evidence, we deduced that FNDC1 may also play a key role in myogenesis. Thus, the aim of this study was to explore the role of FNDC1 in myogenesis and muscle regeneration and to understand how this process is precisely controlled.

We identified FNDC1 as a novel myokine involved in myogenesis and muscle regeneration. Knockdown of *Fndc1* in C2C12 cells reduced myotube formation, whereas *Fndc1* overexpression promoted myogenic differentiation. We also generated a truncated form of recombinant FNDC1 and demonstrated that it promotes cardiotoxin (CTX)-induced muscle regeneration in adult mice and alleviates muscular atrophy in *mdx* mice. These beneficial effects are dependent on the membrane integrin receptor α5β1. FNDC1 binds to α5β1 and activates the downstream FAK/PI3K/AKT/mTOR pathway to promote myogenesis and muscle regeneration. Our findings identify a new myokine FNDC1 and reveal how it works in myogenesis, with the therapeutic potential for the treatment of muscle diseases.

## Results

### FNDC1 is a previously unrecognized candidate for regulating myogenesis

To identify potential regulators involved in myogenesis, we compiled and intersected differentially expressed genes (DEGs) from four independent microarray datasets related to myogenesis from Gene Expression Omnibus (GEO) database: (i) DEGs during C2C12 cell differentiation (GSE4694); (ii) DEGs during CTX-induced muscle regeneration in mice (GSE56903); (iii) DEGs in skeletal muscle of Duchenne muscular dystrophy (DMD) patients (GSE38417); and (iv) DEGs in gastrocnemius muscle of chronic muscular dystrophic (*mdx*) mice (GSE466). Integrative analysis of the datasets yielded fifteen overlapping genes, ten of which (*Myl4* (Schiaffino et al, 2015), *Tnnt2* (Wei and Jin, 2016), *Myh3* (Toydemir et al, 2006), *Bgn* (Lechner et al, 2006), *Islr* (Zhang et al, 2018), *Postn* (Ito et al, 2021), *Cxcr4* (Lahmann et al, 2021), *Abca1* (Tan et al, 2019), *Apoe* (Arnold et al, 2015), and *Tgfbi* (Girardi et al, 2021)) have been well-characterized in myogenesis

(Fig. 1A). The roles of the remaining five genes (*Fndc1, Gng2, Pltp, Ctsc,* and *Rgs2*) in myogenesis have not been documented (Fig. 1A). Therefore, we analyzed their expression levels and found that *Fndc1* was the most significantly elevated gene during primary myoblast differentiation compared with the other four genes (*Gng2, Pltp, Ctsc,* and *Rgs2*) (Fig. 1B). During C2C12 cell differentiation, *Fndc1* mRNA expression was upregulated on day 2 and remained high throughout the differentiation stage (Fig. 1C). mRNA expression of *Gng2, Pltp,* and *Ctsc* remained unchanged on day 2 and was elevated during the terminal differentiation stage (Fig. 1C). In contrast, mRNA expression of *Rgs2* showed a decreasing trend on day 2 (Fig. 1C). Consistent with the mRNA expression pattern, FNDC1 protein expression was also increased in primary myoblasts and C2C12 cells during differentiation (Fig. 1D,E; Appendix Fig. S1A,B). These results suggest that FNDC1 may be involved in the entire process of myoblast differentiation and thus was chosen for further investigation.

We next analyzed FNDC1 expression in mice with CTX-induced acute muscle injury as well as in *mdx* mice. In CTX-treated mice, compared with uninjured muscle (day 0), *Fndc1* expression was significantly upregulated during the initial phase of muscle regeneration (days 3–5) and downregulated on day 14 when myoblast fusion was decreased, similar to the expression profile of myogenic marker *Myog* (Appendix Fig. S2; Fig. 1F). At day 5 after injury, immunofluorescence analysis revealed increased FNDC1 in newly regenerated muscle fibers (Fig. 1G). We also noticed markedly increased *Fndc1* mRNA expression (~12-fold) in the anterior tibial (TA) muscle of *mdx* mice compared with WT mice, similar to the expression profile of *Myog* (Fig. 1H). Consistent with this, immunofluorescence analysis showed significantly higher FNDC1 in the muscle fibers of *mdx* mice compared with WT mice (Fig. 1I). These results show that FNDC1 is differentially expressed in different myogenic systems, implying that it may play key roles in myogenic differentiation.

### FNDC1 promotes myoblast differentiation

We next investigated the role of FNDC1 in myoblast differentiation. Knockdown of *Fndc1* significantly inhibited myogenic differentiation (Fig. 2A–D), with reduced mRNA expression of *Caveolin-3* and *Myomaker* (Fig. 2E), two critical markers related to cell fusion, as well as protein levels of MYHC and MYOG (Fig. 2F; Appendix Fig. S3A). In contrast, *Fndc1* overexpression increased

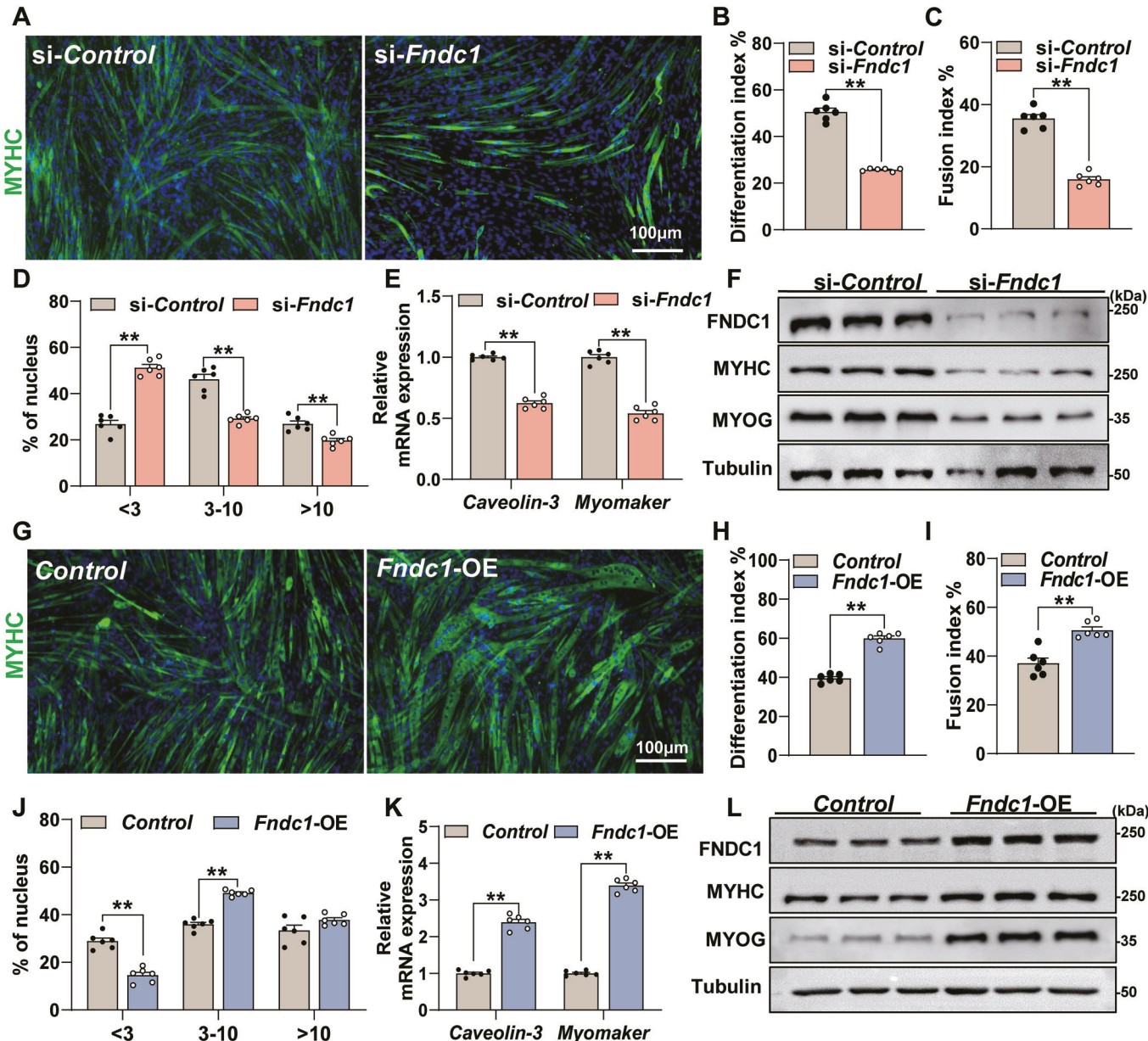

**Figure 2. FNDC1 promotes myoblast differentiation-mediated myogenesis in vitro.**

(A) Representative immunofluorescence staining of MYHC in si-Control or si-*Fndc1* (*Fndc1* knockdown) C2C12 cells at day 4 post-differentiation (*n* = 6 independent experiments). Staining for MYHC (in green) marks differentiated cells, and nuclei were counterstained with DAPI (in blue). C2C12 cells were transfected with si-Control or si-*Fndc1* for 12 h before the initiation of differentiation. Scale bars = 100 μm. (B) Quantification of the differentiation index (*n* = 6 independent experiments). Two-tailed t-test, *P* = 2.99 × 10⁻⁸. (C, D) Quantification of fusion index (a MYHC⁺ cell with at least three nucleus) and the nucleus distribution per myotube (*n* = 6 independent experiments). (C) Two-tailed t-test, *P* = 1.70 × 10⁻⁷. (D) Two-tailed t-test, nucleus distribution per myotube <3, *P* = 1.70 × 10⁻⁷; nucleus distribution per myotube 3–10, *P* = 1.64 × 10⁻⁵ and nucleus distribution per myotube >10, *P* = 8.73 × 10⁻⁴. (E) mRNA expression of *Caveolin-3* and *Myomaker* (*n* = 6 independent experiments). Two-tailed t-test, *P* = 5.71 × 10⁻⁹ and *P* = 4.12 × 10⁻⁸ for *Caveolin-3* and *Myomaker*, respectively. (F) Representative immunoblot analysis of FNDC1, MYHC, and MYOG (*n* = 3 independent experiments). Cells were collected at day 4 post-differentiation. (G) Representative immunofluorescence staining of MYHC (in green) in Control (empty vector) or *Fndc1*-overexpressing (*Fndc1*-OE) C2C12 cells at day 4 post-differentiation. Scale bars = 100 μm. (H) Quantification of differentiation index (*n* = 6 independent experiments). Two-tailed t-test, *P* = 1.11 × 10⁻⁷. (I, J) Quantification of fusion index and the nucleus distribution per myotube (*n* = 6 independent experiments). (I) Two-tailed t-test, *P* = 3.20 × 10⁻⁴. (J) Two-tailed t-test, nucleus distribution per myotube <3, *P* = 1.41 × 10⁻⁵; nucleus distribution per myotube 3–10, *P* = 1.41 × 10⁻⁷ and nucleus distribution per myotube >10, *P* = 0.0992; (K) mRNA expression of *Caveolin-3* and *Myomaker* (*n* = 6 independent experiments). Two-tailed t-test, *P* = 1.64 × 10⁻⁸ and *P* = 1.98 × 10⁻¹¹ for *Caveolin-3* and *Myomaker*, respectively. (L) Representative immunoblot analysis of FNDC1, MYHC, and MYOG in Control or *Fndc1*-OE C2C12 cells (*n* = 3 independent experiments). Cells were collected at day 4 post-differentiation. Data are represented as mean ± SEM. **p < 0.01. Source data are available online for this figure.

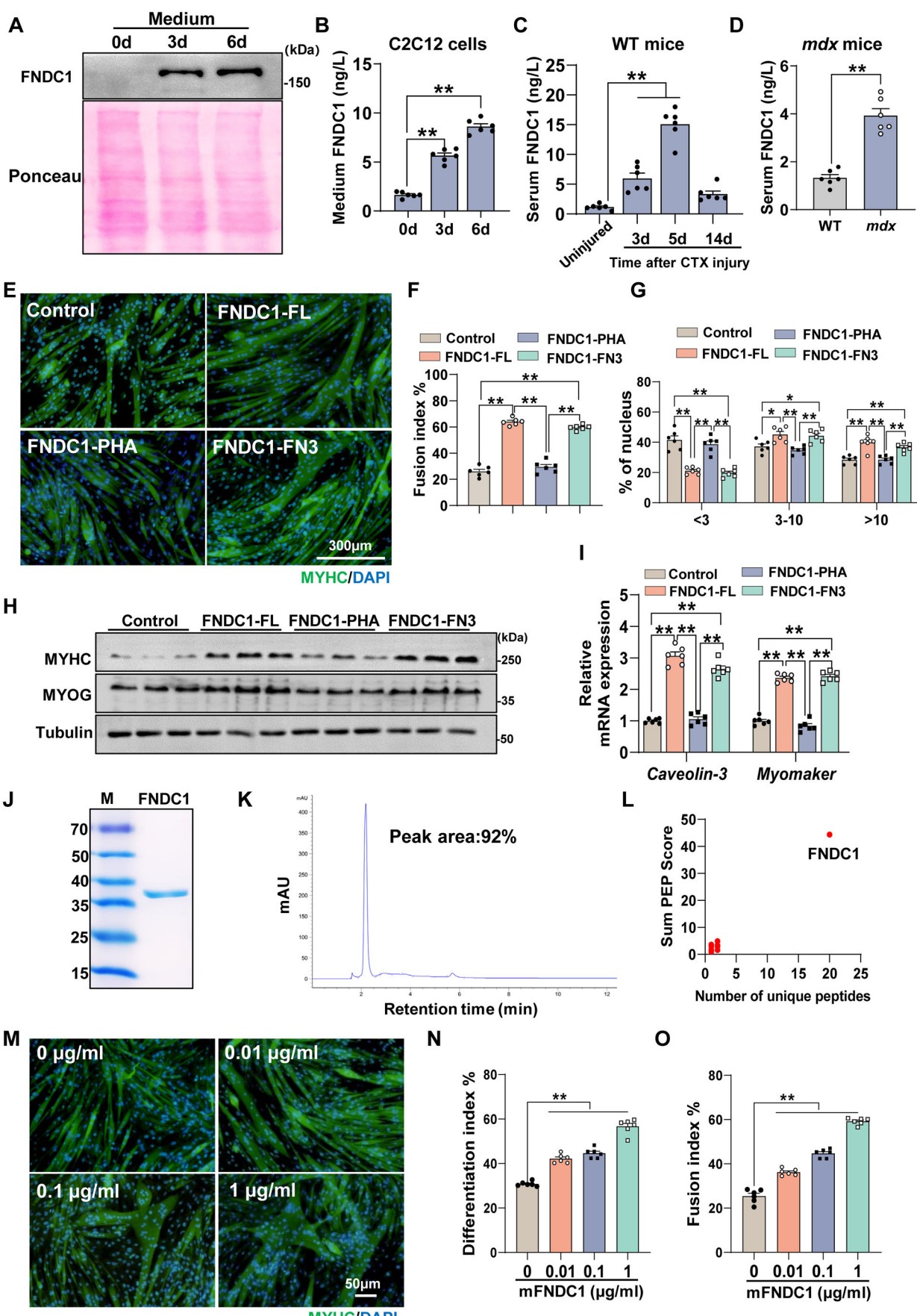

**Figure 3. FNDC1 is a novel myokine.**

(A) Representative immunoblot analysis of medium FNDC1 in C2C12 cells at days 0, 3, or 6 post-differentiations. For determination of released FNDC1, equal volumes of conditioned media were subjected to western blot analysis; Ponceau S staining was used as a loading control. (B) Medium FNDC1 levels in C2C12 cells at days 0, 3, or 6 post-differentiation ($n = 6$ independent experiments). One-way ANOVA, 0 d $vs.$ 3 d, $P = 1.53 \times 10^{-8}$ and 0 d $vs.$ 6 d, $P = 6.48 \times 10^{-12}$. (C, D) Serum levels of FNDC1 in CTX-injured mice (C) and $mdx$ mice (D) ($n = 6$ mice). (C) One-way ANOVA, Uninjured $vs.$ 3 d, $P = 1.14 \times 10^{-3}$; Uninjured $vs.$ 5 d, $P = 1.79 \times 10^{-10}$ and Uninjured $vs.$ 14 d, $P = 0.2061$. (D) Two-tailed t-test, $P = 1.01 \times 10^{-5}$. (E) Representative immunofluorescence staining of MYHC (in green) in C2C12 cells differentiated for 4 days after transfection with Control (empty vector), FNDC1-FL, FNDC1-FN3, or FNDC1-PHA. Scale bar = 300 μm. (F, G) Quantification of the fusion index (a MYHC$^+$ cell with at least three nucleus) and the nucleus distribution per myotube ($n = 6$ independent experiments). (F) One-way ANOVA, Control $vs.$ FNDC1-FL, $P = 1.56 \times 10^{-13}$; Control $vs.$ FNDC1-FN3, $P = 1.49 \times 10^{-12}$; FNDC1-FL $vs.$ FNDC1-PHA, $P = 8.75 \times 10^{-13}$ and FNDC1-PHA $vs.$ FNDC1-FN3, $P = 1.06 \times 10^{-11}$. (G) One-way ANOVA, Control $vs.$ FNDC1-FL, $P = 1.04 \times 10^{-6}$; Control $vs.$ FNDC1-FN3, $P = 3.76 \times 10^{-7}$; FNDC1-FL $vs.$ FNDC1-PHA, $P = 9.10 \times 10^{-6}$; FNDC1-PHA $vs.$ FNDC1-FN3, $P = 3.06 \times 10^{-6}$ for nucleus distribution per myotube <3; Control $vs.$ FNDC1-FL, $P = 0.0132$; Control $vs.$ FNDC1-FN3, $P = 0.0295$; FNDC1-FL $vs.$ FNDC1-PHA, $P = 0.0010$; FNDC1-PHA $vs.$ FNDC1-FN3, $P = 0.0024$ for nucleus distribution per myotube 3–10; Control $vs.$ FNDC1-FL, $P = 4.65 \times 10^{-6}$; Control $vs.$ FNDC1-FN3, $P = 0.0021$; FNDC1-FL $vs.$ FNDC1-PHA, $P = 5.28 \times 10^{-5}$; FNDC1-PHA $vs.$ FNDC1-FN3, $P = 0.0024$ for nucleus distribution per myotube >10. (H) Representative immunoblotting of MYHC and MYOG in cells transfected with Control (empty vector), FNDC1-FL, FNDC1-FN3 or FNDC1-PHA ($n = 3$ independent experiments). Cells were collected at day 4 post-differentiation. (I) mRNA expression of $Caveolin$-$3$ and $Myomaker$ ($n = 6$ independent experiments). One-way ANOVA, Control $vs.$ FNDC1-FL, $P = 2.04 \times 10^{-12}$; Control $vs.$ FNDC1-FN3, $P = 1.26 \times 10^{-10}$; FNDC1-FL $vs.$ FNDC1-PHA, $P = 3.36 \times 10^{-12}$; FNDC1-PHA $vs.$ FNDC1-FN3, $P = 2.29 \times 10^{-10}$ for $Caveolin$-$3$. Control $vs.$ FNDC1-FL, $P = 3.72 \times 10^{-12}$; Control $vs.$ FNDC1-FN3, $P = 1.83 \times 10^{-12}$; FNDC1-FL $vs.$ FNDC1-PHA, $P = 4.48 \times 10^{-13}$; FNDC1-PHA $vs.$ FNDC1-FN3, $P = 2.39 \times 10^{-13}$ for $Myomaker$. (J) Coomassie-stained SDS-PAGE of protein fractions. M represents protein marker. (K) Reverse-phase high-performance liquid chromatography with absorption at 280 nm with visible single peak pattern. (L) Representative MS analysis of the recombinant mFNDC1 protein showing high purity by sum PEP score and number of unique peptides. (M) Representative immunofluorescence staining of MYHC (in green) in C2C12 cells treated with different concentrations mFNDC1 at day 4 post differentiation. Nuclei were counterstained with DAPI (in blue). Scale bars = 50 μm. (N) Quantification of differentiation index ($n = 6$ independent experiments). One-way ANOVA, 0 $vs.$ 0.01, $P = 2.55 \times 10^{-7}$; 0 $vs.$ 0.1, $P = 9.68 \times 10^{-9}$; 0 $vs.$ 1, $P = 1.21 \times 10^{-13}$. (O) Quantification of the fusion index (a MYHC$^+$ cell with at least three nucleus) ($n = 6$ independent experiments). One-way ANOVA, 0 $vs.$ 0.01, $P = 1.34 \times 10^{-7}$; 0 $vs.$ 0.1, $P = 5.30 \times 10^{-12}$; 0 $vs.$ 1, $P = 2.3 \times 10^{-14}$. Data are represented as mean ± SEM. *$p < 0.05$ and **$p < 0.01$. Source data are available online for this figure.

the myogenic differentiation (Fig. 2G–J), showing enhanced mRNA expression of $Caveolin$-$3$ and $Myomaker$ (Fig. 2K) and increased MYHC and MYOG protein levels (Fig. 2L; Appendix Fig. S3B). In addition, knockdown of $Fndc1$ by shRNA lentivirus in mouse primary myoblasts impaired myogenic differentiation, with significantly reduced fusion index (Appendix Fig. S4A–C). Together, these results suggest that FNDC1 promotes myogenic differentiation, implying a critical role for FNDC1 in myogenesis.

## Identification of FNDC1 as a novel myokine promoting myoblast differentiation

To gain further insight into FNDC1, we performed the protein sequence analysis by SignalP and DeepTMHMM, and predicted a signal peptide cleavage site and the absence of a hydrophobic transmembrane region (Appendix Fig. S5), suggesting that it is a potential secreted protein. We further detected the FNDC1 levels in different myogenic systems. During C2C12 cell differentiation, medium FNDC1 levels were significantly elevated (Fig. 3A,B), with no obvious difference in LDH release (Appendix Fig. S6A), suggesting that FNDC1 is secreted by myoblasts rather than passively released after cell death. In CTX-injured mice, circulating FNDC1 levels were increased on day 3 and further increased on day 5, then decreased on day 14 as skeletal muscle architecture was largely restored (Fig. 3C). In $mdx$ mice, serum FNDC1 levels were elevated nearly 4-fold (Fig. 3D). These results show that FNDC1 is a novel myokine elevated during myogenic differentiation.

Since the molecular weight of mouse FNDC1 is ~195 kDa and this large size may not be conducive to drug development (Fosgerau and Hoffmann, 2015; Sato et al, 2006), truncation experiments were performed to determine the functional domain of FNDC1 and to synthesize recombinant protein for peptide drug development. FNDC1 contains two types of domains: the FN3 domain and the PHA03247 domain (Appendix Fig. S7A). We constructed three vectors containing full-length FNDC1, FNDC1-FN3 (lacking residues 519–1092 of the PHA03247 domain), and FNDC1-PHA (lacking residues 38–332 and

1480–1571 of the FN3 domain) (Appendix Fig. S7A). The vectors were transfected separately into C2C12 cells. FNDC1-FN3 overexpression was sufficient to promote myotube formation (Fig. 3E–G) and increase the protein levels of MYHC and MYOG (Fig. 3H; Appendix Fig. S7B), as well as mRNA expression of $Caveolin$-$3$ and $Myomaker$ (Fig. 3I), similar to full-length FNDC1, whereas FNDC1-PHA transfection had no obvious effect on myotube formation (Fig. 3E–I), suggesting that the FN3 domain is essential for the myogenic function of FNDC1. Based on these truncation results, we then generated recombinant mouse FNDC1 (mFNDC1) containing the FN3 domain of FNDC1 to facilitate drug development. SDS-PAGE with Coomassie Brilliant Blue staining showed a single purified protein (~35 kDa) (Fig. 3J). Reverse-phase high-performance liquid chromatography exhibited a single peak for the purified recombinant protein (92% peak area) (Fig. 3K). To further identify the recombinant protein, we performed mass spectrometry for the immunoaffinity-purified, His-tagged bands. Peptides identified band mapped to FNDC1 (Fig. 3L; Appendix Table S1). These results collectively show that the purified recombinant protein is FNDC1 and the protein purity is 92%.

Functional assessment revealed that recombinant mFNDC1 increased myogenic differentiation in a dose-dependent manner (Fig. 3M–O). On day 4 after differentiation, mFNDC1 treatment increased the proportion of large myotubes (>10 nuclei) (Appendix Fig. S8A,B) and the myoblast fusion index compared to controls in C2C12 cells (Appendix Fig. S8C). Furthermore, mFNDC1 promoted the mRNA expression of $Caveolin$-$3$ and $Myomaker$ (Appendix Fig. S8D) and increased the protein levels of MYHC and MYOG (Appendix Fig. S8E), consistent with those in $Fndc1$-overexpressing cells. Overall, these results indicate that mFNDC1 retains reliable activity to promote myoblast differentiation.

## Integrin α5β1 is the receptor for FNDC1 in myoblast

To explore potential receptors for FNDC1, we employed a combination of transcriptomic and MS-based proteomic strategies. RNA-sequencing was performed in $Fndc1$ knockdown and control

**A**

| | Gene Symbol | Score Sequest HT | Peptides |
|---|---|---|---|
| 1 | FNDC1 | 35.25 | 6 |
| 2 | ITGB1 | 24.36 | 4 |
| 3 | APOA1 | 14.36 | 3 |
| 4 | ATP6V0C | 10.35 | 3 |
| 5 | C3 | 10.6 | 3 |
| 6 | CRYAB | 8.32 | 4 |
| 7 | EVPL | 5.36 | 3 |
| 8 | FGA | 3.23 | 2 |
| 9 | FGB | 4.36 | 3 |
| 10 | FGG | 5.32 | 2 |

**B**

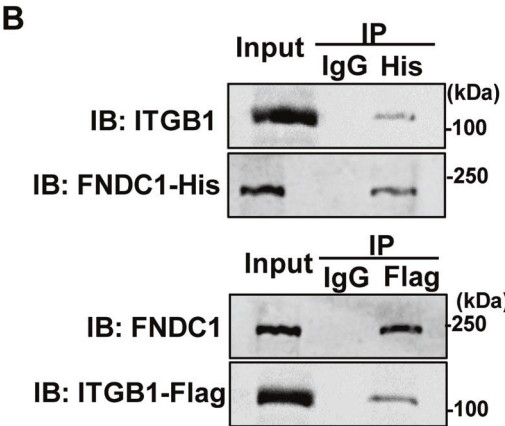

**C**

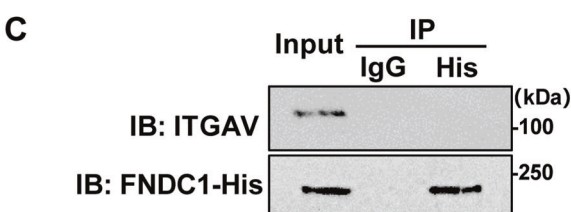

**D**

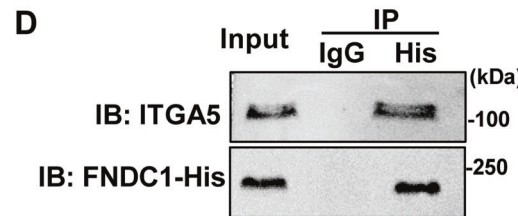

**E**

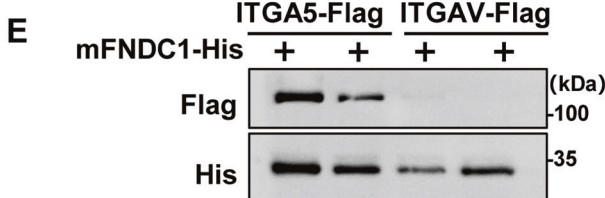

**F**

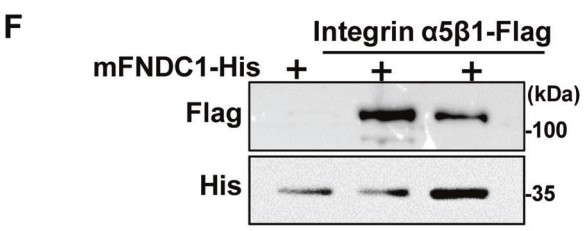

**G**

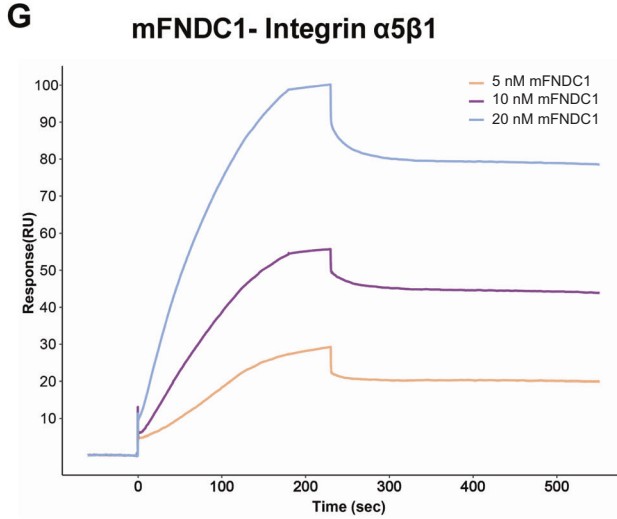

**H**

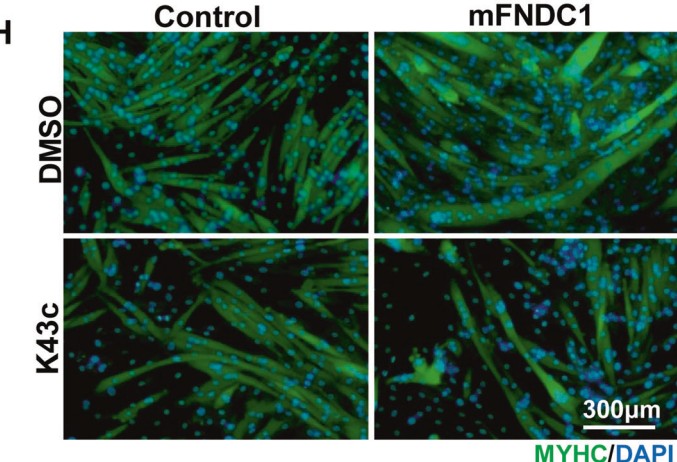

MYHC/DAPI

**I**

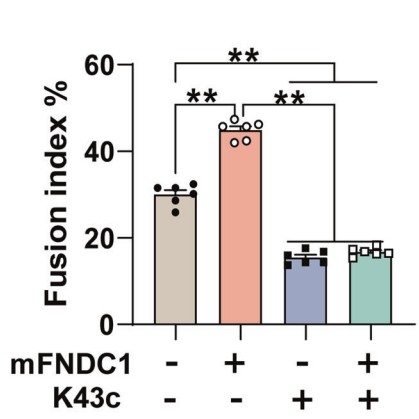

**Figure 4.  Integrin α5β1 is the receptor for FNDC1 in myoblast.**

(A) The top 10 proteins enriched with FNDC1 by cross-linking/Co-IP/MS analysis. (B) Co-IP of FNDC1 and Integrin β1 (ITGB1) in C2C12 cells. (C, D) Immunoprecipitation of FNDC1 and Integrin αv (ITGAV) or Integrin α5 (ITGA5) in C2C12 cells. (E, F) Pull-down assays. 100 nM His-tagged FNDC1 (truncated recombinant FNDC1 protein) was incubated with the indicated Flag-tagged integrin (5 nM) and then affinity adsorbed through a nickel column. Immunoblotting was performed to analyze protein interactions between integrins (α5 or αv) and mFNDC1 (E) or to analyze protein interaction between integrin α5β1 and mFNDC1 (F). (G) Surface plasmon resonance analysis of the FNDC1-Intergin α5β1 interaction. Numbers above curves indicate the analyte concentration tested. All experiments were performed in triplicate with a mobile phase of HBS-EP. (H) Representative immunofluorescence staining of MYHC (in green) in C2C12 cells treated with Control or mFNDC1 4 days post-differentiation in the presence or absence of K43c. Scale bars = 300 µm. (I) Quantification of fusion index (a MYHC$^+$ cell with at least three nucleus) ($n$ = 6 independent experiments). One-way ANOVA, Control *vs.* mFNDC1, $P$ = 8.87 × 10$^{-11}$; Control *vs.* K43c, $P$ = 1.15 × 10$^{-10}$; Control *vs.* mFNDC1 + K43c, $P$ = 6.10 × 10$^{-10}$; mFNDC1 *vs.* K43c, $P$ = 2.30 × 10$^{-14}$; mFNDC1 *vs.* mFNDC1 + K43c, $P$ = 2.30 × 10$^{-14}$. Data are represented as mean ± SEM. **$p$ < 0.01. Source data are available online for this figure.

C2C12 cells on day 4 after differentiation. Gene ontology (GO) analysis revealed that the DEGs had significant enrichment in integrin-mediated cell adhesion, an important biological process during myoblast differentiation (Appendix Fig. S9A). In addition, we used PathwayNet (http://pathwaynet.princeton.edu/) (Park et al, 2015) analysis to screen for potential co-complex partners of FNDC1 and found that integrin beta1 (ITGB1) cell surface interactions were significantly enriched (Appendix Fig. S9B). We further performed MS-based proteomics and showed that ITGB1, a vital membrane receptor involved in cell adhesion and muscle development, is a potential candidate receptor for FNDC1 (Fig. 4A). Co-immunoprecipitation revealed direct physical interaction between FNDC1 and ITGB1 in C2C12 cells (Fig. 4B). Ligand binding to integrins typically initiates canonical signaling through phosphorylation of focal adhesion kinase (FAK). We thus assessed FAK phosphorylation and observed reduced levels in *Fndc*1 knockdown C2C12 cells (Appendix Fig. S9C), whereas mFNDC1 addition increased FAK phosphorylation in C2C12 cells (Appendix Fig. S9D). Knockdown of *Itgb1* inhibited mFNDC1-induced myogenic cell fusion (Appendix Fig. S10A,B), multinucleated myotube formation (Appendix Fig. S10C), and the elevations in MYHC and MYOG protein as well as p-FAK in myoblasts (Appendix Fig. S10D,E). In addition, we tested whether the ITGB1 inhibitor RGDs (Rozo et al, 2016) is able to disrupt the effects of FNDC1. Treatment of C2C12 cells with RGDs abolished the promoting effect of mFNDC1 on myogenic differentiation (Appendix Fig. S10F–H). These results suggest that ITGB1 could be a receptor for FNDC1.

There are two ITGB1 isotypes ITGB1A and ITGB1D (Zhang et al, 2007), which we found differentially expressed in myoblasts and myotubes (Appendix Fig. S11A). To determine which ITGB1 subtype interacts with FNDC1, we performed immunoprecipitation in C2C12 cells at day 3 post differentiation (both ITGB1A and ITGB1D were detectable at this time point). The result showed that FNDC1 physically interacts with either ITGB1A or ITGB1D (Appendix Fig. S11B). However, knockdown of ITGB1D, but not ITGB1A, in C2C12 cells significantly reduced FNDC1-induced MYHC protein levels (Appendix Fig. S11C,D), suggesting that FNDC1 promotes myoblast differentiation by physical interaction with ITGB1D. It is well accepted that integrins are cell surface receptors composed of α and β subunits that form heterodimers (Hynes, 2002). Our results of the FNDC1 protein truncation test and pharmacological blockade of integrins revealed that the RGD-receptors are candidate receptors for pairing with the β1 subunit (Fig. 3E–I; Appendix Fig. S10F–H). To further determine which integrin α subunit pairs with the integrin β1 for FNDC1 activation, we examined the expression of the RGD-

receptors in skeletal muscle. RNA-seq analysis and PCR validation revealed that the α5 and αv, which are part of the RGD-receptors (Hynes, 2002) expressed in skeletal muscle, were potential pairs with FNDC1 (Appendix Fig. S11E,F). Immunoprecipitation analysis revealed physical interaction between FNDC1 and integrin α5, but not αv, in C2C12 cells (Fig. 4C,D), implying that integrin α5β1 complex serves as the receptor for FNDC1. This result was also confirmed in primary myoblasts (Appendix Fig. S12). Furthermore, by performing extracellular pull-down assay and surface plasmon resonance (SPR), we showed direct binding of FNDC1 to integrin α5β1 (Fig. 4E–G). Finally, to examine whether integrin α5β1 functions as a receptor for FNDC1 in skeletal muscle, we treated C2C12 cells with K34c, a specific inhibitor of integrin α5β1 (Martinkova et al, 2010), and found abolished effect of FNDC1 on promoting myogenic differentiation (Fig. 4H,I). These results collectively demonstrate that the integrin complex α5β1 binds to FNDC1 in skeletal muscle.

## FNDC1 binds to integrin α5β1 to promote myogenesis and activate the FAK/PI3K/AKT/mTOR pathway

To further investigate the downstream pathways involved in FNDC1-mediated myogenesis, KEGG pathway enrichment analysis was performed on the identified DEGs from *Fndc1* KD C2C12 cells. The result indicated that *Fndc1* knockdown markedly altered the PI3K/AKT and the mTOR signaling pathways (Fig. 5A), typical pathways in myogenesis. We then examined the critical proteins of the pathway and showed reduced phosphorylation of AKT and mTOR in *Fndc1* knockdown C2C12 cells at day 4 post-differentiation (Fig. 5B,C), whereas mFNDC1 treatment increased their expressions (Fig. 5D). Downstream of FNDC1, we found that knockdown of *Itgb1* or pharmacological inhibition of α5β1 by K34c completely blocked FNDC1-induced phosphorylation of AKT and mTOR in C2C12 cells (Appendix Fig. S13A–D). Similarly, knockdown *Fak* inhibited mFNDC1-induced phosphorylation of AKT and mTOR in C2C12 cells (Appendix Fig. S14; Fig. 5E). Moreover, treatment of C2C12 cells with LY294002 (a PI3K inhibitor) significantly inhibited mFNDC1-induced phosphorylation of AKT and mTOR, without affecting the phosphorylation of FAK (Fig. 5F,G). Immunofluorescence staining showed that treatment of C2C12 cells with LY294002 abolished the promoting effect of mFNDC1 on myogenic differentiation (Fig. 5H,I). These results suggest a direct involvement of the PI3K/AKT/mTOR signaling pathway in FNDC1-mediated myogenesis. In conclusion, FNDC1 regulates myogenic differentiation by binding to integrin α5β1 which subsequently activates the FAK/PI3K/AKT/mTOR pathway.

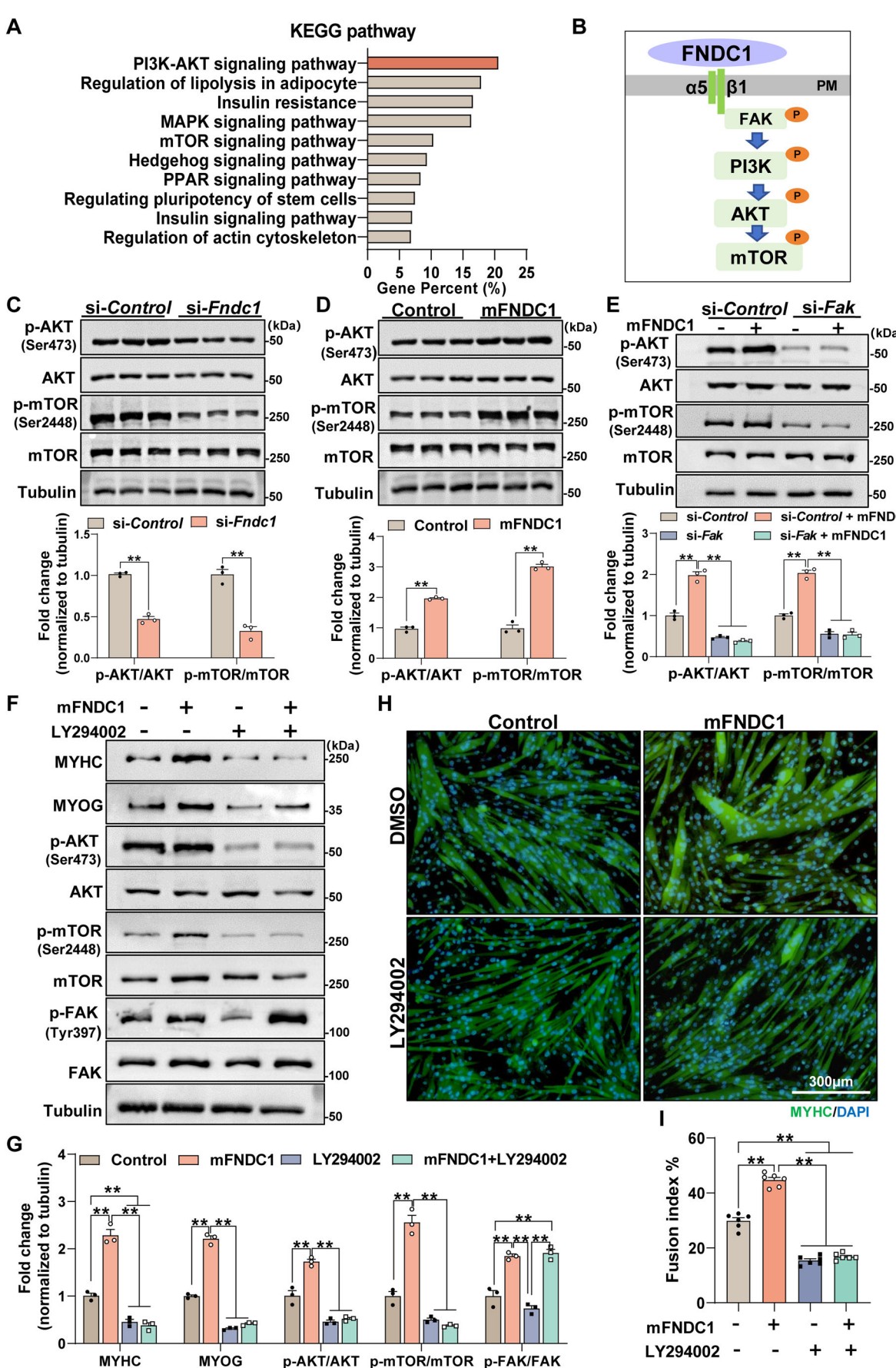

**Figure 5.  FNDC1 improves myogenesis through activating the Integrin/FAK/AKT/mTOR pathway.**

(A) The KEGG signaling pathways in *Fndc1* knockdown and control C2C12 cells after 4 days of differentiation. (B) Model of typical integrin signaling. Ligand FNDC1 binds to integrin, resulting in the phosphorylation of FAK (Tyr397) followed by phosphorylation of PI3K, AKT, and mTOR. PM, plasma membrane. (C, D) Representative immunoblotting and quantification of total and phosphorylated AKT (Ser473) and mTOR (Ser2448) in si-*Fndc1* or mFNDC1-treated C2C12 cells ($n = 3$ independent experiments). (C) Two-tailed t-test, $P = 1.07 \times 10^{-4}$ and $P = 1.05 \times 10^{-3}$ for p-AKT/AKT and p-mTOR/mTOR, respectively. (D) Two-tailed t-test, $P = 8.40 \times 10^{-5}$ and $P = 1.02 \times 10^{-4}$ for p-AKT/AKT and p-mTOR/mTOR, respectively. (E) Representative immunoblotting and quantification of total and phosphorylated AKT (Ser473) and mTOR (Ser2448) in si-control or si-*Fak* C2C12 cells treated with Control or mFNDC1 ($n = 3$ independent experiments). One-way ANOVA, si-Control *vs.* si-Control +mFNDC1, $P = 5.67 \times 10^{-6}$; si-Control+mFNDC1 *vs.* si-*Fak*, $P = 1.65 \times 10^{-7}$; si-Control+mFNDC1 *vs.* si-*Fak* + mFNDC1, $P = 9.54 \times 10^{-8}$ for p-AKT/AKT; si-Control *vs.* si-Control+mFNDC1, $P = 5.46 \times 10^{-6}$; si-Control+mFNDC1 *vs.* si-*Fak*, $P = 3.08 \times 10^{-7}$; si-Control+mFNDC1 *vs.* si-*Fak* + mFNDC1, $P = 2.81 \times 10^{-7}$ for p-mTOR/mTOR. (F, G) Representative immunoblotting analysis and quantification of indicated proteins in control (DMSO) or LY294002 (PI3Ks inhibitor) treated C2C12 cells receiving control or mFNDC1 ($n = 3$ independent experiments). One-way ANOVA, Control *vs.* mFNDC1, $P = 1.10 \times 10^{-5}$; Control *vs.* LY294002, $P = 0.0039$; Control *vs.* mFNDC1 + LY294002, $P = 0.0018$, mFNDC1 *vs.* LY294002, $P = 6.88 \times 10^{-7}$, mFNDC1 *vs.* mFNDC1 + LY294002, $P = 4.95 \times 10^{-7}$ for MYHC; Control *vs.* mFNDC1, $P = 4.68 \times 10^{-8}$, mFNDC1 *vs.* LY294002, $P = 3.60 \times 10^{-9}$, mFNDC1 *vs.* mFNDC1 + LY294002, $P = 5.28 \times 10^{-9}$ for MYOG; Control *vs.* mFNDC1, $P = 1.93 \times 10^{-4}$, mFNDC1 *vs.* LY294002, $P = 2.81 \times 10^{-6}$, mFNDC1 *vs.* mFNDC1 + LY294002, $P = 4.11 \times 10^{-6}$ for p-AKT/AKT; Control *vs.* mFNDC1, $P = 1.03 \times 10^{-5}$, mFNDC1 *vs.* LY294002, $P = 1.25 \times 10^{-6}$, mFNDC1 *vs.* mFNDC1 + LY294002, $P = 7.72 \times 10^{-7}$ for p-mTOR/mTOR; Control *vs.* mFNDC1, $P = 3.23 \times 10^{-4}$, Control *vs.* mFNDC1 + LY294002, $P = 1.92 \times 10^{-4}$, mFNDC1 *vs.* LY294002, $P = 4.62 \times 10^{-5}$, LY294002 *vs.* mFNDC1 + LY294002, $P = 3.03 \times 10^{-5}$ for p-FAK/FAK. (H) Representative immunofluorescence staining of MYHC (in green) in control (DMSO) or LY294002 treated C2C12 cells receiving Control or mFNDC1 for 4 days post differentiation. Nuclei are counterstained with DAPI (in blue). Scale bars = 300 μm. (I) Quantification of fusion index (a MYHC$^+$ cell with at least three nucleus) ($n = 6$ independent experiments). One-way ANOVA, Control *vs.* mFNDC1, $P = 4.87 \times 10^{-10}$; Control *vs.* LY294002, $P = 7.98 \times 10^{-10}$; Control *vs.* mFNDC1 + LY294002, $P = 4.73 \times 10^{-9}$, mFNDC1 *vs.* LY294002, $P = 2.5 \times 10^{-14}$ and mFNDC1 *vs.* mFNDC1 + LY294002, $P = 2.7 \times 10^{-14}$. Data are represented as mean ± SEM. **$p < 0.01$. Source data are available online for this figure.

## FNDC1 promotes skeletal muscle regeneration in CTX-induced muscle injured mice

Since myogenic differentiation is an important step in muscle regeneration, we explored the function of FNDC1 in CTX-induced muscle regeneration in mice. CTX-treated mice were given mFNDC1 (2.5 mg/kg body weight, every 2 days for 14 days) by intramuscular injection (Fig. 6A). By day 5 post-injury, control mice showed inflammatory infiltrate and numerous small myofibers with centralized nuclei (Appendix Fig. S15; Fig. 6B–E), indicative of regeneration. In contrast, inflammatory cells were almost cleared in mFNDC1-treated mice, with more newly formed myofibers (centralized nuclei) having bigger diameters than in control mice (Appendix Fig. S15; Fig. 6B–E). Immunofluorescence staining of eMYHC, a marker for newly formed muscle fibers during myogenesis and muscle regeneration, revealed significantly increased positive staining in mFNDC1-treated mice at day 5 post-injury (Fig. 6C), with more myofibers containing ≥ two condensed myonuclear (Fig. 6F), indicating increased fusion indices and differentiation. Consistent with this, mRNA expression of *Myh3*, *Myog*, and *Myod* was significantly elevated in mFNDC1-treated group (Fig. 6G). Although a substantial portion of the muscle architecture was restored on day 14 post-injury in both groups of mice, mFNDC1-treated mice had larger myofibers than controls (Fig. 6B). We also explored the role of FNDC1 in CTX-induced regeneration in *Fndc1* knockdown mice. Compared to scramble shRNA muscles, the number of eMYHC$^+$ regenerating fibers and average myofiber size was significantly reduced in AAV-sh*Fndc1* muscles (Appendix Fig. S16A–E), along with significantly lower protein levels of eMYHC (Appendix Fig. S16F), as well as mRNA levels of *Myog* and *Myod* (Appendix Fig. S16G). Together, these data suggest that FNDC1 promotes skeletal muscle regeneration in mice.

Given that activation of the quiescent satellite cells, proliferation of the activated satellite cells (myoblasts), as well as differentiation and fusion of myoblasts are essential for muscle regeneration, we next assessed the effect of FNDC1 on these events of satellite cells

during skeletal muscle regeneration. In vivo assessment of the effects of FNDC1 on the proliferation and differentiation of satellite cells in TA muscle by immunofluorescence staining showed that FNDC1 does not affect the percentage of PAX7$^+$MYOD$^-$ and PAX7$^+$MYOD$^+$ cells from CTX-injured mice at day 3 post-injury (Appendix Fig. S17A–C). However, at day 5 post-injury, mFNDC1 treatment decreased the percentage of PAX7$^+$MYOD$^+$ cells while increased that of the PAX7$^-$MYOD$^+$ cells (myogenic differentiation) (Appendix Fig. S17D; Fig. 6H). In contrast, opposite patterns were observed when *Fndc1* was knockdown (Appendix Fig. S17D; Fig. 6I). These results suggest that FNDC1 does not affect satellite cell proliferation during the early stage of muscle regeneration, but promotes satellite cell differentiation during the stage of myofiber remodeling in muscle-injured mice.

We further assessed the effect of FNDC1 on the proliferation of cultured satellite cells (myoblasts) by EdU staining, showing similar proliferation between mFNDC1-treated cells and controls (Appendix Fig. S18A,B). Meanwhile, we observed similar proliferation of satellite cells treated with sh*Fndc1* and control shRNA (Appendix Fig. S18A,C). Consistent with the in vivo results (day 3 post-injury) (Appendix Fig. S17A–C), these results collectively demonstrate that FNDC1 does not affect the proliferation of satellite cells.

To further determine the effect of FNDC1 on satellite cells differentiation, we performed immunofluorescence staining for MYOG in mFNDC1-treated satellite cells and showed increased percentage of MYOG$^+$ cells (Fig. 6J). In contrast, the percentage of MYOG$^+$ cells were decreased in *Fndc1* knockdown satellite cells (Fig. 6J). These results corroborate that FNDC1 promotes satellite cells differentiation. Finally, we performed immunofluorescence staining for MYHC, and the results showed that mFNDC1 treatment promoted myogenic differentiation of satellite cells (Fig. 6K; Appendix Fig. S18D,E), with increased mRNA expression of *Caveolin-3* and *Myomaker* (Appendix Fig. S18F). In contrast, the myogenic differentiation was inhibited in *Fndc1* knockdown satellite cells (Fig. 6K; Appendix Fig. S18G,H), with reduced mRNA expression of *Caveolin-3* and *Myomaker* (Appendix Fig. S18I). Collectively, these results show that FNDC1 does not

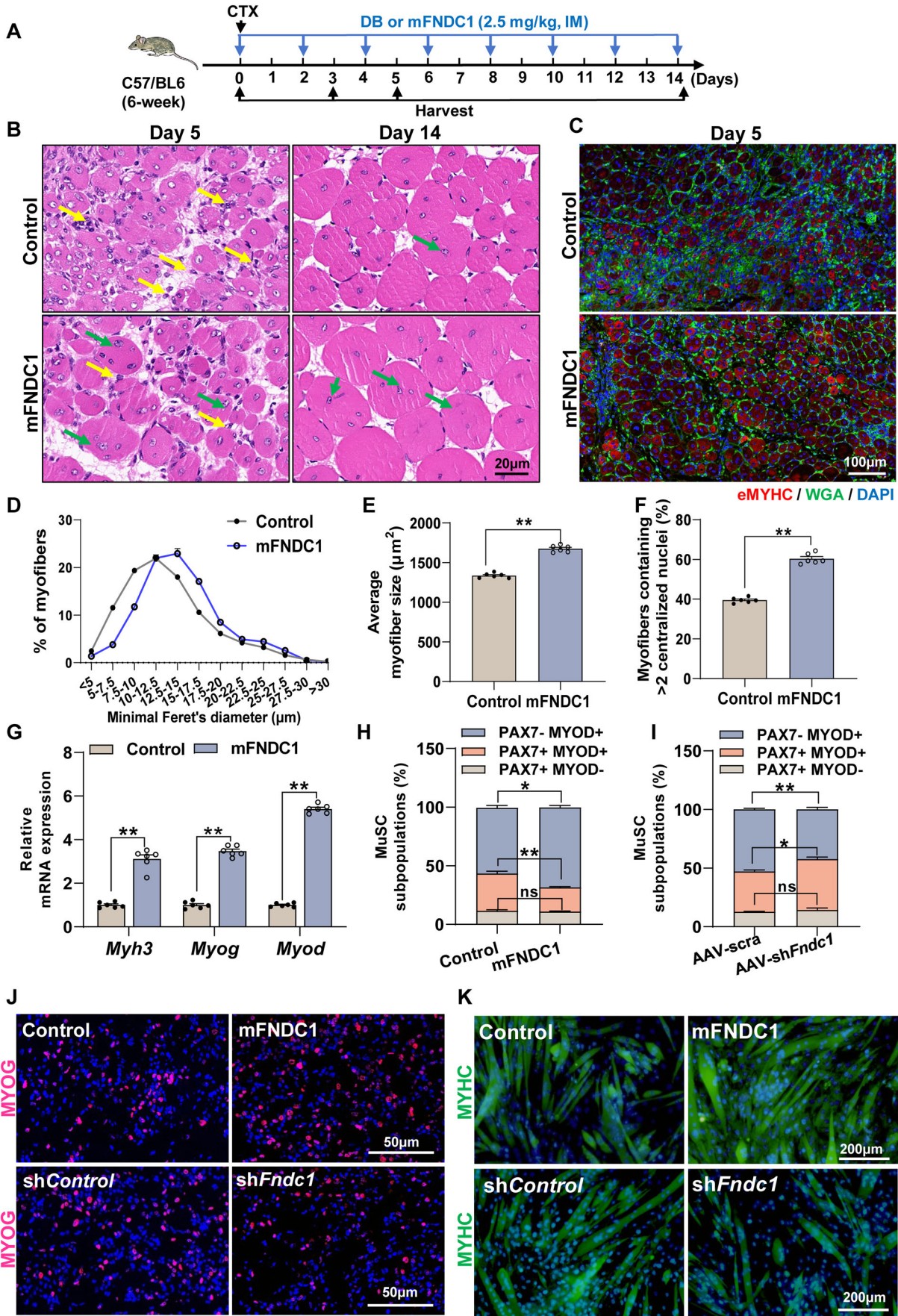

**Figure 6.  FNDC1 promotes skeletal muscle regeneration in CTX-induced muscle injury in mice.**

(**A**) Experimental procedures. CTX-treated mice (6-week-old C57BL/6J) were either given Control (dialysis buffer, DB) or mFNDC1 (2.5 mg/kg body weight) every 2 days via intramuscular injection (IM) for two weeks starting with the treatment of CTX. (**B**) Representative H&E staining of TA muscle sections at days 5 and 14 post-injury in Control or mFNDC1-treated mice ($n = 6$ mice). Yellow arrows indicate inflammatory infiltrates and green arrows indicate newborn muscle fibers. Scale bar = 20 μm. (**C**) Representative immunofluorescence staining of eMYHC$^+$ fibers in Control or mFNDC1-treated TA muscle at day 5 post-injury ($n = 6$ mice). Staining for eMYHC (in red) marks the newborn myofibrils. Cell membrane was stained with WGA (in green) and nuclei were counterstained with DAPI (in blue). Scale bar = 100 μm. (**D, E**) Frequency distribution of minimal Feret's diameter and average CSA of eMYHC$^+$ myofiber in TA muscle from CTX-injured mice treated with Control or mFNDC1 at day 5 post-injury ($n = 6$ mice). (**E**) Two-tailed t-test, $P = 1.58 \times 10^{-8}$. (**F**) Percentage of newly formed myofibers containing two or more central nuclei in muscle fibers from CTX-injured mice at day 5 ($n = 6$ mice). Two-tailed t-test, $P = 8.40 \times 10^{-9}$. (**G**) mRNA expression of *Myh3*, *Myog*, and *Myod* ($n = 6$ mice). Two-tailed t-test, $P = 6.12 \times 10^{-7}$, $P = 1.01 \times 10^{-9}$, and $P = 2.30 \times 10^{-13}$ for *Myh3*, *Myog*, and *Myod*, respectively. (**H, I**) Quantification of different states of satellite cell (MuSC) in TA muscle from CTX-injured mice receiving mFNDC1 or AAV-sh*Fndc1* at day 5 post-injury ($n = 6$ mice). (**H**) Two-tailed t-test, $P = 0.5802$, $P = 0.0060$, and $P = 0.0128$ for *PAX7*$^+$*MYOD*$^-$, *PAX7*$^+$*MYOD*$^+$, and *PAX7*$^-$*MYOD*$^+$, respectively. (**I**) Two-tailed t-test, $P = 0.4170$, $P = 0.0137$, and $P = 0.0077$ for *PAX7*$^-$*MYOD*$^-$, *PAX7*$^+$*MYOD*$^+$, and *PAX7*$^-$*MYOD*$^+$, respectively. (**J**) Immunofluorescence staining for MYOG (in red) in satellite cell-derived primary myoblasts after differentiation for 36 h. Isolated satellite cell from WT mice treated with recombinant proteins (Control or mFNDC1) or lentivirus (sh*Control* or sh*Fndc1*) were induced to differentiate for 36 h. Scale bars = 50 μm. (**K**) Representative immunofluorescence staining of MYHC (in green) in satellite cell-derived primary myoblasts treated with mFNDC1 or sh*Fndc1* at day 3 post-differentiation ($n = 6$ independent experiments). Scale bars = 200 μm. Data are represented as mean ± SEM. *$p < 0.05$ and **$p < 0.01$. Source data are available online for this figure.

affect satellite cells proliferation, but facilitates myocyte fusion and multinucleated myotube formation.

## ITGB1 is essential for FNDC1-promoted skeletal muscle regeneration in CTX-induced muscle injury

To test whether FNDC1-promoted muscle regeneration depends on the integrin receptor by a similar mechanism as we showed in the cellular level, we generated AAV-sh*Itgb1*, the core subunit of the integrin heterodimers α5β1 for the activation of FAK (Kuppuswamy, 2002; Quach and Rando, 2006), in mouse skeletal muscle. The TA muscle receiving AAV scramble shRNA or sh*Itgb1* was subjected to a single CTX injury followed by mFNDC1 treatment every 2 days for a fortnight. In AAV-scramble mice, mFNDC1 treatment promoted larger newborn muscle fibers than dialysis buffer controls (Fig. 7A–D). Nonetheless, in AAV-sh*Itgb1* mice, mFNDC1-treated group showed similar area of new muscle fibers as dialysis buffer controls (Fig. 7A–D), indicating that the promotive effect of mFNDC1 on muscle regeneration was eliminated upon *Itgb1* knockdown. mFNDC1 treatment increased eMYHC protein levels and phosphorylation of FAK, AKT, and mTOR in AAV-scramble mice, while these changes were abolished following *Itgb1* knockdown (Fig. 7E,F). Together, these results indicate that, in agreement with the in vitro results, ITGB1 is essential for FNDC1-mediated muscle regeneration in vivo.

## FNDC1 ameliorates the pathological phenotype in *mdx* mice

The beneficial effect of FNDC1 on CTX-induced muscle regeneration prompted us to test whether it is also beneficial for impaired regeneration-induced muscular dystrophy. For this systemic treatment, we first assessed the pharmacokinetics and pharmacodynamics of mFNDC1 were assessed. For pharmacokinetics, a single dose of mFNDC1 (2.5 mg/kg body weight) was injected intraperitoneally into C57BL/6J mice. The serum mFNDC1 concentrations reached $C_{max}$ (50.36 ± 2.69 mg/kg) at 30 min and then slowly decreased with a half-life of ~40 min (Appendix Fig. S19A; Table S2). Pharmacodynamics of mFNDC1 was determined by the activation of integrin-FAK signaling in skeletal muscle. Mice were given intraperitoneal injection of 2.5 mg/kg body weight of mFNDC1, and TA muscles were harvested at the indicated time points for signal transduction assays. We showed

activation of integrin-FAK pathway by mFNDC1 at 30 min (Appendix Fig. S19B).

We next explored the therapeutic potential of mFNDC1 for muscular dystrophy by injecting 4-week-old *mdx* mice with mFNDC1 (2.5 mg/kg body weight) every 2 days for 4 weeks (Fig. 8A). Systemic administration of mFNDC1 mitigated the loss of body weight in *mdx* mice (Appendix Fig. S20A). Compared with *mdx* control group, mFNDC1 treatment markedly increased the muscle weight of tibialis anterior, gastrocnemius, and quadriceps (Appendix Fig. S20B–D). H&E staining showed that mFNDC1 treatment decreased the number of myofibers with smaller cross-sectional areas (Fig. 8B) and increased the average myofiber area in *mdx* mice (Fig. 8B). In line with this, mFNDC1 treatment significantly increased the mRNA expression of *Myog* and *Myh3* in *mdx* mice (Fig. 8C). These results indicate that mFNDC1 treatment stimulates muscle regeneration in *mdx* mice. Subsequently, we examined the effect of FNDC1 on inflammation of *mdx* mice and observed significantly reduced circulating levels of creatine phosphate kinase (CK) (Fig. 8D), pro-inflammatory factor TNFα (Fig. 8E), IL-1β (Fig. 8F), and IL-6 (Fig. 8G), indicating improvement of systemic inflammation with mFNDC1 treatment. Moreover, the mRNA expression of *Nfkb1*, *Tnfα*, and *Il1α* was significantly reduced in skeletal muscle of *mdx* mice receiving mFNDC1 (Appendix Fig. S20E). Immunofluorescence staining showed reduced proportion of F4/80$^+$ macrophages in the TA muscle of mFNDC1-treated *mdx* mice (Appendix Fig. S20F,G). These results suggest that FNDC1 counteracts skeletal muscle inflammation in *mdx* mice. In addition, Sirius red staining of muscle sections for TA, QUA, and DIA showed significantly reduced proportions of interstitial fibrosis areas in mFNDC1-treated *mdx* mice (Fig. 8H). Along with this, mFNDC1 treatment significantly reduced TA muscle fibrosis, as shown by Masson's trichrome staining (Appendix Fig. S20H,I) and mRNA expression of fibrosis marker genes *Col1a1*, *Col3a1*, *Col5a3,* and *Fn1* (Fig. 8I). Immunofluorescence staining of IgM, a marker of muscle tissue necrosis (Podkalicka et al, 2020), showed significantly reduced necrotic fibers in TA muscles of *mdx* mice treated with mFNDC1 (Appendix Fig. S20J,K). These results showed that FNDC1 treatment increases muscle membrane integrity and reduces muscle fibrosis and necrosis in *mdx* mice. Collectively, FNDC1 ameliorated the pathological phenotype of *mdx* mice.

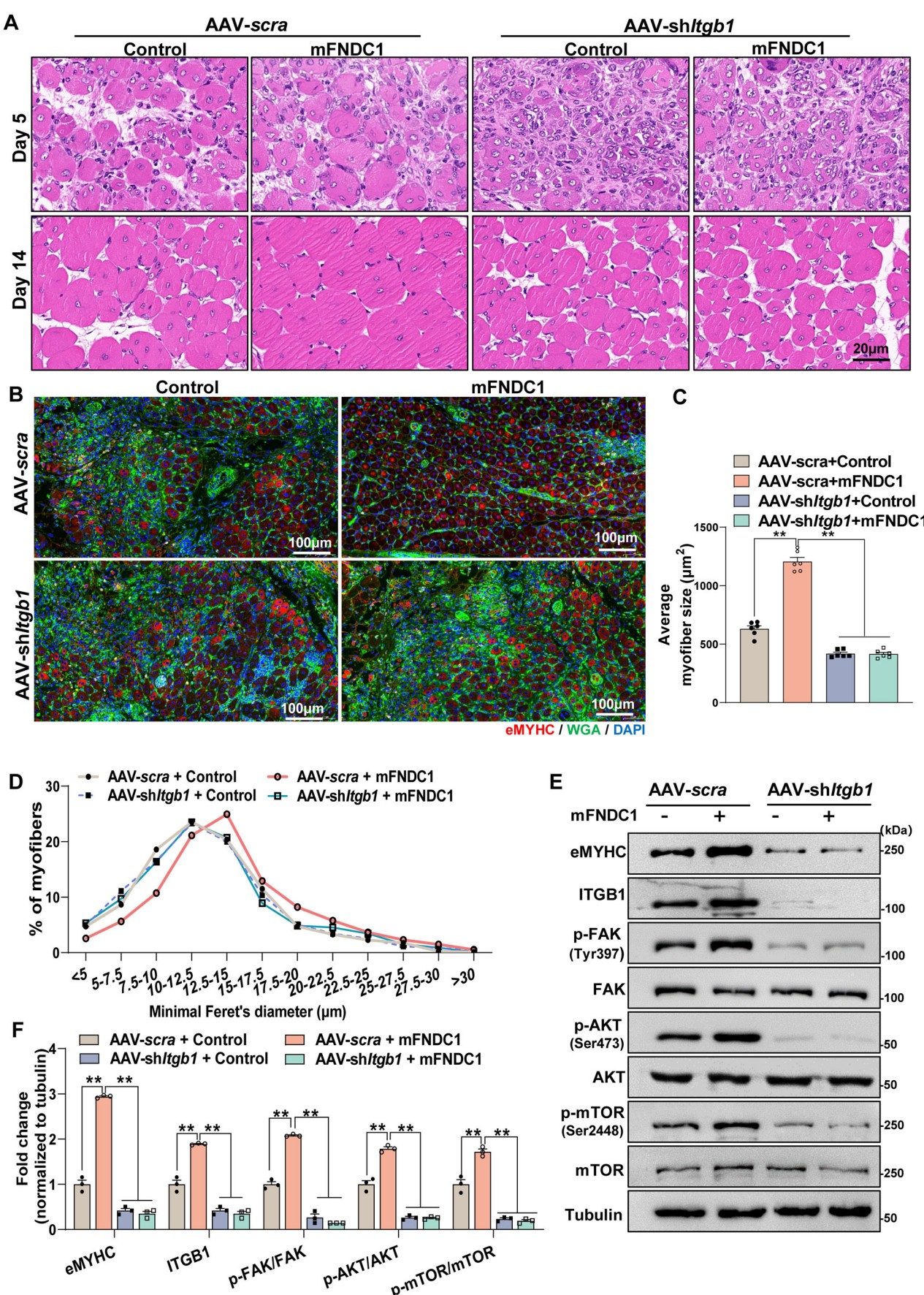

**Figure 7. ITGB1 is essential for FNDC1-promoted skeletal muscle regeneration in CTX-induced muscle injury in mice.**

(A) Representative H&E staining of TA muscle from AAV-*Scra* or AAV-sh*Itgb1* mice at days 5 and 14 post-injury ($n = 6$ mice). TA muscle from AAV-*Scra* or AAV-sh*Itgb1* mice were either given Control or mFNDC1 (2.5 mg/kg body weight) every 2 days via intramuscular injection starting with treatment with CTX. Scale bars = 20 µm. (B) Representative immunofluorescence staining of eMYHC$^+$ fibers in TA muscle at day 5 post-injury ($n = 6$ mice). Staining for eMYHC (in red) marks the newborn myofibrils. Cell membrane was stained with WGA (in green) and nuclei were counterstained with DAPI (in blue). Scale bars = 100 µm. (C, D) Average CSA and frequency distribution of eMYHC$^+$ myofiber minimal Feret's diameter in TA muscle from *Itgb1* knockdown mice treated with Control or mFNDC1 at day 5 post-injury ($n = 6$ mice). One-way ANOVA, AAV-scra+Control *vs.* AAV-scra+mFNDC1, $P = 1.03 \times 10^{-12}$; AAV-scra+mFNDC1 *vs.* AAV-sh*Itgb1*+Control, $P = 2.6 \times 10^{-14}$ and AAV-scra+mFNDC1 *vs.* AAV-sh*Itgb1* + mFNDC1, $P = 2.6 \times 10^{-14}$. (E, F) Representative immunoblotting (E) and quantification (F) of indicated proteins in TA muscle at day 5 post-injury ($n = 3$ mice). One-way ANOVA, AAV-scra+Control *vs.* AAV-scra+mFNDC1, $P = 4.30 \times 10^{-8}$; AAV-scra+mFNDC1 *vs.* AAV-sh*Itgb1*+Control, $P = 1.04 \times 10^{-8}$ and AAV-scra+mFNDC1 *vs.* AAV-sh*Itgb1* + mFNDC1, $P = 9.06 \times 10^{-9}$ for eMYHC. AAV-scra+Control *vs.* AAV-scra+mFNDC1, $P = 2.27 \times 10^{-5}$; AAV-scra+mFNDC1 *vs.* AAV-sh*Itgb1*+Control, $P = 4.64 \times 10^{-7}$ and AAV-scra+mFNDC1 *vs.* AAV-sh*Itgb1* + mFNDC1, $P = 3.13 \times 10^{-7}$ for ITGB1; AAV-scra+Control *vs.* AAV-scra+mFNDC1, $P = 1.42 \times 10^{-6}$; AAV-scra+mFNDC1 *vs.* AAV-sh*Itgb1*+Control, $P = 2.13 \times 10^{-8}$ and AAV-scra+mFNDC1 *vs.* AAV-sh*Itgb1* + mFNDC1, $P = 1.55 \times 10^{-8}$ for p-FAK/FAK; AAV-scra+Control *vs.* AAV-scra+mFNDC1, $P = 1.58 \times 10^{-5}$; AAV-scra+mFNDC1 *vs.* AAV-shItgb1+Control, $P = 7.10 \times 10^{-8}$ and AAV-scra+mFNDC1 *vs.* AAV-sh*Itgb1* + mFNDC1, $P = 6.80 \times 10^{-8}$ for p-AKT/AKT; AAV-scra+Control *vs.* AAV-scra+mFNDC1, $P = 1.89 \times 10^{-4}$; AAV-scra+mFNDC1 *vs.* AAV-sh*Itgb1*+Control, $P = 8.23 \times 10^{-7}$ and AAV-scra+mFNDC1 *vs.* AAV-sh*Itgb1* + mFNDC1, $P = 6.54 \times 10^{-7}$ for p-mTOR/mTOR. Data are represented as mean ± SEM. **$p < 0.01$. Source data are available online for this figure.

## FNDC1 improves exercise performance and muscle strength in *mdx* mice

Having observed improvement in pathological phenotype by FNDC1 treatment in *mdx* mice, we further assessed the effects of FNDC1 on motor performance and muscle function since these are common manifestations of Duchenne muscular dystrophy patients. Compared with WT controls, *mdx* mice showed increased hindlimb stride width (Fig. 9A) and decreased stride length (Fig. 9B). mFNDC1 treatment markedly ameliorated these abnormal behaviors in *mdx* mice (Fig. 9A,B). Running time (Fig. 9C) and distance (Fig. 9D) were significantly lower in *mdx* mice than in WT group, whereas mFNDC1 treatment increased running time (Fig. 9C) and the average distance in *mdx* mice (Fig. 9D). These results indicate that FNDC1 treatment improved the locomotor ability of *mdx* mice. To determine whether improvement in exercise capacity is due to enhanced muscle function, muscle function was measured in *mdx* mice. In vivo, FNDC1 treatment partially restored muscle function, with an ~35% increase in endurance (Fig. 9E) and ~31% increase in grip strength (Fig. 9F). In vitro, assessment of contractile properties in TA muscle showed significantly increased twitch force and tetanic force from mFNDC1-treated *mdx* mice (Fig. 9G,H; Appendix Fig. S21A,B). To further determine whether the force enhancement is due to increased muscle mass, we normalized the force to muscle CSA to obtain the specific force (i.e., mN • mm$^{-2}$), and showed increased twitch and tetanic forces in TA muscle from FNDC1-treated *mdx* mice (Fig. 9I,J). These results indicate that FNDC1 increases motor ability and absolute muscle strength in *mdx* mice.

## Discussion

In this study, by informatic analysis and experimental validation, we identified FNDC1 as a novel myogenic regulator that promotes myogenic differentiation and muscle regeneration. Mechanistically, by binding to the integrin receptor α5β1, FNDC1 activates the FAK/PI3K/AKT/mTOR pathway. We, for the first time, revealed FNDC1 as a novel myokine. To facilitate its potential translation, we generated a recombinant mouse FNDC1 (mFNDC1) that retains the reliable activity of FNDC1 in promoting myogenic differentiation. mFNDC1 accelerates the regeneration process in CTX-induced muscle injury and ameliorates muscular dystrophy in *mdx* mice. Thus, our results provide mechanistic insights into the role of FNDC1 in regulating myogenesis and contributing to muscle regeneration, promoting that mFNDC1 holds promise as a therapeutic agent for the treatment of acute and chronic muscle disease.

Although both FNDC1 and FNDC5/irisin are myokines, their secretion patterns and receptors are different. FNDC1 is produced in response to myogenesis or muscle regeneration, whereas FNDC5 is an exercise-induced myokine (Ma et al, 2021). Previous studies have shown that FNDC4 and FNDC5 need to be cleaved before being released into circulation (Fruhbeck et al, 2020; Kim et al, 2018). However, FNDC1 structurally has a typical secretory peptide and absence of a hydrophobic transmembrane region. Similar molecular weight of FNDC1 was shown in culture medium and cell lysate by protein blotting, indicating a non-cleavage secretion. Future studies exploring the mechanism of FNDC1 secretion should provide further clarification. Functionally, FNDC1 and its two family members FNDC5 and FNDC4 are able to promote myogenic differentiation (Li et al, 2020; Reza et al, 2017). Deciphering the mechanism underlying FNDC1-promoted myogenesis reveals the activation of integrin α5β1-FAK/PI3K/AKT/mTOR pathway. However, FNDC4 has been reported to promote myogenic differentiation via activating the Wnt/β-catenin pathway and integrin αV/β5 has been identified as the receptor for FNDC5/Irisin in bone and fat (Kim et al, 2018; Li et al, 2020). These findings demonstrate the diversity of receptors for members of the FNDC family and the distinct regulatory pathways implicated in myogenesis. In addition, the receptor for FNDC1, α5β1 integrin, is not exclusively expressed in muscle, and it is involved in a variety of biological processes such as osteogenesis (Hamidouche et al, 2009), angiogenesis (Cascone et al, 2005), and tumor progression (Dudvarski Stanković et al, 2018). FNDC1, as a myokine, may be involved in these biological processes through the circulation, which is a fertile area to explore.

Myokines facilitate drug development and clinical applications due to their biological properties (Barrientos et al, 2014). For large molecular weight proteins, recombinant protein fragments that retain intact activity are often synthesized to facilitate drug development. For example, the recombinant SLIT3 fragment retains the activity of SLIT3 and is promoted as a potential drug for the treatment of muscle loss (Cho et al, 2021). Truncated Wnt7a retains intact biological activity in skeletal muscle and is able to

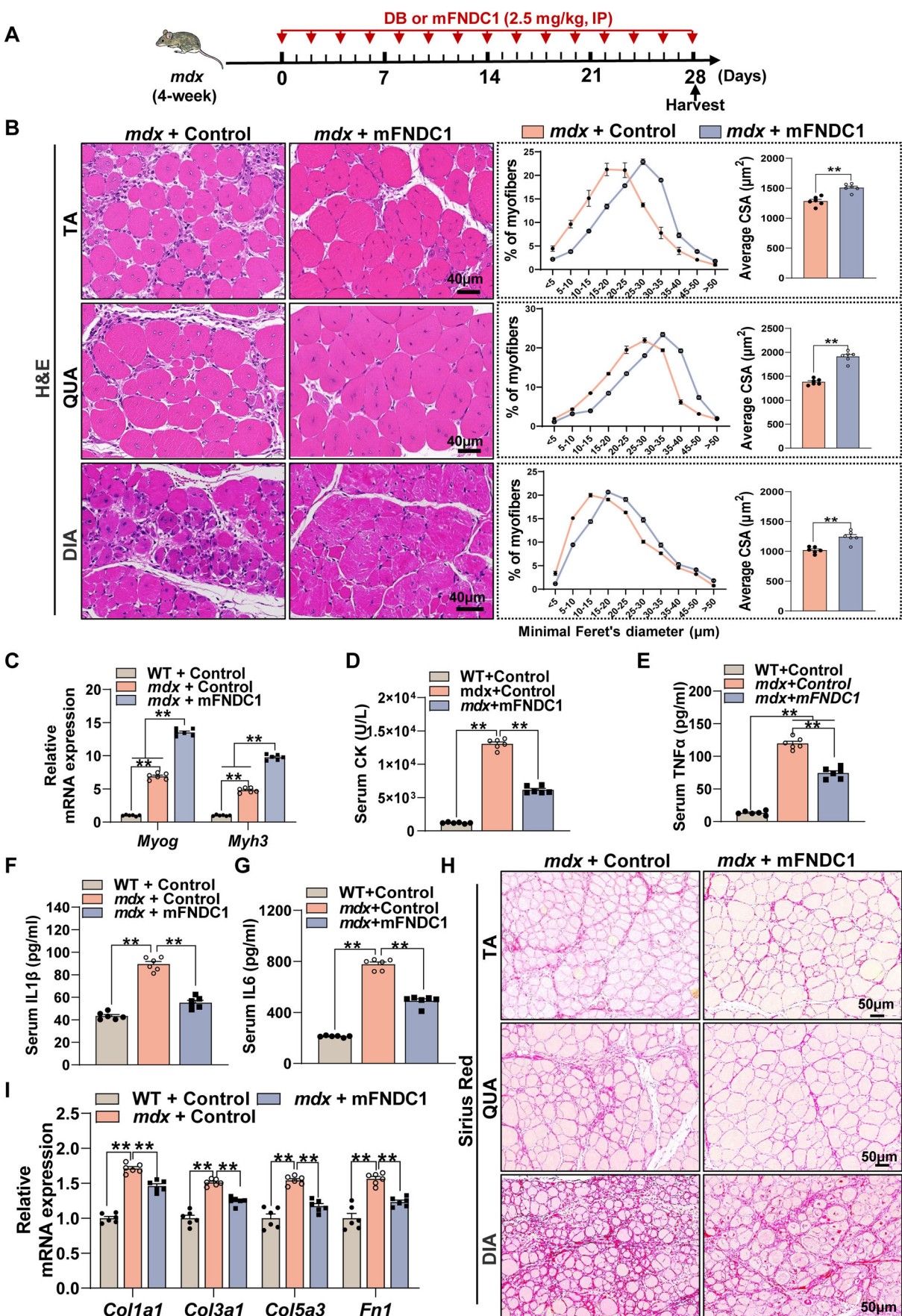

**Figure 8. FNDC1 ameliorates the pathological phenotype in *mdx* mice.**

(A) Experimental procedures. Four-week-old *mdx* mice were intraperitoneally injected with either control (dialysis buffer, DB) or mFNDC1 (2.5 mg/kg body weight) every 2 days for 4 weeks. Samples were collected 6 h after mFNDC1 treatment. (B) Representative images of H&E staining and quantify of TA (top), QUA (middle), and DIA (bottom) muscles from *mdx* mice treated with Control or mFNDC1 for 4 weeks ($n = 6$ mice). Scale bar = 40 µm. Two-tailed t-test, $P = 2.64 \times 10^{-4}$, $P = 1.44 \times 10^{-6}$, and $P = 6.28 \times 10^{-4}$ for TA, QUA, and DIA, respectively. (C) mRNA expression of myogenic marker genes *Myog* and *Myh3* ($n = 6$ mice). One-way ANOVA, WT+Control *vs.* *mdx*+Control, $P < 1 \times 10^{-15}$; WT+Control *vs. mdx* + mFNDC1, $P < 1 \times 10^{-15}$ and *mdx*+Control *vs. mdx* + mFNDC1, $P < 1 \times 10^{-15}$ for *Myog*; WT+Control *vs.* *mdx*+Control, $P = 3.84 \times 10^{-12}$; WT+Control *vs. mdx* + mFNDC1, $P < 1 \times 10^{-15}$ and *mdx*+Control *vs. mdx* + mFNDC1, $P < 1 \times 10^{-15}$ for *Myh3*. (D–G) Serum concentrations of CK, TNFα, IL1β, and IL6 in WT and *mdx* mice treated with mFNDC1 or control for 4 weeks ($n = 6$ mice). Serum was taken 4 weeks after mFNDC1 treatment from the indicated mouse models. (D) One-way ANOVA, WT+Control *vs.* *mdx*+Control, $P < 1 \times 10^{-15}$ and *mdx*+Control *vs. mdx* + mFNDC1, $P = 1.55 \times 10^{-12}$ for CK. (E) WT+Control *vs.* *mdx*+Control, $P < 1 \times 10^{-15}$, WT+Control *vs.mdx* + mFNDC1, $P = 7.44 \times 10^{-10}$, and *mdx*+Control *vs. mdx* + mFNDC1, $P = 4.86 \times 10^{-8}$ for TNFα. (F) WT+Control *vs.* *mdx*+Control, $P = 1.31 \times 10^{-10}$ and *mdx*+Control *vs. mdx* + mFNDC1, $P = 8.73 \times 10^{-9}$ for IL1β. (G) WT+Control *vs.* *mdx*+Control, $P < 1 \times 10^{-15}$ and *mdx*+Control *vs. mdx* + mFNDC1, $P = 8.87 \times 10^{-10}$ for IL6. (H) Representative images of Sirius Red staining of TA (top), QUA (middle), and DIA (bottom) muscles from *mdx* mice treated with Control or mFNDC1 for 4 weeks ($n = 6$ mice). Scale bar = 50 µm. (I) mRNA expression of myofiber marker genes (*Col1a1*, *Col3a1*, *Col5a3*, and *Fn1*) ($n = 6$ mice). One-way ANOVA, WT+Control *vs.* *mdx*+Control, $P = 5.37 \times 10^{-11}$ and *mdx*+Control *vs. mdx* + mFNDC1, $P = 4.68 \times 10^{-5}$ for *Col1a1*; WT+Control *vs.* *mdx*+Control, $P = 1.10 \times 10^{-8}$ and *mdx*+Control *vs. mdx* + mFNDC1, $P = 4.30 \times 10^{-5}$ for *Col3a1*; WT+Control *vs.* *mdx*+Control, $P = 5.74 \times 10^{-7}$ and *mdx*+Control *vs. mdx* + mFNDC1, $P = 5.80 \times 10^{-5}$ for *Col5a3*; WT+Control *vs.* *mdx*+Control, $P = 1.05 \times 10^{-6}$ and *mdx*+Control *vs. mdx* + mFNDC1, $P = 4.25 \times 10^{-4}$ for *Fn1*. Data are represented as mean ± SEM. **$p < 0.01$. Source data are available online for this figure.

induce skeletal muscle hypertrophy and resistance to muscular atrophy (von Maltzahn et al, 2013). The large molecular weight of FNDC1 (195 kDa) makes drug development difficult. Therefore, based on the results of truncation experiments, we synthesized mouse recombinant FNDC1 protein containing the FN3 domain and showed that it retains the intact biological activity of FNDC1. The essential role of FN3 domain for FNDC family members has been corroborated by previous studies which showed that both FNDC5 and FNDC4 function via the FN3 structural domain (Bosma et al, 2016). More importantly, our results support that mFNDC1 is sufficient to promote CTX-induced muscle regeneration and alleviate muscular atrophy in *mdx* mice, suggesting that mFNDC1 is a promising therapeutic drug for muscle diseases.

FNDC1 ameliorates the muscle pathological phenotype of *mdx* mice possibly via activation of the integrin α5β1 pathway. DMD is a chronic muscle disease accompanied by muscle damage and sustained regeneration (Mazala et al, 2020). Enhancement of sarcolemma integrity by integrins is an effective way to alleviate DMD (Han et al, 2009; Wang et al, 2008). It has been shown that moderate upregulation or activation of integrin α7β1 increases sarcolemma integrity and reduces mechanical stress-induced muscle fiber damage in *mdx* mice (Burkin et al, 2001; Rooney et al, 2006; Sarathy et al, 2017; Schwander et al, 2003). In line with α7β1, integrin α5β1 is required for long-term sarcolemma integrity (Taverna et al, 1998a). Deficiency of integrin α5 impairs sarcolemma integrity and results in muscular dystrophy (Boppart et al, 2006; Taverna et al, 1998b). Both α5β1 and α7β1 are present in fibers at sites where mechanical stress occurs, thereby maintaining the connection between muscle fibers and the matrix (Taverna et al, 1998a). We therefore deduced that FNDC1 binding to integrin α5β1 increases sarcolemma integrity and attenuates the pathological phenotype in *mdx* mice. Furthermore, dystrophin–glycoprotein complex, which connects the extracellular matrix to the actin cortex, plays a role in maintaining the mechanical structure of the cell membrane and in signal transduction (Eid Mutlak et al, 2020; Noguchi et al, 1995; Waite et al, 2012). Like the dystrophin–glycoprotein complex, integrin complexes mediate interactions between the ECM and the membrane cytoskeleton (Pang et al, 2023). Thus, FNDC1 may regulate cytoskeletal desmin and myofibril stabilization via integrin complex in systems lacking an intact dystrophin–glycoprotein

complex. Further studies are needed to understand how FNDC1 attenuates DMD and to evaluate its role in treating chronic muscle disease.

In hundreds of clinical trials testing for DMD, only glucocorticoids consistently demonstrated efficacy in the preservation of muscle function and ambulation (Morrison-Nozik et al, 2015). The effectiveness of mFNDC1 in the treatment of *mdx* mice provides an alternative option for this genetic muscular disease. Glucocorticoids are able to maintain muscle function and ambulation in DMD patients by inhibiting chronic and excessive inflammatory processes and slowing the progression of DMD (Dort et al, 2021). Similar results were observed in mFNDC1-treated *mdx* mice which exhibited reduced inflammation and skeletal muscle fibrosis, increased muscle strength, and improved muscle atrophy. In addition, previous study has shown that FNDC4 and FNDC1, but not FNDC5, are significantly increased in human inflammatory bowel disease (Wuensch et al, 2019), and FNDC4 has been described as an anti-inflammatory factor (Bosma et al, 2016). Our findings suggest that FNDC1 treatment reduces inflammation in *mdx* mice, suggesting that FNDC1 may play an anti-inflammatory role similar to FNDC4. These findings may open new avenues for the development of biologic agents for the treatment of muscle injury and dystrophy disorders.

One limitation of our study is the absence of evidence regarding the effects of FNDC1, apart from ameliorating skeletal muscle function, on the dysfunction of other key organs that are often affected in DMD. Further research is necessary to explore the systemic role of FNDC1 in these directions especially in heart function since DMD patients often die from dilated cardiomyopathy (Kamdar and Garry, 2016; Muntoni et al, 2003). FNDC1 may exert direct effect on cardiac muscle because integrin receptors for FNDC1 are also expressed in the myocardium (Parvatiyar et al, 2019). Moreover, the beneficial effect of FNDC1 on ameliorating systemic inflammation might indirectly benefit cardiomyopathy in DMD by reducing inflammatory damage and promoting a favorable cardiac environment as shown in other situations (Cozzoli et al, 2011). Also, the cross-talk between skeletal muscle and cardiac muscle upon FNDC1 treatment in *mdx* mice is another fertile area to explore. Thus, it is worthwhile to investigate whether FNDC1 treatment also alleviate other abnormalities such as cardiac dysfunction in *mdx* mice and DMD patients.

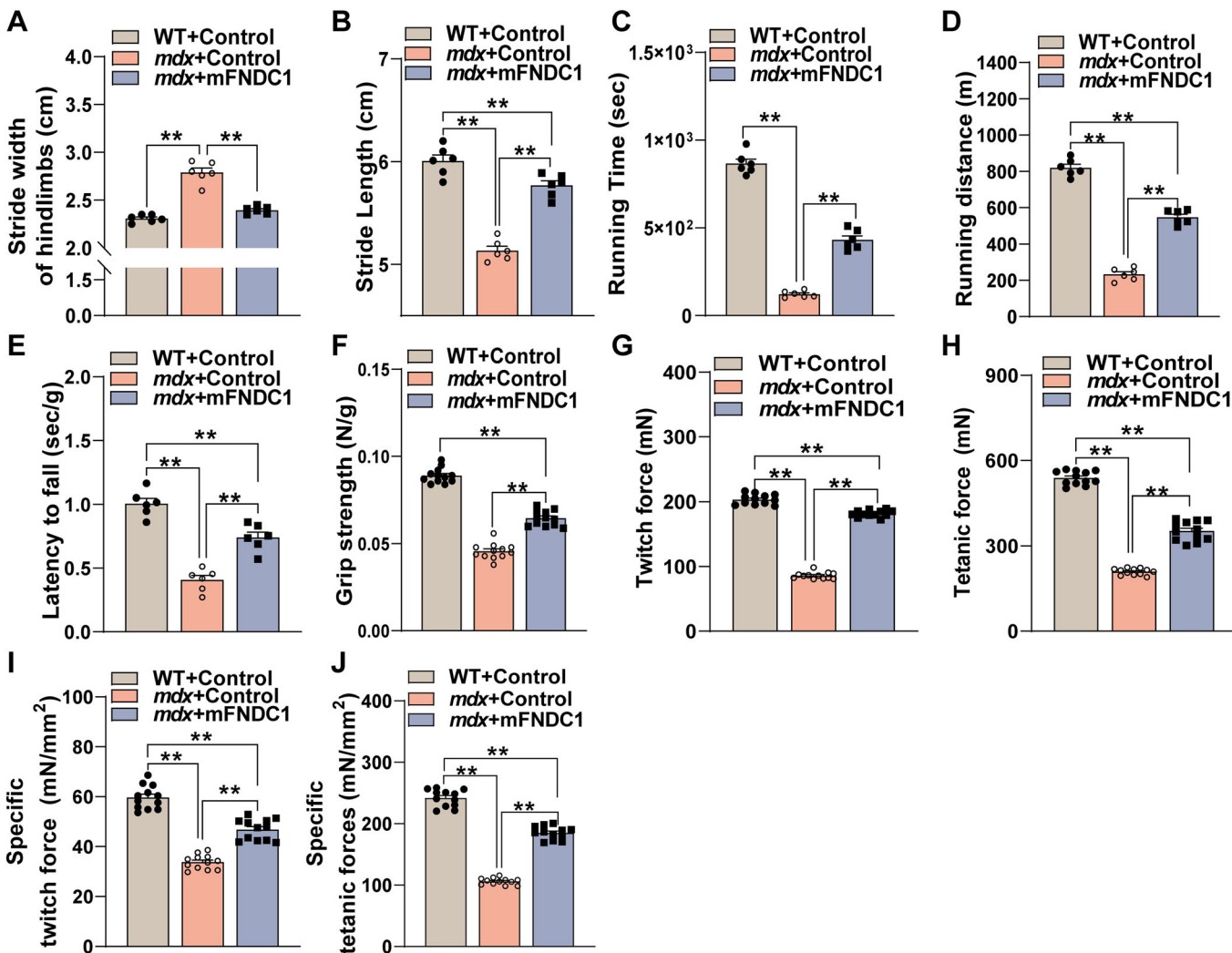

**Figure 9.   FNDC1 improves muscle performance in *mdx* mice.**

(A, B) Quantification of stride width of hindlimbs (A) and stride length (B) in WT (WT+Control), *mdx* (*mdx*+Control), and mFNDC1-treated *mdx* (*mdx* + mFNDC1) mice ($n = 6$ mice). (A) One-way ANOVA, WT+Control *vs. mdx*+Control, $P = 1.81 \times 10^{-8}$ and *mdx*+Control *vs. mdx* + mFNDC1, $P = 2.67 \times 10^{-7}$; (B) One-way ANOVA, WT+Control *vs. mdx*+Control, $P = 5.88 \times 10^{-9}$, WT+Control *vs. mdx* + mFNDC1, $P = 0.0091$ and *mdx*+Control *vs. mdx* + mFNDC1, $P = 4.17 \times 10^{-7}$. (C, D) Physical performance was evaluated in age-matched *mdx* mice by treadmill exhaustion test ($n = 6$ mice). Time to exhaustion and running distance were measured. (C) One-way ANOVA, WT+Control *vs. mdx*+Control, $P < 1 \times 10^{-15}$ and mdx+Control *vs. mdx* + mFNDC1, $P = 5.20 \times 10^{-8}$; (D) One-way ANOVA, WT+Control *vs. mdx*+Control, $P < 1 \times 10^{-15}$, WT+Control *vs. mdx* + mFNDC1, $P = 2.20 \times 10^{-8}$ and *mdx*+Control *vs. mdx* + mFNDC1, $P = 2.93 \times 10^{-9}$. (E) Fatigability was evaluated by an inverted-grid test, the latency to fall represents an average of two sessions of evaluation normalized to total body weight ($n = 6$ mice). One-way ANOVA, WT+Control *vs. mdx*+Control, $P = 6.78 \times 10^{-8}$, WT+Control *vs. mdx* + mFNDC1, $P = 8.04 \times 10^{-4}$ and *mdx*+Control *vs. mdx* + mFNDC1, $P = 8.26 \times 10^{-5}$. (F) Grip strength test in WT and *mdx* mice treated with Control or mFNDC1 for 4 weeks ($n = 12$ mice). One-way ANOVA, WT+Control *vs. mdx*+Control, $P = 6.20 \times 10^{-14}$, WT+Control *vs. mdx* + mFNDC1, $P = 7.70 \times 10^{-14}$ and *mdx*+Control *vs. mdx* + mFNDC1, $P = 1.12 \times 10^{-11}$. (G, H) Quantification of TA muscle twitch and tetanic force ($n = 12$ mice). Twitch force was produced by the TA muscle under 50 Hz stimulation. Tetanic force was produced by the TA muscle induced by 500 ms at 100 Hz stimulation. (G) One-way ANOVA, WT+Control *vs. mdx*+Control, $P = 6.20 \times 10^{-14}$, WT+Control *vs. mdx* + mFNDC1, $P = 1.29 \times 10^{-9}$ and *mdx*+Control *vs. mdx* + mFNDC1, $P = 6.20 \times 10^{-14}$. (H) One-way ANOVA, WT+Control *vs. mdx*+Control, $P = 6.20 \times 10^{-14}$, WT+Control *vs. mdx* + mFNDC1, $P = 6.20 \times 10^{-14}$ and *mdx*+Control *vs. mdx* + mFNDC1, $P = 6.80 \times 10^{-14}$. (I, J) Specific twitch force and tetanic force for TA muscle in WT and *mdx* mice treated with mFNDC1 or control ($n = 12$ mice). (I) One-way ANOVA, WT+Control *vs. mdx*+Control, $P = 6.30 \times 10^{-14}$, WT+Control *vs. mdx* + mFNDC1, $P = 1.34 \times 10^{-8}$ and *mdx*+Control *vs. mdx* + mFNDC1, $P = 1.01 \times 10^{-8}$. (J) One-way ANOVA, WT+Control *vs. mdx*+Control, $P = 6.20 \times 10^{-14}$, WT+Control *vs. mdx* + mFNDC1, $P = 9.4 \times 10^{-14}$ and *mdx*+Control *vs. mdx* + mFNDC1, $P = 6.20 \times 10^{-14}$. Data are represented as mean ± SEM. **$p < 0.01$. Source data are available online for this figure.

In conclusion, we identified FNDC1 as a novel myokine that promotes myogenesis. To facilitate drug development and explore the translation potential of FNDC1, we generated a truncated form of recombinant FNDC1, which possesses the activity of the full-length FNDC1 protein. Function tests showed its effectiveness in the amelioration of muscle repair after injury in adult mice and improvement of muscular dystrophy in *mdx* mice. These results support that FNDC1 is a potential therapeutic agent for the treatment of DMD or other skeletal muscle degenerative diseases. More studies are needed to assess the efficacy and safety of FNDC1 in advanced animals.

# Methods

## Reagents and tools table

| Reagent/Resource | Reference or Source | Identifier or Catalog Number |
|---|---|---|
| **Experimental models** | | |
| B10-Dmd-KO (*M. musculus*) | GemPharmatech | Strain NO. T003035 |
| C57BL/6J (*M. musculus*) | Xi'an Jiaotong University | N/A |
| C2C12 Scrambled (Mus musculus) | ATCC | RRID:CVCL_0063 |
| DH5α competent *E. coli* cells | NEB | N/A |
| BL21(DE3)RIL competent *E. coli* cells | Novagen | N/A |
| **Recombinant DNA** | | |
| pcDNA3.1-3XFLAG | This study | N/A |
| pcDNA3.1-6XHis | This study | N/A |
| pcDNA3.1-HA | This study | N/A |
| pcDNA3.1-3XFLAG-FNDC1 | This study | N/A |
| pcDNA3.1-3XFLAG-FNDC1-FL | This study | N/A |
| pcDNA3.1-3XFLAG-FNDC1-FN3 | This study | N/A |
| pcDNA3.1-3XFLAG-FNDC1-PHA | This study | N/A |
| pET28-6XHis-FNDC1 | This study | N/A |
| pcDNA3.1-3XFLAG-ITGB1 | Mailing Biologicals | N/A |
| pcDNA3.1-3XFLAG-ITGBA5 | Mailing Biologicals | N/A |
| pcDNA3.1-3XFLAG-ITGBAV | Mailing Biologicals | N/A |
| pLenti-3xFLAG-FNDC1 | This study | N/A |
| **Antibodies** | | |
| Mouse anti-FNDC1 | Atagenix Laboratories | Cat #ATA29400 |
| Rabbit anti-FNDC1 | Bioss | Cat #bs-8460R |
| Mouse anti-ITGB1 | Abcam | Cat #ab8991 |
| Mouse anti-ITGB1 | Santa Cruz | Cat #sc-374429 |
| Mouse anti-p-FAK (Tyr 397) | Santa Cruz | Cat #sc-81493 |
| Mouse anti-FAK | Santa Cruz | Cat #sc-271126 |
| Rabbit anti-eMYHC | Bioss | Cat #bs-10905R |
| Mouse anti-MYOG | Novus Biologicals | Cat #NB100-56510 |
| Mouse anti-MYOD | Novus Biologicals | Cat #NBP1-54153 |
| Rabbit anti-AKT | CS | Cat #4685 |
| Rabbit anti-p-AKT(Ser473) | CS | Cat #4060 |

| Reagent/Resource | Reference or Source | Identifier or Catalog Number |
|---|---|---|
| Rabbit anti-mTOR | CS | Cat #2983 |
| Rabbit anti-p-mTOR(S2448) | CS | Cat #5536 |
| Rabbit anti-α-TUBULIN | Proteintech | Cat #11224-1-AP |
| Mouse anti-MYHC | R&D Systems | Cat #MAB4470 |
| Goat anti-rabbit (HRP) | Abbkine | Cat #A21020 |
| Goat anti-mouse (HRP) | Abbkine | Cat #A21010 |
| Goat anti-rabbit (Alexa Fluor 546) | Invitrogen | Cat #A-11035 |
| Goat anti-mouse (Alexa Fluor 546) | Invitrogen | Cat #A-11003 |
| Goat anti-rabbit (Alexa Fluor 350) | Beyotime Biotechnology | Cat #A0412 |
| Goat anti-mouse (FITC) | Transgene | Cat #HS211-01 |
| **Oligonucleotides and other sequence-based reagents** | | |
| Mouse *Fndc1* gRNA1 | This study | 5′-GUGGCCGGAA GAUGAAUUAUT-3' |
| Mouse *Itgb1* gRNA1 | This study | 5′-GAGGCUCUCAA ACUAUAAAUT-3' |
| scrambled gRNA1 | This study | 5′-TTCTCCGAACG TGTCACGTAA-3' |

## qPCR primers for mouse

| Gene | Forward | Reverse |
|---|---|---|
| *Fndc1* | GGTGGAGTATT ACAACATTGCCT | AAGGAGTGTGT CTCCGCATTC |
| *Pltp* | CGCAAAGGGCC ACTTTTACTA | GCCCCCATCATA TAAGAACCAG |
| *Ctsc* | CAACTGCACCTA CCCTGATCT | TAAAATGCCC GGAATTGCCCA |
| *Gng2* | ACCGCCAGC ATAGCACAAG | AGTAGGCCAT CAAGTCAGCAG |
| *Rgs2* | GAGAAAATGAA GCGGACACTCT | GCAGCCAGCCC ATATTTACTG |
| *Myog* | GAGACATCCCCC TATTTCTACCA | GCTCAGTCCGC TCATAGCC |
| *Caveolin-3* | ACATTAAGGTGG TGCTGCGA | CCTCTTAAGGA ACCCTCCGC |
| *Myomaker* | GGGTCCAGGATA AAAGGCTCC | GCCAAGCATTG TGAAGGTCG |
| *Myod* | CCGTGTTTCGA CTCACCAGA | GTAGTAGGCGGT GTCGTAGC |
| *Myh3* | AAAAGGCCATC ACTGACGC | CAGCTCTCTGAT CCGTGTCTC |
| *Col1a1* | GCTCCTCTTA GGGGCCACT | CCACGTCTCA CCATTGGGG |
| *Col5a3* | AAATGCCACGG TCTCGAAGG | CCTCCTCAGTA GCGTCCATC |
| *Fn1* | ATGTGGACCCC TCCTGATAGT | GCCCAGTGAT TTCAGCAAAGG |

| Reagent/Resource | Reference or Source | Identifier or Catalog Number |
|---|---|---|
| *Col3a1* | CTGTAACATGGAAACTGGGGAAA | CCATAGCTGAACTGAAAACCACC |
| *Itga5* | CTTCTCCGTGGAGTTTTACCG | GCTGTCAAATTGAATGGTGGTG |
| *Itgav* | CCGTGGACTTCTTCGAGCC | CTGTTGAATCAAACTCAATGGGC |
| *Itga8* | CGAAGCCGAACTCTTTGTTATCA | GGCCTCAGTCCCTTGTTGT |
| *Itgb1* | ATGCCAAATCTTGCGGAGAAT | TTTGCTGCGATTGGTGACATT |
| *Fak* | CGAAAGGCTTCCCCAGTCTT | ACTCAGAAGGCAGCAGTGAC |
| **Chemicals, Enzymes and other reagents** | | |
| IPTG | Solarbio Life Sciences | Cat#367-93-1 |
| Ni-NTA agarose | Smart-Lifesciences Biotechnology | N/A |
| Cardiotoxin (CTX) | Sigma-Aldrich | Cat #11061-96-4 |
| Dulbecco's modified Eagle medium | HyClone | Cat #H30022.01 |
| Fetal bovine serum | ZETA LIFE | Cat #Z7186FBS-500 |
| Lipofectamine 3000 | Invitrogen | Cat #L3000015 |
| FLAG-M2 agarose beads | Sigma-Aldrich | Cat #A2220 |
| Protein A/G beads | Santa Cruz | Cat #sc-2003 |
| Horse serum | HyClone | Cat #SV30208.01 |
| Paraformaldehyde | Sigma-Aldrich | Cat #16005-250G-F |
| Mouse FNDC1 ELISA kit | Mlbio | Cat #YJ111201 |
| Mouse CK kit | Mlbio | Cat #ml092958 |
| Mouse LDH kit | Abcam | Cat #ab102526 |
| Mouse TNF-α kit | Mlbio | Cat #ml002095 |
| Mouse IL-6 kit | Mlbio | Cat #ml002293-1 |
| HisProbe-HRP conjugate | Thermo Scientific | Cat#15165 |
| Mouse IL-1β ELISA kit | Mlbio | Cat #ml063132-1 |
| RevertAid First Strand cDNA Synthesis | Thermo Scientific | Cat #K1621 |
| FreeStyle293 Expression Medium | Thermo Scientific | Cat #12338018 |
| DTSSP | Sigma-Aldrich | Cat #81069-02-5 |
| RIPA buffer | MilliporeSigma | Cat #20-188 |
| **Software** | | |
| Uniprot | https://www.uniprot.org/ | N/A |
| SPSS Statistics 20 | https://www.ibm.com/products/spss-statistic | N/A |
| Origin | OriginLab https://www.originlab.com/ | N/A |
| Fiji ImageJ.2 | https://imagej.net/software/fiji/ | N/A |
| ImageJ | https://imagej.nih.gov/ij/index.html | N/A |
| **Other** | | |

## Cloning, expression, and purification of mFNDC1

The recombinant mFNDC1 expression plasmid was constructed by cloning the FN3 structural domain of FNDC1 into a pET28(+) vector. Plasmids were transfected into *E. coli* BL21 (DE3) competent cells, and cultured at 37 °C. Protein expression was induced by adding 1 mM IPTG (Cat#367-93-1, Solarbio Life Sciences, Beijing, PRC) at OD600 = 0.6. Cells were cultured at 16 °C overnight. The harvested cells were lysed by lysis buffer and sonicated. The supernatant was incubated with Ni-NTA agarose at 4 °C for 1 h. The Ni-NTA agarose (Smart-Lifesciences Biotechnology, Changzou, PRC) was subsequently washed and then eluted with 500 mM imidazole. The eluted protein was further purified by gel filtration chromatography using the Superdex 200 10/300 GL column (GE Healthcare) at 4 °C in an ÄKTA FPLC system (GE Healthcare). SDS-PAGE, reverse-phase high-performance liquid chromatography, and Mass spectrometry were used to evaluate protein purity and integrity. Following purification, the protein was aliquoted and stored at −80 °C for up to three freeze-thaw cycles.

## Animals

The C57BL/6J mice were purchased from the animal center of Xi'an Jiaotong University (Xi'an, PRC). The classic *mdx* mice on a C57BL/10SnJ genetic background (Strain NO. T003035) were purchased from GemPharmatech (Nanjing, PRC). All mouse experiments were approved by the Animal Ethics and Welfare Committee of Northwest A&F University (Yangling, PRC) [Approval ID: NWAFU-314020140]. At the end of treatment, mice were sacrificed and tissues were immediately snap-frozen or formalin-fixed for subsequent histological evaluation.

AAV serotype 9 vectors encoding a control scrambled shRNA (scrambled; 5′-TTCTCCGAACGTGTCACGTAA-3′), a short hairpin targeting FNDC1 (sh*Fndc1*; 5′-GUGGCCGGAAGAUGAAUUAUU-3′), or a short hairpin targeting *Itgb1* (sh*Itgb1*; GAGGCUCUCAAACUAUAAAUU-3′) under the control of a U6 promoter and expressing Zsgreen (driven by a CMV promoter) were obtained from Scilia (Scilia Life Science, Inc, Shanghai, PRC). A single dose of $1.1 \times 10^{12}$ vg/mice in 40 μl of AAV9 expressing sh*Fndc1* or sh*Itgb1* was injected locally into the right TA muscle of mice, and the same dose of AAV9 expressing shRNA control was injected into the left TA muscle as control.

To assess the effect of mFNDC1 on CTX group, muscle injury was induced by a single dose of intramuscular injection of 50 μL

CTX (10 μM) into TA muscle of 6-week-old C57BL/6J mice. CTX-treated mice were given dialysis buffer or mFNDC1 (2.5 mg/kg body weight) every 2 days via intramuscular injection starting with the treatment of CTX. Serum and TA muscle samples were collected at days 0 (uninjured), 3, 5, and 14 after injury, respectively. Samples were collected 24 h after mFNDC1 injection on days 3 and 5, while 6 h after last mFNDC1 treatment at day 14. Serum FNDC1 levels were measured using an ELISA kit (YJ111201, mlbio, Shanghai, PRC).

To assess the impact of mFNDC1 on *mdx* mice, 4-week-old *mdx* mice were intraperitoneally injected with either dialysis buffer (DB) or mFNDC1 (2.5 mg/kg body weight) every 2 days for 4 weeks. Serum was collected 6 h after the last mFNDC1 treatment in mice and followed by exercise performance assessments.

To ensure unbiased and reliable results, all experimental mice followed the principles of both rationalization and blinding. The individuals responsible for conducting the treatments and assessments were blinded to the treatment groups. The investigators for sample collection were also blinded to the experimental conditions of the mice examined. No mice were excluded from the analysis.

## Treadmill exhaustion test

Age-matched WT (WT + Control), untreated *mdx* (*mdx* + Control), and mFNDC1-treated *mdx* (*mdx* + mFNDC1) mice were exercised on a treadmill. Three days prior to the exhaustion test, all animals were acclimated by running on a treadmill at 10 m/min for 5 min at a 10% incline. Each run-to-exhaustion trial produced a maximum exercise capacity estimate based on two parameters: the duration of the run (sec) and the distance of the run (m). The two sessions' values were averaged to calculate exercise capacity.

## Inverted-grid test

To assess limb endurance, the inverted-grid hanging test was used. Age-matched (8 weeks) WT (WT + Control), untreated *mdx* (*mdx* + Control), and mFNDC1-treated *mdx* (*mdx* + mFNDC1) mice were suspended 60 cm above a padded surface on a mesh grid. After that, the grid was turned upside down, and the mouse was suspended upside down. The time it took to fall off the grid was recorded, and the maximum trial time was 6 min. Each mouse was tested twice over the course of one week, with a 4-day break in between. The latency to fall in both sessions was averaged and normalized to total body weight.

## Grip tests

To assess mouse forelimb strength, we used a Grip Strength Meter (Columbus Instruments, USA) and allowed the mice to grip the horizontal bar attached to the front of the meter. Mice were pulled horizontally by the tail until the bar was released. Repetitions of the procedure were performed and the peak force was recorded for each mouse.

## In situ muscle force measurements

In situ force measurements of TA muscles were performed as previously reported (Son et al, 2019) using a BL-420F Biofunctional Laboratory System (Tymon Software, Chengdu, PRC). The TA muscle was then surgically exposed and the distal tendon of the muscle was tied with 3-0

sutures. After cutting the tendon, the TA muscle was connected to the lever arm of the dynamometer with a suture. Electrical stimulations were elicited with 50 HZ pulses by two electrodes placed on either side of the muscle to elicit single-twitch or tetanic contractions. At the end of the measurement, the optimal length and wet weight of the TA muscle were measured to calculate the physiological cross-sectional area (muscle mass/optimal length * muscle density 1.06 g/cm$^3$), which was then used to obtain specific isometric twitches and limb strength.

## Cell culture

C2C12 cells were obtained from China Infrastructure of Cell Line Resource and cultured in high-glucose Dulbecco's modified Eagle medium (H30022.01, HyClone, CT, USA) supplemented with 10% fetal bovine serum (FBS) (Z7186FBS-500, ZETA LIFE, CA, USA), and 1% penicillin/streptomycin. The cell density was assessed by a hemocytometer. 50,000 cells per well (12-well culture plate) were seeded for experiments. Myoblasts were cultured to 80% confluence (~24 h), after which the growth medium was replaced with differentiation medium (high sugar DMEM containing 2% horse serum and 1% penicillin/streptomycin). The medium was changed and collected daily until the myotubes were fully mature. The collected medium was centrifuged to remove cell debris and then concentrated 100-fold by ultrafiltration tubes (Millipore UFC801024). Finally, medium FNDC1 was measured by ELISA (Cat#YJ111201, mlbio, Shanghai, PRC) and Western blot.

## siRNA and plasmids transfection

For seeding and transfection, C2C12 cells were seeded at a density of 50,000 cells per well in a 12-well plate. After 12 h, siRNA and plasmids transfection were performed using jetPRIME transfection reagent (Polyplus-transfection, 101000015). Twelve hours later, the medium was replaced with differentiation medium containing DMEM supplemented with 2% horse serum and 1% penicillin/streptomycin. The differentiation medium was replaced every 2 days. The medium and cells were collected at the indicated time points for subsequent experiments. The si-control, si-*Fndc1*, si-*Itgb1*, and si-*Fak* were synthesized from Genepharma (Shanghai, PRC). The siRNA sequences for *Fndc1*, *Itgb1*, and *Fak* were as follows: *Fndc1*-siRNA1, 5′-GUGGCCGGAAGAUGAAUUATT-3′; *Itgb1*-siRNA1, 5′-GAGGCUCUCAAACUAUAAATT-3′. *Itgb1D*-siRNA, 5′-GAAAAUCCGAUUUACAAGAGU-3′, *Itgb1A*-siRNA, 5′-CUGAUAUGGAAGCUUUUAAUG-3′, *Fak*-siRNA, 5′-GGAAAUAUGAGUUGAGAAU-3′. The plasmids encoding FNDC1-His, FNDC1-FN3 (missing residues 519–1092 of the PHA03247 domain) and FNDC1-PHA (missing residues 38–332 and 1480–1571 of the FN3 domain), ITGB1A, ITGB1D, ITGAV were synthesized by Mailing Biologicals (Shanghai, PRC).

## Isolation of primary myoblasts

Primary myoblasts were isolated as described previously (Rozo et al, 2016). Cells were grown on collagen-coated dishes in a humidified 5% CO$_2$ incubator at 37 °C in growth media (Ham's F10 supplemented with 20% FBS and 5 ng/ml basic fibroblast growth factor). Myogenic differentiation was induced in differentiation medium (high sugar DMEM supplemented with 2% horse serum and 1% penicillin/streptomycin).

## mRNA sequencing (RNA-seq)

Total RNA was extracted from *Fndc1* knockdown and control C2C12 cells at 4 days of differentiation using TRIzol reagent. The four RNA-seq libraries were sequenced on the Illumina NextSeq 500 platform at UNITED KAWA to produce over 60 million, 100 nucleotide paired-end reads per sample. The reads were then trimmed for sequencing adapters and aligned to the reference mouse genome version mm10 (GRCm38) using TopHat version 2.0.10. Gene quantification was performed on the mapped sequences using the htseq-count software version 0.6.1. Differential expression analysis was performed using the DESeq2 package. Log transformation was used to normalize raw read counts and to calculate normalized expression counts. (Data can be accessed in GEO database, GSE217125). DEGs were submitted to the GO and Kyoto Encyclopaedia of Genes and Genomes (KEGG) websites for enrichment analysis.

## Mass spectrometry analysis

C2C12 cells were seeded in 90 mm culture plates and cultured in the DMEM. At 80% confluence, cells were switched to differentiation medium (DMEM with 2% horse serum). On day 3 of differentiation, the medium was switched to FreeStyle293 Expression Medium. After 4 h of incubation, cells were chilled on ice for 10 min and then treated with 100 nM His-tag mFNDC1 for 20 min. The cells were then incubated with 1.5 mM DTSSP for 30 min on ice to cross-link after washing with 15 ml cold PBS twice. The addition of a final concentration of 20 mM Tris-pH 7.5 quenched the cross-linking. The cells were then harvested and homogenized in 1 ml RIPA buffer containing protease-inhibitor cocktail and phosphatase-inhibitor cocktail. His-tag protein-purified magnetic beads were used to enrich FNDC1-treated C2C12 cell lysates. Briefly, 1 ml of whole cell extract was added to the pre-treated magnetic beads and incubated at 4 °C for 90 min with rotation. After 3 repeated washes in NP40-free wash buffer, 0.8 ml of elution buffer is added. To the collected elution buffer, add protein loading buffer and denature at 95 °C. The samples were up-sampled to 4–12% gradient SDS-PAGE for separation, followed by Caulmers Brilliant Blue staining. The gel was used for mass spectrometry analysis.

## Histological analysis

The TA muscle of mice was fixed with 4% paraformaldehyde for more than 72 h and then subjected to dehydration embedding. Paraffin sections of muscle were obtained at a thickness of 2–4 μm for hematoxylin-eosin staining, and whole-slide digital images were collected with an Pannoramic DESK Scanner (P-MIDI, P250, 3D HISTECH, Hungary).

## RNA and protein analyses

RT-PCR, Western blot, immunoblotting, and immunofluorescence analyses were performed as described previously (Zhao et al, 2021; Li et al, 2021; Liu et al, 2012). Primers and antibodies used are listed in Reagents Table. Total RNA extraction was performed using TRizol (Takara) according to the manufacturer's protocol. cDNA was synthesized by SmArt RT Master Premix (5×) Kits (DY10502, DEEYEE, Shanghai, China) according to kit's instructions.

## Immunoprecipitation

For immunoprecipitation analysis, the TA muscle and cultured cells were homogenized with IP lysis buffer (containing 1 M pH 7.4 Tris-HCl 25 ml, NP40 25 ml, NaCl 4.383 g, EDTA 0.146 g, glycerin 50 ml and protease inhibitor cocktail), and the total protein was incubated with 5 μg of the Rabbit monoclonal antibody to His, Flag, or non-specific Rabbit IgG for 2 h at room temperature and then immunoprecipitated with protein A/G magnetic beads (B23201, Bimaker, Shanghai, PRC) at 4 °C overnight. After washing three times with TBS (5 min per time), the protein-bound beads were finally resuspended in 20 μl 1 × SDS-PAGE loading buffer. The samples were boiled at 95 °C for 10 min, and the supernatant was loaded on the gel for immunoblotting. For Co-IP analysis, C2C12 cells were transfected with pcDNA3.1-His-FNDC1, pcDNA3.1-His empty vector, pcDNA3.1-FLAG-ITGB1, or pcDNA3.1-FLAG empty vector for 12 h, respectively and then the differentiation medium was replaced. 48 h after differentiation, cells were digested with trypsin and washed once with PBS, then lysed with hypotonic lysis buffer and incubated on ice for 30 min. Total proteins were immunoprecipitated with His or flag magnetic beads at 4 °C overnight. After washing with TPBS 3 times (5 min each), the protein-bound beads were finally suspended in 20 μl of 1× SDS-PAGE up sampling buffer. The samples were boiled at 95 °C for 10 min and the supernatant was loaded onto a gel for immunoblotting.

## Pull-down assays

Pull-down assays were performed as described previously (Kim et al, 2018). 100 nM His-tagged FNDC1 was incubated with 5 nM of the indicated Flag-tagged integrins in a final volume of 600 ml for 5 min at room temperature under rotation. After rotation, 60 ml Ni-NTA agarose was applied to immunoprecipitated integrins (Smart-Lifesciences Biotechnology, Changzou, PRC). Precipitated integrins were detected by immunoblot analysis against His-tag.

## Biochemical analyses

Serum of CTX-induced mice was collected at days 0, 3, 5, and 14 after injury, respectively. Serum of *mdx* mice was collected at day 28 after FNDC1 treatment. FNDC1 protein from cell culture medium or serum of mice was measured by ELISA kit (catalog YJ111201, mlbio, Shanghai, PRC) according to manufacturer's instructions. Medium LDH was measured by LDH Assay kit (ab102526, Abcam) according to manufacturer's instructions. Serum levels of CK, TNFα, IL-1β, and IL-6 were measured by commercial ELISA kits (ml092958, ml002095, ml063132-1, and ml002293-1, mlbio, Shanghai, PRC).

## Pharmacokinetics of FNDC1 in mice

Serum concentrations of mFNDC1 were determined by anti-His tag ELISA. Recombinant mFNDC1 protein was injected intraperitoneally at a dose of 2.5 mg/kg body weight. Serum levels of FNDC1 were measured using HisProbe-HRP conjugate according to the manufacturer's instructions (Cat#15165, Thermo, USA).

## LC-MS/MS

LC-MS/MS were performed as previously described (Jiang et al, 2021). Coomassie staining was used to resolve samples run on a 4–12% SDS-PAGE gel. Extracted gel bands were placed in 1.5 ml Eppendorf tubes and then cut into $1 \times 1$ mm squares. The mass spectrometry sample preparation and purification procedure were followed by the cut gels. Experiments were carried out using a mass spectrometer in conjunction with liquid chromatography. The RAW data files were processed for analysis using Byonic v3.2.0 (Protein Metrics, San Carlos, CA) to identify peptides and infer proteins using the Uniprot Mus musculus database. Mass spectrometry results were listed in Appendix Tables S1 and S3.

## Statistical analysis

All experiments were at least performed in three independent experiments. Data are presented as mean ± standard error of the mean and were analyzed by two-tailed Student's t tests for comparisons between two groups or one-way analysis of variance (ANOVA) with Duncan post hoc test for multiple comparisons. Statistical significance was defined as $*p < 0.05$, $**p < 0.01$, and $***p < 0.001$. All data were analyzed using SPSS Statistics 20 (SPSS, Chicago, IL, USA).

# Data availability

Raw and processed data for RNA sequencing is available at the NCBI GEO website under accession number GSE217125. Other data and materials are provided within the manuscript and supplementary materials, or available upon reasonable request.

The source data of this paper are collected in the following database record: biostudies:S-SCDT-10_1038-S44318-024-00285-0.

# Peer review information

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

## Acknowledgements

This work was supported by National Key Research and Development Program of China (2021YFF1000602), STI2030 Major Projects (2023ZD0404702), the Program for Shaanxi Science and Technology (2023-CX-TD-57), and Shaanxi Livestock and Poultry Breeding Common Technology Research and Development Platform (2023GXJS-02-01). The authors would like to thank Life Science Research Core Services of Northwest A&F University for LC-MS/MS analysis.

## Author contributions

**Rui Xin Zhang**: Conceptualization; Data curation; Formal analysis; Investigation; Methodology; Writing—original draft; Writing—review and editing. **Yuan Yuan Zhai**: Conceptualization; Data curation; Validation; Writing—original draft. **Rong Rong Ding**: Writing—review and editing. **Jia He Huang**: Resources; Writing—review and editing. **Xiao Chen Shi**: Visualization; Methodology. **Huan Liu**: Investigation; Methodology. **Xiao Peng Liu**: Resources; Methodology. **Jian Feng Zhang**: Resources; Supervision; Validation; Investigation; Visualization; Methodology. **Jun Feng Lu**: Validation; Investigation; Visualization. **Zhe Zhang**: Software; Formal analysis; Investigation. **Xiang Kai Leng**: Resources; Software; Validation; Methodology. **De Fu Li**: Resources; Software. **Jun Ying Xiao**: Software; Formal analysis; Investigation; Visualization. **Bo Xia**: Supervision; Writing—original draft; Writing—review and editing. **Jiang Wei Wu**: Conceptualization; Resources; Data curation; Formal analysis; Supervision; Funding acquisition; Validation; Investigation; Visualization; Methodology; Writing—original draft; Project administration; Writing—review and editing.

Source data underlying figure panels in this paper may have individual authorship assigned. Where available, figure panel/source data authorship is listed in the following database record: biostudies:S-SCDT-10_1038-S44318-024-00285-0.

## Disclosure and competing interests statement

The authors declare no competing interests.

