## [Peer Review File · The EMBO Journal]

FNDC1 is a myokine that promotes myogenesis and muscle regeneration

Rui Xin Zhang, Yuan Yuan Zhai, Rong Rong Ding, Jia He Huang, Xiao Chen Shi, Huan Liu, Xiao Peng Liu, Jian Feng Zhang, Jun Feng Lu, Zhe Zhang, Xiang Kai Leng, De Fu Li, Jun Ying Xiao, Bo Xia, and Jiang Wei Wu

Corresponding author: Jiang Wu (wujiangwei@nwafu.edu.cn)

Review Timeline:

Submission Date:	23rd Mar 24
Editorial Decision:	12th May 24
Revision Received:	11th Aug 24
Editorial Decision:	29th Sep 24
Revision Received:	4th Oct 24
Accepted:	17th Oct 24

Editors: Kelly Anderson / Ieva Gailite

Transaction Report:

Dear Prof. Wu,

Thank you for submitting your manuscript for consideration by the EMBO Journal. It has now been seen by three referees whose comments are shown below.

Given the referees' recommendations, I would like to invite you to submit a revised version of the manuscript, addressing the comments of all three reviewers. I should add that it is EMBO Journal policy to allow only a single round of revision, and acceptance of your manuscript will therefore depend on the completeness of your responses in this revised version.

Thank you for the opportunity to consider your work for publication. I look forward to your revision.

Yours sincerely,

Kelly M Anderson, PhD
Editor, The EMBO Journal
k.anderson@embojournal.org

We realize that it is difficult to revise to a specific deadline. In the interest of protecting the conceptual advance provided by the

work, we recommend a revision within 3 months (10th Aug 2024). Please discuss the revision progress ahead of this time with the editor if you require more time to complete the revisions.

Referee #1:

The authors identified FNDC1 as a novel myokine promoting myoblasts differentiation and muscle regeneration upon acute injury and in dystrophic mice. They demonstrate that FNDC1 interacts with integrin $\alpha 5 \beta 1$ and activates FAK/PI3K/AKT/mTOR pathway. Although these findings are interesting and relevant for the muscle field, several main concerns should be addressed.

Major points:

- 1) There is a general issue with the immunofluorescence staining that are not convincing. Indeed, the signals appear oversaturated in most of the pictures, casting doubt on their specificity. As an example, the signal for Pax7 is not convincing as it should colocalize with the nuclear staining and a membrane staining (e.g. laminin) should be added to undoubtedly identify the satellite cells based on their sublaminar localization.
- 2) Apart from two experiments, all in vitro experiments have been performed with C2C12 cell line, but the limitations associated with its usage are now firmly established within the field. Hence, the authors should confirm the key experiments in primary myoblasts (e.g. treatment with recombinant FNDC1 protein, pulldown assays).
- 3) The authors claim that FNDC1 promotes differentiation but not proliferation of myoblasts. However, they show only fusion index and number of nuclei while they should also calculate the differentiation index by performing staining for MF20. On the same line, it would be interesting to evaluate whether the expression of FNDC1 is differently modulated in proliferating and differentiating myoblasts by performing immunofluorescence staining for FNDC1 combined with Pax7, MyoD or MyoG on single myofibers and/or on muscle sections after CTX or in mdx mice.
- 4) The authors should compare the myogenic properties of FNDC1 (at least in vitro) with other proteins of the family. Indeed, they mentioned in the discussion that FNDC4 has been reported to promote myogenic differentiation (line 414).

Minor points:

- 1) The authors should include more information in the M&M about the cell assays (e.g. number of plated cells, number of days in growth medium before switching to DM) and check the numbering of the figures as they are not sequential in the text.
- 2) In fig. 1B, the results of the Q-PCR should be confirmed by WB as done for C2C12.
- 3) In fig 2E, the authors should include a more representative image of the quantified difference included in fig. S1A.
- 4) Fig. 2A-B or 2K-L should be included in the supplementary as including both in the main figure is redundant.
- 5) In fig 3A, why did the authors decide to analyze the level of FNDC1 in medium of C2C12 cells at day 3 and not at day 6 when they observed the highest level of mRNA (fig. 1E)? To exclude that the presence of FNDC1 might be due to a passive release following cell necrosis, it would be useful to also analyze the level of a necrotic marker (e.g. LDH).
- 6) Does the difference between CTR and FNDC1-FN3 is significant in fig 3G-H? If yes, it should be indicated on the graph.
- 7) The authors should explain how they determined the doses to be used in fig. 3N-O and fig 6.
- 8) The timepoint indicated in line 177 is not the same in the corresponding figure legend (fig. S3A-B).
- 9) In line 219, the authors mentioned fig. 3E-J which appear not to refer to the text. In addition, the authors mentioned the expression of $\alpha 5 \beta 1$ integrin but it is important to underline that $\alpha 7 \beta 1$ integrin is also expressed as it is a marker of MuSCs.
- 10) Fig. 5E: the authors do not show the validation of FAK knockdown in C2C12 cells.
- 11) The authors should comment in the discussion that $\alpha 5 \beta 1$ integrin is not expressed only in muscle and that it is involved in various biological processes such as osteogenesis, angiogenesis, and tumor progression. In addition, the authors claim that they provide evidence that mFNDC1 treatment significantly improves sarcomere integrity (line 468), but their data did not provide

such evidence. Hence, this sentence should be removed or modified.

Referee #2:

This work is an excellent and timely contribution, addressing novel mechanisms of muscle regeneration through diverse in vitro and in vivo methodologies. Not only does the paper reveal a protein with beneficial effects on injured muscles, but it also elucidates the underlying mechanism and suggests the utilization of a truncated active version of FNDC1 to enhance muscle regeneration in cases of injury and dystrophies. As such, its anticipated impact on the field is substantial. My main critique is about the lack of detailed methodology descriptions, which hinder a thorough understanding of how the experiments were done.

Overall, the conclusions drawn in the manuscript are justified. However, in some instances, there is a lack of direct evidence to support the authors' conclusions, such as the absence of direct evidence that FNDC1 is a secreted protein. Additionally, the manuscript suffers significantly from a lack of detail and description of methodologies and experiments, which reduces clarity, may lead to confusion for the reader, and makes it difficult for this reviewer to assess major parts of the data.

Major comments:

1. Please provide accession numbers or links to the four independent microarray datasets on GEO.
2. Fig 1D: to confirm myogenesis, please provide additional specific differentiation markers, such as MYOD and myogenin.
3. Fig 1F: "In CTX-treated mice....during the initial phase of muscle regeneration (days 3-5) and downregulated on day 14 when myoblast fusion was decreased", please support this statement and presented findings with images of the regenerating tissue, so the reader can appreciate the regeneration process and cell fusion.
4. Fig 1G: the provided images do not sufficiently depict the extent of the injury and regeneration.
5. Fig 1I: it is unclear whether the observed staining pattern of FNDC1 in mdx mice muscles indicates a problem with staining specificity or represents its spatial cellular distribution. If the staining is specific, please comment on the altered distribution compared to CTX-treated mice. The observed pattern suggests that FNDC1 aggregates.
6. Fig 2A: Based on the legend, C2C12 cells were transfected. I could not find anywhere in the manuscript how these cells were transfected. Transfecting C2C12 myoblasts with a transfection reagent is a serious challenge and the only way to transfect C2C12 myotubes is with an adenovirus. Did the authors use short hairpin RNAs? shRNA? Viruses or transfection reagents? Please describe the methodology in the text in detail.
7. Figs 2F and 3F-H and other figures involving C2C12 transfection: see comment #6, it is unclear how C2C12 cells were transfected with a plasmid.
8. Fig 3A-B: it is inaccurate to draw conclusions from the presented data due to the absence of a loading control. In addition, the lack of a protocol describing how the medium was prepared raises concerns about the origin of the presented protein, perhaps it was released during cell necrosis? or apoptosis? Or it is part of cell debris? The presented data does not support the assertion that this protein is secreted to the medium by vital cells.
9. Fig 3- again, I would be cautious in interpreting the data to conclude that FNDC1 is a secreted protein, as the authors have not provided direct evidence supporting this claim. Please note that this protein is found in the blood under conditions where skeletal muscle integrity is compromised, therefore the presence of muscle proteins in the serum is only expected.
10. Fig 3I: please provide additional more specific markers for cell differentiation and fusion.
11. Fig S4A- it is unclear how the experiment was done. Fndc1 knockdown in C2C12 myotubes is crucial, especially when cells are subsequently analyzed by RNAseq. The use of adenoviruses is necessary to achieve such efficiency. However, without detailed methodology provided, this reader is unable to evaluate the experiment and its results effectively.
12. Fig 4A-B: unfortunately, I am unable to review the data because there is no detailed description in the text or legend how the experiments were done.
13. Fig S4C-D: please provide densitometric measurements of presented blots, i.e. ratio of phosphorylated/unmodified protein.
14. Fig S6C- changes in MYHC levels may not necessarily indicate changes in cell differentiation. Therefore, throughout the paper, whenever the authors wish to show an effect on myogenesis, it is essential that they present specific myogenic markers in addition to MYHC.
16. p. 11: RNA-seq analysis and PCR validation revealed that only the $\alpha 5$ and αv , ..., were detectable in skeletal muscle (Fig. S6E-F)", please explain how this shows "which integrin α subunit pairs with the integrin $\beta 1$ for FNDC1 activation".
17. Fig 6A: "CTX-treated mice were given mFNDC1", how? Ip injection? Iv injection? In water?
18. Fig 6D-E and Fig 7A, p.13: "By day 5 post-injury.... inflammatory cells were almost cleared in mFNDC1-treated mice", unfortunately I cannot appreciate the author's claim from the presented data. Please quantitate the infiltrating cells as in Fig S12G.
19. Fig 8: I appreciate the beautiful data on mdx mice. Could the authors provide a mechanistic explanation for the beneficial effects of FNDC1 on dystrophic muscles? Given the ongoing absence of dystrophin in these muscles, it remains unclear how muscle performance can be enhanced under such conditions. How would the cytoskeletal desmin and the bound myofibrils be stabilized in a system lacking intact dystrophin-glycoprotein complex?

Minor comment:

Fig 5C-D: please specify in the figure which phosphorylated residue in AKT the antibody detects.

Referee #3:

This study investigates the role of FNDC1 as a novel myogenic regulator and potential therapeutic agent for muscle regeneration and dystrophy. The authors conducted a series of experiments using in vitro and in vivo models to elucidate the mechanisms underlying FNDC1-mediated myogenesis and its therapeutic potential.

1. Please include a table listing the abbreviations, some of which lack explanations.
2. If applicable, it would be beneficial to include one gene whose role in myogenesis has been documented to compare with the other genes lacking documentation in Figures 1B and C.
3. In Figure 1F, at day 0, both *Fndc1* and *Myog* exhibit similar relative mRNA expression levels. However, during days 3 and 5, both genes show an increase, yet after 14 days, *Myog* expression falls below that of *Fndc1*, which was higher during days 3 and 5.
5. Can you provide an explanation for this trend?
4. In Figure 1I, it appears that the scale bars for WT and *mdx* are not consistent and need to be verified. Additionally, WGA staining is incomplete in some parts of the fibers, and the image quality is inferior to that of the WT. It requires clarification.
5. Authors concluded in line 124 that "immunofluorescence analysis showed significantly higher FNDC1 in the muscle fibers of *mdx* mice compared with WT mice." However, the images do not sufficiently support this statement. To strengthen this conclusion, higher-quality images should be provided.
6. The cells used as controls in Figures 2C and H are the same, yet the percentage of nucleus differs between these two charts. Can you explain this discrepancy?
7. The sentence in line 205, "These results suggest that ITGB1 is the receptor for FNDC1," may be overstated. It would be more appropriate to state: "These results suggest that ITGB1 could be a receptor for FNDC1."
8. In Figure 6C, please provide an image with a larger tissue area.
9. In Figure S9A, the staining of PAX7 and MYOD is not sufficiently clear to accurately determine the percentage of PAX7 and MYOD, making comparisons between different groups unreliable.
10. In Figure 6E, please set the y-axis origin to zero (it currently starts at 1200, thus overrepresenting the difference in fiber size).
11. In Figure 6J, please provide an image with a larger area or lower magnification for clarity.
12. In Figure 6A, the arrows are green, contrary to the mention of blue in line 978. Please verify this discrepancy.
13. In Figure 6K, please explain how many microscopic fields did the authors count for evaluating the data, which the authors concluded promote the formation of multinucleates in line 295.
14. For Figure 7B, a larger tissue area needs to be provided.
15. In Figure S12J, regarding the immunofluorescence staining of IgM, what are the green lines surrounding the fibers? Since this staining is for necrotic fibers, how does it appear strongly on the cell membrane in FNDC1 group not in fibers?
16. What specific assays were used to confirm the activation of the FAK/PI3K/AKT/mTOR pathway by FNDC1 binding to integrin $\alpha 5 \beta 1$?
17. How does mFNDC1 compare to other potential therapeutic agents for muscle regeneration, such as glucocorticoids, in terms of efficacy and safety?
18. Can the therapeutic effects of mFNDC1 be sustained over the long term, or are there concerns about tolerance or resistance development?
19. Given the diverse roles of integrins in various cellular processes, are there any potential off-target effects of mFNDC1 via interactions with other integrins or pathways?
20. Can the findings regarding FNDC1's role in promoting myogenesis be extrapolated to other muscle-related conditions beyond acute and chronic muscle diseases, such as age-related muscle loss or sarcopenia?
21. Has the safety profile of mFNDC1 been assessed, particularly in terms of potential immunogenicity, toxicity, or adverse

effects on other tissues or organs?

Point-by-point response to the reviewer's comments

Dear Reviewers:

We greatly appreciate your constructive comments and valuable suggestions. In response, we have conducted additional experiments and provided explanations for new figures. We also revised the manuscript as requested (indicated by highlight). The following is a point-by-point response to address each of your comments.

Response to Reviewer #1

The authors identified FNDC1 as a novel myokine promoting myoblasts differentiation and muscle regeneration upon acute injury and in dystrophic mice. They demonstrate that FNDC1 interacts with integrin $\alpha 5\beta 1$ and activates FAK/PI3K/AKT/mTOR pathway. Although these findings are interesting and relevant for the muscle field, several main concerns should be addressed.

Response: We appreciate your positive comments describing our work as “interesting and relevant”.

Major points:

1) There is a general issue with the immunofluorescence staining that are not convincing. Indeed, the signals appear oversaturated in most of the pictures, casting doubt on their specificity. As an example, the signal for Pax7 is not convincing as it should colocalize with the nuclear staining and a membrane staining (e.g. laminin) should be added to undoubtedly identify the satellite cells based on their sublaminar localization.

Response: Thanks for your careful evaluation. To validate the immunofluorescence staining, we re-performed staining for FNDC1, MYOD, PAX7 and WGA involved in this study.

The specificity for FNDC1 staining was validated in skeletal muscle sections from CTX-injured mice using a blocking peptide specific for the first antibody (Catalog Number: bs-8460P). In the absence of the blocking peptide, the FNDC1 antibody exhibited labeling mainly in the cytoplasm, with few positive staining in nucleus and cell membrane (**Fig R1, Left**). However, FNDC1 staining was completely abolished when the antibody was preincubated with the FNDC1 Blocking Peptide (bs-8460P) (**Fig R1, Middle**). Moreover, no positive staining was observed in the secondary only antibody panel (**Fig R1, Right**). These results indicate that the FNDC1 antibody is specific.

Figure for reviewers removed

We have updated the images of immunofluorescence staining for FNDC1 in **Figure 1** as follows:

Figure 1. FNDC1 is closely associated with myogenesis. (G) Representative immunofluorescence staining of FNDC1⁺ fibers (in red) in untreated or CTX-treated TA muscle at day 5 post-injury. Cell membrane was stained with WGA (in green) and nuclei were counterstained with DAPI (in blue) (n = 6). Scale bar = 20µm. **(I)** Representative immunofluorescence staining of FNDC1⁺ fibers (in red) in TA muscle of WT or mdx mice. The cell membrane was stained with WGA (in green) and nuclei were counterstained with DAPI (in blue) (n = 6). Scale bar = 20µm.

Immunofluorescence staining of eMYHC was often used to label actively regenerating
muscle fibers, which were localized in the cytoplasm of muscle fibers with centrally
localized nuclei (Zheng X, et al. *Redox Biol.* 2023). We reperformed immunofluorescence
staining in skeletal muscle from CTX-treated mice and showed that eMYHC⁺ fibers were
only present in new myofibers with the central nucleus but not in uninjured myofibers,
suggesting the specificity of the eMYHC staining (**Fig R2**). To rule out the issue of signal
oversaturation in the picture, the immunofluorescence staining images of eMYHC were
replaced as follows (**Fig 6C and 7B**):

Figure for reviewers removed

**Figure 6. FNDC1 promotes skeletal muscle regeneration in CTX-induced muscle injury in**
**mice. (C)** Representative immunofluorescence staining of eMYHC⁺ fibers in Control or mFNDC1-
treated TA muscle at day 5 post-injury (n = 6). Staining for eMYHC (in red) marks the newborn
myofibrils. Cell membrane was stained with WGA (in green) and nuclei were counterstained with DAPI
(in blue). Scale bar = 50µm.

**Figure 7. ITGB1 is essential for FNDC1-promoted skeletal muscle regeneration in CTX-**
 **induced muscle injury in mice. (B)** Representative immunofluorescence staining of eMYHC⁺
 fibers in TA muscle at day 5 post-injury (n = 6). Staining for eMYHC (in red) marks the newborn
 myofibrils. Cell membrane was stained with WGA (in green) and nuclei were counterstained with
 DAPI (in blue). Scale bars = 50µm.

80

Furthermore, as suggested, we performed co-staining with PAX7, MYOD, and WGA
 in post-injury muscle sections to identify satellite cells (**Appendix Fig S17A and D**).

**Appendix Figure S17. FNDC1 promotes satellite cell differentiation during skeletal muscle**
**regeneration. (A)** Immunofluorescence staining of TA muscle for WGA (in white), PAX7 (in green)
and MYOD (in red) in CTX-injured mice receiving mFNDC1 or AAV-sh*Fndc1* at day 3 post-injury (n
= 6). Nuclei were counterstained with DAPI (in blue). Scale bar = 150 μ m. **(D)** Immunofluorescence
staining of TA for WGA (in white), PAX7 (in green) and MYOD (in red) in CTX-injured mice
receiving mFNDC1 or AAV-sh*Fndc1* at day 5 post-injury (n = 6). Scale bar = 150 μ m.

**2) Apart from two experiments, all in vitro experiments have been performed with**
**C2C12 cell line, but the limitations associated with its usage are now firmly**
**established within the field. Hence, the authors should confirm the key experiments**
**in primary myoblasts (e.g. treatment with recombinant FNDC1 protein, pulldown**
**assays).**

**Response:** We appreciate your comments on the limitations associated with the use of
C2C12 cell line. As requested, we re-performed the following experiments in satellite cell-
derived primary myoblasts: ① the effect of mFNDC1 on myogenic differentiation of
primary myoblasts. mFNDC1 treatment promoted primary myoblast differentiation (**Fig 6J-**
**K and Appendix Fig S18D-E**) and increased mRNA expression of *Caveolin-3* and
*Myomaker* (**Appendix Fig S18F**), consistent with the results in C2C12 cells. ② the effect
of *Fndc1* knockdown by shRNA lentivirus on the differentiation of mouse primary
myoblasts. *Fndc1* knockdown reduced differentiation index and the fusion index in primary
myoblasts (**Fig 6J-K and Appendix Fig S18G-H**), as well as decreased mRNA expression
of *Caveolin-3* and *Myomaker* (**Appendix Fig S18I**). ③ We performed pulldown analyses
to detect the binding of FNDC1 to integrins (β 1, α 5 and α V) in mouse primary myoblasts,
the results showed physical interaction between FNDC1 and ITGB1, ITGA5 but not ITGAV
in primary myoblasts (**Appendix Fig S12A-D**). ④ Furthermore, changes in downstream
pathway of FNDC1- α 5 β 1 were also examined in mFNDC1-treated primary myoblasts,
showing significantly increased phosphorylation of FAK, AKT and mTOR (**Fig R3**). In
conclusion, results obtained in primary myoblasts were consistent with those in C2C12
cells.

Figure 6. FNDC1 promotes skeletal muscle regeneration in CTX-induced muscle injury in mice. (J) Immunofluorescence staining for MYOG (in red) in satellite cell-derived primary myoblasts after differentiation for 36 h. Isolated satellite cell from WT mice treated with recombinant proteins (Control or mFNDC1) or lentivirus (shControl or sh*Fndc1*) were induced to

119 differentiate for 36 h. Scale bars = 50µm. **(K)** Representative immunofluorescence staining of
 120 MYHC (in green) in satellite cell-derived primary myoblasts treated with mFNDC1 or sh*Fndc1* at
 121 day 3 post-differentiation (n = 6). Scale bars = 200µm.

 **Appendix Figure S18. FNDC1 promotes satellite cell-derived primary myoblasts**
 **differentiation during skeletal muscle regeneration in vitro. (D-E)** Quantification of fusion index
 and differentiation index in satellite cell-derived primary myoblasts treated with Control or mFNDC1
 at day 3 post-differentiation (n = 6). **(F)** mRNA expression of *Caveolin-3* and *Myomaker* in primary
 myoblasts treated with Control or mFNDC1 at day 3 post-differentiation (n = 6). **(G-H)**
 Quantification of fusion index and differentiation index in primary myoblasts treated with shControl
 or sh*Fndc1* at day 3 post-differentiation (n = 6). **(I)** mRNA expression of *Caveolin-3* and *Myomaker*
 in primary myoblasts treated with shControl or sh*Fndc1* at day 3 post-differentiation (n = 6). Two-
 tailed t-test was performed to compare all listed conditions unless otherwise noted, and data are
 represented as mean ± SEM. **p* < 0.05, ***p* < 0.01, and ****p* < 0.001.

**Appendix Figure S12. FNDC1 binds to ITGB1 and ITGA5 in primary myoblasts. (A-B)** Co-IP of
FNDC1 and Integrin β 1 (ITGB1) in primary myoblasts. **(C-D)** Immunoprecipitation of FNDC1 and
Integrin α v (ITGAV) or Integrin α 5 (ITGA5) in primary myoblasts.

Figure for reviewers removed

**3) The authors claim that FNDC1 promotes differentiation but not proliferation of**
**myoblasts. However, they show only fusion index and number of nuclei while they**
**should also calculate the differentiation index by performing staining for MF20. On**
**the same line, it would be interesting to evaluate whether the expression of FNDC1**
**is differently modulated in proliferating and differentiating myoblasts by performing**
**immunofluorescence staining for FNDC1 combined with Pax7, MyoD or MyoG on**
**single myofibers and/or on muscle sections after CTX or in mdx mice.**

**Response:** Thanks. As suggested, we calculated the differentiation index based on the
immunofluorescence staining of MF20 (Myosin Heavy Chain, MYHC) (**Fig 2A, G, Fig 3M**
**and Fig 6K**). ① In C2C12 cells, *Fndc1* knockdown decreased while FNDC1
overexpression increased the differentiation index (**Fig 2B and 2H**). ② Furthermore,
mFNDC1 treatment dose-dependently increased the differentiation index of C2C12 cells
(**Fig 3N**). ③ In satellite cell-derived myoblasts, mFNDC1 treatment significantly increased
while *Fndc1* knockdown significantly decreased differentiation index (**Fig 6K and**
**Appendix Fig S18E, H**).

To evaluate the expression of FNDC1 during myocyte proliferation and differentiation,
we performed immunofluorescence staining for FNDC1 combined with PAX7 and MYOD
in skeletal muscle sections of CTX mice. As shown in Fig.R4, FNDC1 is expressed in both
proliferating myoblasts (PAX7⁺MYOD⁻) and differentiated myoblasts (PAX7⁻MYOD⁺) (**Fig**
**R4A**). In addition, we examined FNDC1 protein levels in primary myoblasts. The results
showed no significant difference in the FNDC1 protein levels during the proliferation of

myoblasts, whereas the FNDC1 protein levels gradually increased during the
 differentiation of myoblasts (Fig R4B).

**Figure 2. FNDC1 promotes myoblast differentiation-mediated myogenesis in vitro. (A)**
 Representative immunofluorescence staining of MYHC in si-Control or si-Fndc1 (*Fndc1*
 knockdown) C2C12 cells at day 4 post-differentiation (n = 6). Staining for MYHC (in green) marks
 differentiated cells, and nuclei were counterstained with DAPI (in blue). C2C12 cells were
 transfected with si-Control or si-Fndc1 for 12h before the initiation of differentiation. Scale bars =
 100µm. **(B)** Quantification of differentiation index (n = 6). **(G)** Representative immunofluorescence
 staining of MYHC (in green) in Control or *Fndc1*-overexpressing (*Fndc1*-OE) C2C12 cells at day 4
 post-differentiation. Scale bars = 100µm. **(H)** Quantification of differentiation index (n = 6). Two-
 tailed t-test was performed to compare all listed conditions unless otherwise noted, and data are
 represented as mean ± SEM. **p* < 0.05, ***p* < 0.01, and ****p* < 0.001.

**Figure 3. FNDC1 is a novel myokine. (M)** Representative immunofluorescence staining of MYHC
 (in green) in C2C12 cells treated with different concentrations of mFNDC1 at day 4 post
 differentiation. Nuclei were counterstained with DAPI (in blue). Scale bars = 50µm. **(N)**

Quantification of differentiation index (n = 6). One-way ANOVA was performed to compare all listed
conditions unless otherwise noted, and data are represented as mean ± SEM. * $p < 0.05$, ** $p <$
0.01 , and *** $p < 0.001$.

**Figure 6. FNDC1 promotes skeletal muscle regeneration in CTX-induced muscle injury in**
**mice. (K)** Representative immunofluorescence staining of MYHC (in green) in satellite cell-derived
primary myoblasts treated with mFNDC1 or sh*Fndc1* at day 3 post-differentiation (n = 6). Scale
bars = 200µm.

**Appendix Figure S18. FNDC1 promotes satellite cell-derived primary myoblasts**
**differentiation during skeletal muscle regeneration in vitro. (E)** Quantification of differentiation
index in satellite cell-derived primary myoblasts treated with Control or mFNDC1 at day 3 post-
differentiation (n = 6). **(H)** Quantification of differentiation index in primary myoblasts treated with
sh*Control* or sh*Fndc1* at day 3 post-differentiation (n = 6). Two-tailed t-test was performed to
compare all listed conditions unless otherwise noted, and data are represented as mean ± SEM. * p
< 0.05 , ** $p < 0.01$, and *** $p < 0.001$.

Figure for reviewers removed

**4) The authors should compare the myogenic properties of FNDC1 (at least in vitro)**
**with other proteins of the family. Indeed, they mentioned in the discussion that**
**FNDC4 has been reported to promote myogenic differentiation (line 414).**

**Response:** Thanks. To compare the myogenic properties of FNDC1 with FNDC4 and
FNDC5 *in vitro*, we treated C2C12 cells with 0.1µg/ml or 1µg/ml recombinant FNDC1,
FNDC4 or irisin (recombinant FNDC5 protein), respectively. At day 5 post-differentiation,
both doses of mFNDC1, FNDC4 and irisin significantly increased the differentiation index
of C2C12 cells compared to the control (**Fig. R5**). The effects of FNDC1 on myoblast

differentiation at dose of 0.1µg/ml were more effective than those of corresponding doses
of FNDC4 and irisin. There was no significant difference between FNDC4 and irisin. The
effects of FNDC1 on myoblast differentiation at dose of 1µg/ml was not significantly
different from those of FNDC4 and irisin.

Figure for reviewers removed

**Minor points:**

**1) The authors should include more information in the M&M about the cell assays**
**(e.g. number of plated cells number of days in growth medium before switching to**
**DM) and check the numbering of the figures as they are not sequential in the text.**

**Response:** We apologize for not providing detailed information about the cell assays in the
Methods section. We now add this detailed information in the “Materials and Methods”
section:

"C2C12 cells obtained from China Infrastructure of Cell Line Resource were cultured
in high-glucose Dulbecco's modified Eagle medium (H30022.01, HyClone, CT, USA)

supplemented with 10% fetal bovine serum (FBS) (Z7186FBS-500, ZETA LIFE, CA, USA),
 and 1% penicillin/streptomycin. Cell density was assessed by a hemocytometer. 50 000
 cells per well (12-well culture plate) were seeded for experiments. Cells were cultured to
 80% confluence (approximately 24 hours), after which the growth medium was replaced
 with differentiation medium (high sugar DMEM containing 2% horse serum and 1%
 penicillin/streptomycin). The medium was changed and collected daily until the myotubes
 were fully mature. The collected medium was centrifuged to remove cell debris and then
 concentrated 100-fold by ultrafiltration tubes (Millipore UFC801024). Finally, medium
 FNDC1 was measured by ELISA (Cat#YJ111201, mlbio, Shanghai, PRC) and Western
 blot.”

We further checked the numbering of the figures in the manuscript and made
 corrections throughout the manuscript.

**2) In fig. 1B, the results of the Q-PCR should be confirmed by WB as done for C2C12.**

**Response:** Thanks for your suggestion. Western blots were performed for genes shown in
 **Fig 1B**. Three days after primary myoblast differentiation, protein levels of myogenic
 differentiation markers (MYOD, MYOG and MYHC) were significantly increased compared
 to day 0 (**Fig 1D**). Furthermore, protein levels of FNDC1, GNG2 and PLTP were
 significantly increased at day 3 post-differentiation, with no significant difference in the
 protein levels of CTSC and RGS2 (**Fig 1D and Fig R6**). This is consistent with the q-PCR
 results.

**Figure 1. FNDC1 is closely associated with myogenesis. (B)** mRNA expression of *Myh3*,
 *Fndc1*, *Gng2*, *Pltp*, *Ctsc*, and *Rgs2* in primary myoblasts during myogenic differentiation (n=5). **(D)**
 Representative immunoblot analysis of MYOD, MYOG, MYHC and FNDC1 in primary myoblasts
 during myogenic differentiation(n=3). Two-tailed t-test was performed to compare all listed
 conditions unless otherwise noted, and data are represented as mean ± SEM. **p* < 0.05, ***p* <
 0.01, and ****p* < 0.001.

Figure for reviewers removed

**3) In fig 2E, the authors should include a more representative image of the**
**quantified difference included in fig. S1A.**

**Response:** We apologize for not carefully check these images. As suggested, the image
of the **Fig 2F** has now been replaced with a more representative image of the quantified
difference included in **Appendix Fig S3A (original Fig. S1A).**

**Figure 2. FNDC1 promotes myoblast differentiation-mediated myogenesis in vitro. (F)**
**Representative immunoblot analysis of FNDC1, MYHC and MYOG (n = 3). Cells were collected at**
**day 4 post-differentiation.**

**Appendix Figure S3. The efficiency of FNDC1 interference and overexpression in C2C12**
 **cells. (A)** *Fndc1* interference efficiency. Quantification of FNDC1, MYHC and MYOG immunoblot (n
 = 3). Two-tailed t-test was performed to compare all listed conditions unless otherwise noted, and
 data are represented as mean ± SEM. **p* < 0.05, ***p* < 0.01, and ****p* < 0.001.

**4) Fig. 2A-B or 2K-L should be included in the supplementary as including both in**
 **the main figure is redundant.**

**Response:** Thanks for your suggestion. Fig. 2K-L in the original manuscript has now been
 moved to **Supplementary Fig 4A-C.**

**5) In fig 3A, why did the authors decide to analyze the level of FNDC1 in medium of**
 **C2C12 cells at day 3 and not at day 6 when they observed the highest level of mRNA**
 **(fig. 1E)? To exclude that the presence of FNDC1 might be due to a passive release**
 **following cell necrosis, it would be useful to also analyze the level of a necrotic**
 **marker (e.g. LDH).**

**Response:** Thanks. Based on mRNA and protein results, FNDC1 levels were significantly
 elevated on day 2 after differentiation of C2C12 cells (**Fig1C-E**). Therefore, medium levels
 of FNDC1 at day 3 were collected and assayed (**Fig 3A-B**). The results indicated that
 FNDC1 was secreted into the culture medium. In addition, we also detected medium levels
 of FNDC1 at day 6 and found higher levels than those of day 3 (**Fig 3A-B**). These results
 suggest that FNDC1 is secreted by myoblasts during differentiation.

To examine whether the presence of FNDC1 is due to a passive release following cell
 necrosis, we measured medium LDH levels in C2C12 cells. The results showed similar
 levels of LDH during the whole course of C2C12 cell differentiation (days 0, 3 and 6)
 (**Appendix Fig S6A**), excluding the presence of medium FNDC1 is due to the passive
 release after cell necrosis.

**Figure 1. FNDC1 is closely associated with myogenesis. (C)** mRNA expression of *Myh3*, *Fndc1*,
 *Gng2*, *Pltp*, *Ctsc*, and *Rgs2* in C2C12 cells during myogenic differentiation (n = 3). **(E)**
 Representative immunoblotting of MYOD, MYOG, MYHC and FNDC1 in C2C12 cells during
 myogenic differentiation at the indicated time points (n = 3). One-way ANOVA was performed to
 compare all listed conditions unless otherwise noted, and data are represented as mean ± SEM. **p*
 < 0.05, ***p* < 0.01, and ****p* < 0.001.

**Figure 3. FNDC1 is a novel myokine.** (A) Representative immunoblot analysis of medium FNDC1
 in C2C12 cells at days 0, 3 and 6 post-differentiation. For determination of released FNDC1, equal
 volumes of conditioned media were subjected to western blot analysis; Ponceau S staining was
 used as a loading control. (B) Medium FNDC1 levels in C2C12 cells at days 0, 3 and 6 post-
 differentiation (n = 6). One-way ANOVA was performed to compare all listed conditions, and data
 are represented as mean ± SEM. **p* < 0.05, ***p* < 0.01, and ****p* < 0.001.

**Appendix Figure S6. The LDH levels during differentiation of C2C12 cells.** (A) Medium LDH
 levels in cultured C2C12 cells at days 0, 3 and 6 post-differentiation (n = 6). One-way ANOVA was
 performed to compare all listed conditions, and data are represented as mean ± SEM.

**6) Does the difference between CTR and FNDC1-FN3 is significant in fig 3G-H? If yes,**
 **it should be indicated on the graph.**

**Response:** Thanks. The difference between Control and FNDC1-FN3 in **Fig 3F-G**
 **(Original Fig 3G-H)** is significant, and we now indicated their difference on the graph.

**Figure 3. FNDC1 is a novel myokine. (F-G)** Quantification of the fusion index (a MYHC+ cell with
at least three nucleus) and the nucleus distribution per myotube (n = 6). One-way ANOVA was
performed to compare all listed conditions unless otherwise noted, and data are represented as
mean ± SEM. **p* < 0.05, ***p* < 0.01, and ****p* < 0.001.

**7) The authors should explain how they determined the doses to be used in fig. 3N-**
**O and fig 6.**

**Response:** Thanks for your suggestion. Currently, there are no reports on the dose of
FNDC1 treatment *in vivo* and *in vitro*. In this study, to explore the optimal dose of FNDC1 in
regulating myogenic differentiation *in vitro*, we set dose gradients for FNDC1 in **Fig 3M-O**
based on studies of FNDC1 homolog (FNDC5) (Bosma et al., *Nat Commun*, 2016; Kim et
al., *Cell*, 2018; Reza et al., *Nat Commun*, 2017). The results showed that FNDC1 promotes
myogenic differentiation in a dose-dependent manner, and 0.1 µg/ml mFNDC1 was sufficient to
significantly promote myotube fusion.

**Figure 3. FNDC1 is a novel myokine. (M)** Representative immunofluorescence staining of MYHC
(in green) in C2C12 cells treated with different concentrations of mFNDC1 at day 4 post
differentiation. Nuclei were counterstained with DAPI (in blue). Scale bars = 50µm. **(N)**
Quantification of differentiation index (n = 6). **(O)** Quantification of the fusion index (a MYHC+
cell with at least three nucleus) (n = 6). One-way ANOVA was performed to compare all listed
conditions unless otherwise noted, and data are represented as mean ± SEM. **p* < 0.05, ***p* < 0.01,
and ****p* < 0.001.

In addition, to explore the optimal dose of FNDC1 to promote muscle regeneration *in vivo*,
we used different doses of FNDC1 to treat CTX-injured mice. We chose three doses of
mFNDC1 (1.25, 2.5, and 5 mg/kg body weight) for mouse treatment based on FNDC5 (Bosma
et al., *Nat Commun*, 2016; Kim et al., *Cell*, 2018; Reza et al., *Nat Commun*, 2017).
Preliminary testing of these mFNDC1 doses in CTX-treated mice showed significantly
improved muscle regeneration at day 5 post-injury compared to control (**Fig R7A-B**), exhibiting
similar effects between that of 2.5 and 5 mg/kg of mFNDC1. Meanwhile, we showed in non-
injured normal mice that both 2.5 mg/kg and 5 mg/kg of mFNDC1 induced high FAK
phosphorylation (**Fig R7C-D**). Thus, 2.5 mg/kg body weight of mFNDC1 was chosen for the
experiments in Figure 6.

Figure for reviewers removed

**8) The timepoint indicated in line 177 is not the same in the corresponding figure**
**legend (fig. S3A-B)**

**Response:** We apologize for the incorrect description in figure legend (**Appendix Fig**
**S8A-B; Original Fig S3A-B**). We have corrected the error. The details are as follows:

(Corrected line 169-172): “**On day 4 after differentiation**, mFNDC1 treatment increased
the proportion of large myotubes (>10 nuclei) (**Appendix Fig S8A-B**) and the myoblast
fusion index compared to controls in C2C12 cells (**Appendix Fig S8C**).”

**Appendix Figure S8. Identification of the purified recombinant FNDC1 protein. (A)**
 Representative immunofluorescence staining of MYHC in Control (dialysis buffer, DB) or
 mFNDC1 (truncated recombinant FNDC1 protein, 0.1µg/ml)-treated C2C12 cells **at day 4**
 **post-differentiation**. Staining for MYHC (in green) marks differentiated cells. Scale bar =
 300µm. **(B-C)** Quantification of the nucleus distribution per myotube and fusion index (a
 MYHC⁺ cell with at least three nucleus) (n = 6). **(D)** mRNA expression of *Caveolin-3* and
 *Myomaker* in Control or mFNDC1-treated C2C12 cells **at day 4 post-differentiation** (n =
 6). **(E)** Representative immunoblot analysis of FNDC1 and MYHC in Control or mFNDC1-
 treated C2C12 cells (n = 3). Two-tailed t-test was performed to compare all listed
 conditions unless otherwise noted, and data are represented as mean ± SEM. **p* < 0.05,
 ***p* < 0.01, and ****p* < 0.001.

 **9) In line 219, the authors mentioned fig. 3E-J which appear not to refer to the text.**
 **In addition, the authors mentioned the expression of α5β1 integrin but it is**
 **important to underline that α7β1 integrin is also expressed as it is a marker of**
 **MuSCs.**

**Response:** Thanks. In line 206-210: “It is well accepted that integrins are cell surface
 receptors composed of α and β subunits that form heterodimers⁴¹. Our results on FNDC1
 protein truncation test and pharmacological blockade of integrins revealed that the RGD-
 receptors are candidate receptors for pairing with the β1 subunit (**Fig 3E-I and Appendix**
 **Fig S10F-H).**” Figure 3E-I shows the results of the FNDC1 protein truncation test.

 We completely agree with you that α7β1 integrin is a marker of MuSCs and is highly
 expressed in skeletal muscle. Notably, α7β1 integrin was shown to be one of the laminin
 integrins (**purple area in Figure 1**) (Hynes et al., *Cell*, 2002, Takada et al., *Genome Biol*,
 2007). In this study, by performing FNDC1 domain truncation and pharmacological blockade
 of RGD-receptor integrins, we showed that FNDC1 binds to the RGD receptor integrins via
 FN3 domain (**Fig 3E-I and Appendix Fig S10F-H**). Therefore, we have not mentioned the
 expression of laminin-binding integrin α7β1 but detected the expression of α5β1 in skeletal
 muscle.

**Figure 1. Classification of integrin receptor families** (Hynes et al., *Cell*, 2002). In mammals, 18
 α -subunits and 8 β -subunits have been identified, which are capable of producing 24 different
 integrin complexes. In regard to ligand specificity, integrin complexes can be divided into four
 subfamilies. Integrin subunits that bind to each other to form heterodimers are connected by solid
 lines.

 **Figure 3. FNDC1 is a novel myokine.** (E) Representative immunofluorescence staining of MYHC
 (in green) in C2C12 cells differentiated for 4 days after transfection with Control (empty vector),
 FNDC1-FL, FNDC1-FN3 or FNDC1-PHA. Scale bar = 300 μ m. (F-G) Quantification of the fusion
 index (a MYHC⁺ cell with at least three nucleus) and the nucleus distribution per myotube (n = 6).
 (H) Representative immunoblotting of MYHC and MYOG in cells transfected with Control (empty
 vector), FNDC1-FL, FNDC1-FN3 or FNDC1-PHA (n = 3). Cells were collected at day 4 post-
 differentiation. (I) mRNA expression of *Caveolin-3* and *Myomaker* (n = 6). One-way ANOVA was
 performed to compare all listed conditions unless otherwise noted, and data are represented as
 mean \pm SEM. * p < 0.05, ** p < 0.01, and *** p < 0.001.

 **Appendix Figure S10. FNDC1 targeting integrin β 1 promotes myogenic differentiation. (F)**
 Representative immunofluorescence staining of MYHC (in green) in C2C12 cells from control,
 mFNDC1-treated, RGDs-treated, and mFNDC1+RGDs groups at day 4 post differentiation. Scale
 bars = 300µm. **(G-H)** Quantification of fusion index (a MYHC⁺ cell with at least three nucleus)
 and nucleus distribution per myotube (n = 6). One-way ANOVA was performed to compare all listed
 conditions unless otherwise noted, and data are represented as mean \pm SEM. * $p < 0.05$, ** $p < 0.01$,
 and *** $p < 0.001$.

 **10) Fig. 5E: the authors do not show the validation of FAK knockdown in C2C12**
 **cells.**

**Response:** Thanks. To validate FAK knockdown in C2C12 cells, we examined FAK
 expression in C2C12 cells by q-PCR and Western blot. The results showed that FAK
 mRNA and protein levels were decreased by more than 65% in C2C12 cells (**Appendix**
 **Fig S14A-B).**

 **Appendix Figure S14. Efficiency of FAK knockdown in C2C12 cells. (A)** mRNA expression of
 *Fak* in si-control and si-*Fak* C2C12 cells (n = 6). **(B)** Representative immunoblotting of FAK in si-
 control and si-*Fak* C2C12 cells (n = 3). Two-tailed t-test was performed to compare all listed
 conditions unless otherwise noted, and data are represented as mean \pm SEM. * $p < 0.05$, ** $p < 0.01$,
 and *** $p < 0.001$.

11) The authors should comment in the discussion that $\alpha 5\beta 1$ integrin is not
expressed only in muscle and that it is involved in various biological processes
such as osteogenesis, angiogenesis, and tumor progression. In addition, the
authors claim that they provide evidence that mFDNC1 treatment significantly
improves sarcomere integrity (line 468), but their data did not provide such
evidence. Hence, this sentence should be removed or modified.

**Response:** Thanks for your suggestion. We revised the discussion section accordingly as
follows:

(lines 386-404): “Although both FNDC1 and FNDC5/irisin are myokines, their
secretion patterns and receptors are different. FNDC1 is produced in response to
myogenesis or muscle regeneration, whereas FNDC5 is an exercise-induced myokine¹.
Previous studies have shown that FNDC4 and FNDC5 need to be cleaved before being
released into circulation^{2,3}. However, FNDC1 structurally has a typical secretory peptide
and absence of a hydrophobic transmembrane region. Similar molecular weight of FNDC1
was shown in culture medium and cell lysate by protein blotting, indicating a non-cleavage
secretion. Future studies exploring the mechanism of FNDC1 secretion should provide
further clarification. Functionally, FNDC1 and its two family members FNDC5 and FNDC4
are able to promote myogenic differentiation^{4,5}. Deciphering the mechanism underlying
FNDC1-promoted myogenesis reveals the activation of integrin $\alpha 5\beta 1$ -
FAK/PI3K/AKT/mTOR pathway. However, FNDC4 has been reported to promote myogenic
differentiation via activating the Wnt/ β -catenin pathway and integrin $\alpha V/\beta 5$ has been
identified as the receptor for FNDC5/Irisin in bone and fat^{2,5}. These findings demonstrate
the diversity of receptors for members of the FNDC family and the distinct regulatory
pathways implicated in myogenesis. In addition, the receptor for FNDC1, $\alpha 5\beta 1$ integrin, is
not expressed only in muscle, and that it is involved in a variety of biological processes
such as osteogenesis (Hamidouche Z, et al. *Proc Natl Acad Sci U S A*. 2009; PMID:
19843692)., angiogenesis (Cascone I, et al. *J Cell Biol*. 2005, PMID: 16157706), and
tumor progression (Dudvarski Stanković N, et al. *EMBO Mol Med*. 2018, PMID:
30065025). FNDC1, as a myokine, may be involved in these biological processes through
the circulation, which is a fertile area to explore.”

In addition, as suggested, we removed the sentence that mFDNC1 treatment
significantly improves sarcomere integrity.

**Response to Reviewer #2**

This work is an excellent and timely contribution, addressing novel mechanisms of
 muscle regeneration through diverse in vitro and in vivo methodologies. Not only
 does the paper reveal a protein with beneficial effects on injured muscles, but it also
 elucidates the underlying mechanism and suggests the utilization of a truncated
 active version of FNDC1 to enhance muscle regeneration in cases of injury and
 dystrophies. As such, its anticipated impact on the field is substantial. My main
 critique is about the lack of detailed methodology descriptions, which hinder a
 thorough understanding of how the experiments were done.

 Overall, the conclusions drawn in the manuscript are justified. However, in some
 instances, there is a lack of direct evidence to support the authors' conclusions,
 such as the absence of direct evidence that FNDC1 is a secreted protein.
 Additionally, the manuscript suffers significantly from a lack of detail and
 description of methodologies and experiments, which reduces clarity, may lead to
 confusion for the reader, and makes it difficult for this reviewer to assess major
 parts of the data.

 **Response:** Thanks for your positive comments, describing our work as “excellent and
 timely contribution”. These positive comments are very encouraging.

 **Major comments:**

**1. Please provide accession numbers or links to the four independent microarray**
 **datasets on GEO.**

**Response:** Thanks. We have added accession numbers for four independent microarray
 datasets on GEO in Fig 1A and Results section (91-95 line): (i) DEGs during C2C12 cell
 differentiation (**GSE4694**); (ii) DEGs during CTX-induced muscle regeneration in mice
 (**GSE56903**); (iii) DEGs in skeletal muscle of Duchenne muscular dystrophy (DMD)
 patients (**GSE38417**); and (iv) DEGs in gastrocnemius muscle of chronic muscular
 dystrophic (*mdx*) mice (**GSE466**).

**Figure 1. FNDC1 is closely associated with myogenesis. (A)** Venn diagram showing fifteen
overlapping differentially expressed genes among four independent microarray datasets related to
myogenesis.

**2. Fig 1D: to confirm myogenesis, please provide additional specific differentiation**
**markers, such as MYOD and myogenin.**

**Response:** Thanks. To confirm myogenesis, we examined the expression of myogenic
differentiation markers MYOD, MYOG and MYHC during C2C12 cell differentiation. The
results showed that MYOD protein levels were increased at the early stages (days 2-4),
and then decreased at day 6 during differentiation (**Fig 1E and Appendix Fig S1B**). The
MYOG and MYHC protein levels were significantly increased with differentiation, of which
MYOG levels peaked at day 4 and MYHC was highly expressed at the terminal stage (day
6) (**Fig 1E and Appendix Fig S1B**). Meanwhile, FNDC1 protein levels were gradually
increased during C2C12 cell differentiation (**Fig 1E and Appendix Fig S1B**), suggesting
that FNDC1 may be involved in the entire process of myoblast differentiation.

**Figure 1. FNDC1 is closely associated with myogenesis. (E)** Representative immunoblotting of
MYOD, MYOG, MYHC, and FNDC1 in C2C12 cells during myogenic differentiation at the indicated
time points (n = 3).

**Appendix Figure S1. FNDC1 is closely associated with myogenesis. (B)** Quantification of
MYOD, MYOG, MYHC, and FNDC1 protein levels in C2C12 cells during myogenic differentiation
at the indicated time points (n = 3). One-way ANOVA was performed to compare all listed

conditions unless otherwise noted, and data are represented as mean \pm SEM. * p < 0.05, ** p <
0.01, and *** p < 0.001.

**3. Fig 1F: "In CTX-treated mice...during the initial phase of muscle regeneration**
**(days 3-5) and downregulated on day 14 when myoblast fusion was decreased",**
**please support this statement and presented findings with images of the**
**regenerating tissue, so the reader can appreciate the regeneration process and cell**
**fusion.**

**Response:** Thank you for your suggestion. We now present images of the regenerated
tissue form CTX-treated mice. As shown in **Appendix Figure S2**, at day 3 after CTX-
treatment, the muscle is in the initial stage of regeneration, which is mainly characterized
by the infiltration of inflammatory cells; at days 5-7, inflammatory cell infiltration was
decreased and SCs differentiated and fused to form new muscle cells. The new fibers with
central nucleus begin to form to replace damaged fibers; at day 14, major muscle
structures were restored, and myoblast differentiation and fusion were reduced. Our
results are consistent with previous studies (Siles, L., et al. *Nat Commun.* 2019; He S, et al.
*J Clin Invest.* 2021). The mRNA expression of *Fndc1* was significantly up-regulated at the
initial stage of muscle regeneration (days 3-5) and down-regulated at day 14 when
myoblast fusion was reduced, which was similar to the expression of the myogenesis
marker *Myog* (**Fig 1F**).

**Appendix Figure S2. Skeletal muscle pathological phenotype of CTX-injured mice.**
Representative H&E staining of TA muscle sections in CTX-injured mice at the indicated time
points. Scale bar = 50µm.

 **Figure 1. FNDC1 is closely associated with myogenesis. (F)** mRNA expression of *Fndc1* and
 *Myog* in the tibialis anterior (TA) muscle of cardiotoxin (CTX) injured WT mice (n = 6). One-way
 ANOVA was performed to compare all listed conditions unless otherwise noted, and data are
 represented as mean \pm SEM. * $p < 0.05$, ** $p < 0.01$, and *** $p < 0.001$.

 **4. Fig 1G: the provided images do not sufficiently depict the extent of the injury and**
 **regeneration.**

**Response:** Thanks. H&E staining showed that new fibers with central nuclei began to
 form and replaced the damaged fibers on day 5 after CTX-induced injury (**Fig S2**). As
 shown in **Fig 1G**, compared to untreated skeletal muscle, new myofibrils with central
 nucleus appeared in skeletal muscle on day 5 after CTX-induced injury, suggesting the
 occurrence of injury and regeneration of skeletal muscle.

G

 **Figure 1. FNDC1 is closely associated with myogenesis. (G)** Representative
 immunofluorescence staining of FNDC1⁺ fibers (in red) in untreated or CTX-treated TA muscle at
 579 day 5 post-injury. Cell membrane was stained with WGA (in green) and nuclei were counterstained
 with DAPI (in blue). Scale bar = 20µm.

 **5. Fig 1I: it is unclear whether the observed staining pattern of FNDC1 in mdx mice**
 **muscles indicates a problem with staining specificity or represents its spatial**
 **cellular distribution. If the staining is specific, please comment on the altered**
 **distribution compared to CTX-treated mice. The observed pattern suggests that**
 **FNDC1 aggregates.**

**Response:** Thanks. The specificity for FNDC1 staining was validated in skeletal muscle
 sections from CTX-injured mice using a blocking peptide specific for the first antibody
 (Cat#bs-8460P). In the absence of the blocking peptide, the FNDC1 antibody exhibited

labeling mainly in the cytoplasm, with a few positive staining in the nucleus and
extracellular (**Fig R1, Left**). However, FNDC1 staining was completely abolished when the
antibody was preincubated with the FNDC1 Blocking Peptide (bs-8460P) (**Fig R1, Middle**).
Moreover, no positive staining was observed in the secondary only antibody panel (**Fig R1,**
**Right**). These results indicate that the FNDC1 antibody is specific.

Figure for reviewers removed

The possible reason for the FNDC1 aggregation observed in **Fig. 1I** could be due to
signal oversaturation in the immunofluorescence staining images. To avoid the resulting
staining specificity problems, we repeated the immunofluorescence experiments and
provided new images. Immunofluorescence analysis showed increased FNDC1
expression in *mdx* mice (**Fig. 1I**).

**Figure 1. FNDC1 is closely associated with myogenesis. (I)** Representative
immunofluorescence staining of FNDC1⁺ fibers (in red) in TA muscle of WT or *mdx* mice (n = 6).
The cell membrane was stained with WGA (in green) and nuclei were counterstained with DAPI (in
blue). Scale bar = 20µm.

**6. Fig 2A: Based on the legend, C2C12 cells were transfected. I could not find**
**anywhere is the manuscript how these cells were transfected. Transfecting C2C12**
**myoblasts with a transfection reagent is a serious challenge and the only way to**
**transfect C2C12 myotubes is with an adenovirus. Did the authors use short hairpin**

**RNAs? shRNA? Viruses or transfection reagents? Please describe the methodology**
**in the text in detail.**

**Response:** We apologies for the lack of clarity in the method of C2C12 cell transfection.
We completely agree with you that adenoviral transfection of C2C12 cells is an efficient
and classical method (Jiang, Peng et al. *Mol cell*, 2021; An, Yitai et al. *Dev cell*, 2017).
Transfection of C2C12 cells with transfection reagents is also a viable and effective way.
This method has been used and validated in many studies (Vogler TO, et al. *Nature*. 2018;
Pham TCP, et al. *Sci Adv*. 2023; Wohlwend M, et al. *Sci Transl Med*. 2021; Che Y, et al. *Sci*
*Adv*. 2023; Park SY, et al. *Nat Commun*. 2016).

For seeding and siRNA-mediated transfection in this study, C2C12 cells were seeded
at a density of 50000 cells per well in a 12-well plate. After 12 hours, siRNA and plasmids
transfection were performed using jetPRIME transfection reagent (Polyplus-transfection,
101000015). 12h later, the medium was replaced with differentiation medium containing
DMEM supplemented with 2% horse serum and 1% penicillin/streptomycin. The
differentiation medium was replaced every 2 days. Collect the medium and cells at the
indicated time points for subsequent experiments. The si-control, si-*Fndc1*, si-*Itgb1* and si-
*Fak* were synthesized from Genepharma (Shanghai, PRC). The siRNA sequences for
*Fndc1*, *Itgb1*, and *Fak* were as follows: *Fndc1*-siRNA1, 5'-
GUGGCCGGAAGAUGAAUUATT-3'; *Itgb1*-siRNA1, 5'- GAGGCUCUCAACUAUAAATT-
3'. *Itgb1D*-siRNA, 5'-GAAAAUCCGAUUUACAAGAGU-3', *Itgb1A*-siRNA, 5'-
CUGAUAUGGAAGCUUUUAAUG-3', *Fak*-siRNA, 5'-GGAAUAUGAGUUGAGAAU-3'.

According to your comment, we knock down FNDC1 in C2C12 cells by infection with
adenovirus (*Ad-Fndc1*) or transfection with *Fndc1* siRNA (si-*Fndc1*). At day 4 post-
differentiation of C2C12 cells, the expression of FNDC1 mRNA was significantly reduced
after infection with *Fndc1* adenovirus or transfection with *Fndc1* siRNA (**Fig R8A-B**).
Consistently, the protein levels of FNDC1 were significantly decreased in *Ad-Fndc1*
C2C12 cells and si-*Fndc1* C2C12 cells (**Fig R8C and Fig 2F**). Immunofluorescence
staining showed impaired myogenic differentiation and decreased fusion index in C2C12
cells infected with *Fndc1* adenovirus (**Fig R8D-E**). These results were consistent with
those derived from transfection of C2C12 cells using transfection reagents (**Fig 2A-B**).
Adenovirus infection is a better and more convincing method when compared to
transfection reagents in the context of C2C12 cells and we will apply this method in our
future work. Thanks for your constructive comment.

Figure for reviewers removed

**Figure 2. FNDC1 promotes myoblast differentiation-mediated myogenesis in vitro. (A)**

Representative immunofluorescence staining of MYHC in si-Control or si-Fndc1 (*Fndc1*

knockdown) C2C12 cells at day 4 post-differentiation (n = 6). Staining for MYHC (in green) marks
differentiated cells, and nuclei were counterstained with DAPI (in blue). C2C12 cells were
transfected with si-*Control* or si-*Fndc1* for 12h before the initiation of differentiation. Scale bars =
100µm. **(B)** Quantification of differentiation index (n = 6). **(F)** Representative immunoblot analysis
of FNDC1, (n = 3). Cells were collected at day 4 post-differentiation. Two-tailed t-test was performed
to compare all listed conditions unless otherwise noted, and data are represented as mean ± SEM. **p* <
0.05, ***p* < 0.01, and ****p* < 0.001.

**7. Figs 2F and 3F-H and other figures involving C2C12 trasfection: see comment #6,**
**it is unclear how C2C12 cells were transfected with a plasmid.**

**Response:** We apologize again for incomplete description of the method. We have now
included details of how C2C12 cells were transfected with plasmids in the Materials and
Methods section:

(p.28, L587-600): “For seeding and transfection, C2C12 cells were seeded at a density of
50000 cells per well in a 12-well plate. After 12 hours, siRNA and plasmids transfection
were performed using jetPRIME transfection reagent (Polyplus-transfection, 101000015).
12 hours later, the medium was replaced with differentiation medium containing DMEM
supplemented with 2% horse serum and 1% penicillin/streptomycin. The differentiation
medium was replaced every 2 days. The medium and cells were collected at the indicated
time points for subsequent experiments. The plasmids encoding FNDC1-His, FNDC1-FN3
(missing residues 519-1092 of the PHA03247 domain) and FNDC1-PHA (missing residues
38-332 and 1480-1571 of the FN3 domain), ITGB1A, ITGB1D, ITGAV were synthesized by
MiaoLing Biologicals (Shanghai, PRC).”

**8. Fig 3A-B: it is inaccurate to draw conclusions from the presented data due to the**
**absence of a loading control. In addition, the lack of a protocol describing how the**
**medium was prepared raises concerns about the origin of the presented protein,**
**perhaps it was released during cell necrosis? or apoptosis? Or it is part of cell**
**debris? The presented data does not support the assertion that this protein is**
**secreted to the medium by vital cells.**

**Response:** We apologize for the absence of a loading control in original Figure 3A-B,
which may lead to inaccurate conclusions. We recollected medium from C2C12 cells on
699 days 0, 3 and 6 of differentiation and assayed the levels of FNDC1. Ponceau staining was
700 used as loading controls. The result showed that FNDC1 is secreted by myoblasts during
differentiation and its levels were increased with the differentiation process **(Fig 3A-B)**.

**Figure 3. FNDC1 is a novel myokine.** (A) Representative immunoblot analysis of medium FNDC1
 in C2C12 cells at days 0, 3 or 6 post-differentiations. For determination of released FNDC1, equal
 volumes of conditioned media were subjected to western blot analysis; Ponceau S staining was used
 as a loading control. (B) Medium FNDC1 levels in C2C12 cells at days 0, 3 or 6 post-differentiation
 (n = 6). One-way ANOVA performed to compare all listed conditions, and data are represented as
 mean ± SEM. **p* < 0.05, ***p* < 0.01, and ****p* < 0.001.

We now provide a detailed protocol for medium preparation in the Materials and
 Methods section as follows:

(p.29, L582-585): “The medium was changed and collected daily until the myotubes were
 fully matured. The collected medium was centrifuged to remove cell debris and then
 concentrated 100-fold using ultrafiltration tubes (Millipore UFC801024). Finally, medium
 levels of FNDC1 were measured by ELISA (Cat#YJ111201, mlbio, Shanghai, PRC) and
 Western blot.”

As you mentioned, cell death such as necrosis or apoptosis leads to the release of
 intracellular proteins into the extracellular space. Therefore, we assessed cell death by
 LDH release, a key feature of cells death (Ramos, et al. *EMBO J*, 2024; Moench I, et al.
 *PNAS*, 2009), in culture medium of C2C12 cells to examine whether the presence of
 FNDC1 is due to a passive release following cell death. The results showed similar
 medium LDH levels during the whole course of C2C12 cell differentiation (days 0, 3 and 6)
 (Appendix Fig S6A), excluding the effect of cell death. Furthermore, as described above,
 cell debris was removed from the cell culture medium by centrifugation after collection,
 ruling out FNDC1 as part of the cell debris. We hope that these results support our
 conclusion that FNDC1 is secreted into the culture medium from C2C12 cells.

Appendix Figure S6. The LDH levels during differentiation of C2C12 cells. (A) Medium LDH levels in cultured C2C12 cells at days 0, 3 or 6 post-differentiation (n = 6). One-way ANOVA was performed to compare all listed conditions, and data are represented as mean \pm SEM.

9. Fig 3- again, I would be cautious in interpreting the data to conclude that FNDC1 is a secreted protein, as the authors have not provided direct evidence supporting this claim. Please note that this protein is found in the blood under conditions where skeletal muscle integrity is compromised, therefore the presence of muscle proteins in the serum is only expected.

Response: Thanks. As we showed above in response to question 8: ① *In vitro*, by ELISA and immunoblotting analysis, we showed that FNDC1 can be secreted by C2C12 cells (Fig 3A-B). ② *In vivo*, we used ELISA to detect serum levels of FNDC1 from uninjured mice (0d) and CTX-injured mice (3d, 5d, 14d) as well as WT and *mdx* mice (Fig 3C-D). The results showed that FNDC1 was detectable in serum under conditions where mouse skeletal muscle was intact (uninjured WT mice) as well as injured (Fig 3C-D). These results support that FNDC1 is a secreted protein.

Figure 3. FNDC1 is a novel myokine. (A) Representative immunoblotting of medium FNDC1 in C2C12 cells at days 0, 3 or 6 post-differentiations. For determination of released FNDC1, equal volumes of conditioned media were subjected to western blot analysis; Ponceau S staining was used as a loading control. **(B)** Medium FNDC1 levels in C2C12 cells at days 0, 3 or 6 post-differentiation (n = 6). **(C-D)** Serum levels of FNDC1 in WT or CTX-injured mice (C) and *mdx* mice (D) (n = 6). One-way ANOVA (B and C) and Two-tailed t-test (D) were performed to compare all

753 listed conditions unless otherwise noted, and data are represented as mean \pm SEM. * p < 0.05, ** p
< 0.01, and *** p < 0.001.

**10. Fig 3I: please provide additional more specific markers for cell differentiation**
**and fusion.**

**Response:** Thanks for your suggestion. As requested, we performed new experiments
and now provide the protein levels of the specific marker MYOG for cell differentiation.
FNDC1-FN3 overexpression increased the protein levels of MYHC and MYOG (**Fig 3H**
**and Appendix Fig S7B**), similar to full-length FNDC1, whereas FNDC1-PHA transfection
had no obvious effect on myocyte differentiation (**Fig 3H and Appendix Fig S7B**),
suggesting that the FN3 domain is essential for the myogenic function of FNDC1.
Furthermore, we determined the mRNA expression of fusion markers *Caveolin-3* and
*Myomaker* to evaluate the effect of FNDC1 on C2C12 cell fusion. The results showed that
either full-length FNDC1 or FNDC1-FN3 overexpression significantly increased *Caveolin-3*
and *Myomaker* mRNA expression, while FNDC1-PHA overexpression had no significant
effect on these gene expression compared to control (**Fig 3I**).

H

I

**Figure 3. FNDC1 is a novel myokine. (H)** Representative immunoblotting of MYHC and MYOG in
cells transfected with Control (empty vector), FNDC1-FL, FNDC1-FN3 or FNDC1-PHA (n = 3).
Cells were collected at day 4 post-differentiation. **(I)** mRNA expression of *Caveolin-3* and
*Myomaker* (n = 6). One-way ANOVA was performed to compare all listed conditions unless
otherwise noted, and data are represented as mean \pm SEM. * p < 0.05, ** p < 0.01, and *** p <
0.001.

B

**Appendix Figure S7. FN3 domain is essential for the myogenic function of FNDC1. (B)**
Quantification of MYHC and MYOG in cells transfected with Control (empty vector), FNDC1-FL,
FNDC1-FN3 or FNDC1-PHA (n = 3). Cells were collected at day 4 post-differentiation. One-way
ANOVA was performed to compare all listed conditions, and data are represented as mean ± SEM.
* $p < 0.05$, ** $p < 0.01$, and *** $p < 0.001$.

**11. Fig S4A- it is unclear how the experiment was done. Fndc1 knockdown in C2C12**
**myotubes is crucial, especially when cells are subsequently analyzed by RNAseq.**
**The use of adenoviruses is necessary to achieve such efficiency. However, without**
**detailed methodology provided, this reader is unable to evaluate the experiment and**
**its results effectively.**

**Response:** Thanks. The use of adenovirus or transfection reagent for C2C12 cell has
been compared in response to your question 6. Both methods worked well, with a better
efficiency for adenovirus. The detailed methodology for **Appendix Fig S9A** (Original Fig
S4A) is now provided as below:

“C2C12 cells were seeded at a density of 50000 cells per well in a 12-well plate. After 12
793 hours, siRNA transfection was performed using either control siRNA or *Fndc1* siRNA by
794 jetPRIME (Polyplus-transfection, 101000015) transfection reagent. After 12 hours, the
795 medium was replaced with differentiation medium containing DMEM supplemented with 2%
horse serum and 1% penicillin/streptomycin. The differentiation medium was replaced
every 2 days. Total RNA was extracted from *Fndc1* knockdown and control C2C12 cells at
4 days of differentiation using TRIzol reagent. The RNA-seq libraries were sequenced on
the Illumina NextSeq 500 platform at UNITED KAWA to produce over 60 million, 100
nucleotide paired-end reads per sample. The reads were then trimmed for sequencing
adapters and aligned to the reference mouse genome version mm10 (GRCm38) using
TopHat version 2.0.10. Gene quantification was performed on the mapped sequences
using the htseq-count software version 0.6.1. Differential expression analysis was
performed using the DESeq2 package. Log transformation was used to normalize raw
read counts and to calculate normalized expression counts. (Data can be accessed in
GEO database, GSE217125). DEGs were submitted to the GO and Kyoto Encyclopaedia
of Genes and Genomes (KEGG) websites for enrichment analysis.”

**Appendix Figure S9. Integrin β 1 is the receptor for FNDC1. (A)** GO term analysis of DEGs
($p < 0.05$) in *Fndc1* knockdown and control C2C12 cells after 4 days of differentiation.

**12. Fig 4A-B: unfortunately, I am unable to review the data because there is no**
**detailed description in the text or legend how the experiments were done.**

**Response:** We apologize for not providing a detailed description for Fig 4A-B. Figure 4A is
the result of mass spectrometry analysis of protein binding to FNDC1 in C2C12 cells. The
details are as follows:

"C2C12 cells were seeded in 90 mm culture plates and cultured in the DMEM. At 80%
confluence, cells were switched to differentiation medium (DMEM with 2% horse serum).
On day 3 of differentiation, the medium was switched to FreeStyle293 Expression Medium.
After 4 h of incubation, cells were chilled on ice for 10 min and then treated with 100 nM
His-tag mFNDC1 for 20 min. The cells were then incubated with 1.5mM DTSSP for 30
minutes on ice to cross-link after washing with 15ml cold PBS twice. The addition of a final
concentration of 20 mM Tris-pH 7.5 quenched the cross-linking. The cells were then
harvested and homogenized in 1ml RIPA buffer containing protease-inhibitor cocktail and
phosphatase-inhibitor cocktail. His-tag protein-purified magnetic beads were used to
enrich FNDC1-treated C2C12 cell lysates. Briefly, 1ml of whole cell extract was added to
the pre-treated magnetic beads and incubated at 4°C for 90 min with rotation. After 3
repeated washes in NP40-free wash buffer, 0.8 ml of elution buffer is added. To the
collected elution buffer, add protein loading buffer and denature at 95°C. The samples
were up-sampled to 4-12% gradient SDS-PAGE for separation, followed by Caulmers
Brilliant Blue staining. The gel was used for mass spectrometry analysis."

A

	Gene Symbol	Score Sequest HT	Peptides
FNDC1	35.25	6
ITGB1	24.36	4
APOA1	14.36	3
ATP6V0C	10.35	3
C3	10.6	3
CRYAB	8.32	4
EVPL	5.36	3
FGA	3.23	2
FGB	4.36	3
FGG	5.32	2

**Figure 4. Integrin α 5 β 1 is the receptor for FNDC1 in myoblast. (A)** The top 10 proteins
enriched with FNDC1 by cross-linking/Co-IP/MS analysis.

**Figure 4B** shows the Co-IP results of FNDC1 and Integrin β 1 (ITGB1) in C2C12 cells.
The details are as follows: "C2C12 cells were transfected with pcDNA3.1-His-FNDC1,
pcDNA3.1-His empty vector, pcDNA3.1-FLAG-ITGB1, or pcDNA3.1-FLAG empty vector
for 48h, respectively. Cells were digested with trypsin and washed once with PBS, then
lysed with hypotonic lysis buffer and incubated on ice for 30min. Total proteins were
immunoprecipitated with His or flag magnetic beads at 4°C overnight. After washing with

TPBS 3 times (5min each), the protein-bound beads were finally suspended in 20µl of 1×
 SDS-PAGE up sampling buffer. The samples were boiled at 95°C for 10 min and the
 supernatant was loaded onto a gel for immunoblotting.”

 **Figure 4. Integrin $\alpha 5\beta 1$ is the receptor for FNDC1 in myoblast. (B)** Co-IP of FNDC1 and
 Integrin $\beta 1$ (ITGB1) in C2C12 cells.

 **13. Fig S4C-D: please provide densitometric measurements of presented blots, i.e.**
 **ratio of phosphorylated/unmodified protein.**

**Response:** Thanks, we now add the ratio of phosphorylated/total FAK (**Appendix Fig**
 **S9C-D**, original Fig S4C-D). As shown in the **Appendix Fig S9C-D**, knockdown of *Fndc1*
 significantly decreased while mFNDC1 treatment markedly increased the pFAK/FAK ratio
 compared with the corresponding controls.

 **Appendix Figure S9. Integrin $\beta 1$ is the receptor for FNDC1. (C-D)** Representative
 immunoblotting and quantification of focal adhesion kinase (FAK) and phosphorylated FAK in si-
 control, si-*Fndc1*, or mFNDC1-treated C2C12 cells (n = 3). Cells were collected at day 4 post-
 differentiation. Two-tailed t-test was performed to compare all listed conditions unless otherwise
 noted, and data are represented as mean \pm SEM. **p* < 0.05, ***p* < 0.01, and ****p* < 0.001.

**14. Fig S6C- changes in MYHC levels may not necessarily indicate changes in cell**
**differentiation. Therefore, throughout the paper, whenever the authors wish to show**
**an effect on myogenesis, it is essential that they present specific myogenic markers**
**in addition to MYHC.**

**Response:** Thank you for your suggestions. To accurately assess the myogenesis shown
in **Appendix Fig S11C-D** (original Fig S6C-D), protein expression of MYOG was examined
to assess the degree of cell differentiation. We found that knockdown of *Itgb1D*, but not
*Itgb1A*, significantly reduced the protein expression of MYOG in mFNDC1-treated cells,
suggesting that *Itgb1D* is the receptor for FNDC1-mediated myogenesis (**Appendix Fig**
**S11C-D**, original Fig S6C-D). In addition, in accordance with your suggestion, throughout
the paper we have examined the levels of the myogenesis marker MYOG when assessing
myogenesis.

**Appendix Figure S11. Integrin $\alpha 5\beta 1D$ is the receptor for FNDC1. (C-D)** Representative
immunoblotting analysis of MYHC and MYOG in *Itgb1D* or *Itgb1A* knockdown C2C12 cells treated
with mFNDC1 at day 3 post differentiation (n = 3).

**16. p. 11: RNA-seq analysis and PCR validation revealed that only the $\alpha 5$ and αv , ...,**
**were detectable in skeletal muscle (Fig. S6E-F)", please explain how this shows**
**"which integrin α subunit pairs with the integrin $\beta 1$ for FNDC1 activation".**

**Response:** To determine which integrin alpha subunit pairs with the integrin beta1 chain
and is regulated by FNDC1, we performed the following experiments: (i) Based on the
references (Hynes et al., *Cell*, 2002, Takada et al., *Genome Biol*, 2007), the mammalian
integrins have been classified into RGD-receptor integrins (blue labelled in Figure 1
below from *Cell* 2002), laminin-receptor integrins, collagen-receptor integrins, and
leukocyte-specific receptor. Here, by performing FNDC1 domain truncation and
pharmacological blockade of RGD-receptor integrins, we showed that FNDC1 binds to the
RGD receptor integrins via FN3 domain (Fig 3E-I, and Appendix S10F-H). Hence, the α -
subunits that are capable of binding to the $\beta 1$ -subunit in the RGD-receptor integrins are $\alpha 5$, αv ,
and $\alpha 8$ (reasoned by Figure 1 below, *Cell*, 2002). (ii) By RNA-seq analysis and PCR
validation, we showed that $\alpha 5$ and αv are detectable in skeletal muscle (Appendix Fig S11E-
F). (iii) Immunoprecipitation analysis showed a physical interaction between FNDC1 and
integrin $\alpha 5$, but not αv in C2C12 cells (Fig 4C-D), suggesting that the integrin $\alpha 5\beta 1$ complex is
the receptor for FNDC1. (iiii) Furthermore, by performing extracellular pull-down assay and
surface plasmon resonance (SPR), we showed direct binding of FNDC1 to integrin $\alpha 5\beta 1$ (Fig
4E-G).

**Figure 1** (from Hynes et al., *Cell*, 2002). **Classification of integrin receptor families.** In
 mammals, 18 α -subunits and 8 β -subunits have been identified, which are capable of producing 24
 different integrin complexes. In regard to ligand specificity, integrin complexes can be divided into
 four subfamilies. Integrin subunits that bind to each other to form heterodimers are connected by
 solid lines.

**Figure 3. FNDC1 is a novel myokine.** **(E)** Representative immunofluorescence staining of MYHC
 (in green) in C2C12 cells differentiated for 4 days after transfection with Control (empty vector),
 FNDC1-FL, FNDC1-FN3 or FNDC1-PHA. Scale bar = 300 μ m. **(F-G)** Quantification of the fusion
 index (a MYHC+ cell with at least three nucleus) and the nucleus distribution per myotube (n = 6).
 **(H)** Representative immunoblotting of MYHC and MYOG in cells transfected with Control (empty
 vector), FNDC1-FL, FNDC1-FN3 or FNDC1-PHA (n = 3). Cells were collected at day 4 post-
 differentiation. **(I)** mRNA expression of *Caveolin-3* and *Myomaker* (n = 6). One-way ANOVA was

performed to compare all listed conditions unless otherwise noted, and data are represented as
 mean \pm SEM. * $p < 0.05$, ** $p < 0.01$, and *** $p < 0.001$.

**Appendix Figure S10. FNDC1 targeting integrin $\beta 1$ promotes myogenic differentiation. (F)**
 Representative immunofluorescence staining of MYHC (in green) in C2C12 cells from control,
 mFNDC1-treated, RGDs-treated, and mFNDC1+RGDs groups at day 4 post differentiation. Scale
 bars = 300µm. **(G-H)** Quantification of fusion index (a MYHC⁺ cell with at least three nucleus) and
 nucleus distribution per myotube (n = 6). One-way ANOVA was performed to compare all listed
 conditions unless otherwise noted, and data are represented as mean \pm SEM. * $p < 0.05$, ** $p < 0.01$,
 and *** $p < 0.001$.

**Appendix Figure S11. Integrin $\alpha 5\beta 1D$ is the receptor for FNDC1. (E)** mRNA expression of *Itgb1*,
 *Itga5*, and *Itgav* in si-control and si-*Fndc1* C2C12 cells (n = 6). **(F)** mRNA expression of *Itgb1*, *Itga5*,
 and *Itgav* in different depots of skeletal muscles from mice (n = 6). Two-tailed t-test (E and F) were
 performed to compare all listed conditions unless otherwise noted, and data are represented as
 mean \pm SEM. * $p < 0.05$, ** $p < 0.01$, and *** $p < 0.001$.

Figure 4. Integrin $\alpha 5\beta 1$ is the receptor for FNDC1 in myoblast. (C-D) Immunoprecipitation of FNDC1 and ITGAV or ITGA5 in C2C12 cells. (E-F) Pull-down assays. 100 nM His-tagged FNDC1 (truncated recombinant FNDC1 protein) was incubated with the indicated Flag-tagged integrin (5 nM) and then affinity adsorbed through a nickel column. Immunoblotting was performed to analyze protein interactions between integrins ($\alpha 5$ or αv) and mFNDC1 (E) or to analyze protein interaction between integrin $\alpha 5\beta 1$ and mFNDC1 (F). (G) Surface plasmon resonance analysis of the FNDC1-Integrin $\alpha 5\beta 1$ interaction. Numbers above curves indicate the analyte concentration tested. All experiments were performed in triplicate with a mobile phase of HBS-EP.

17. Fig 6A: "CTX-treated mice were given mFNDC1", how? Ip injection? Iv injection? In water?

Response: Sorry for the lack of method description. CTX-injured mice (6-week-old C57BL/6J) were treated with either control (dialysis buffer) or mFNDC1 (2.5 mg/kg body weight) every two days via intramuscular injection (IM) for two weeks starting with the treatment of CTX. We have added details of mouse treatment to the legend of **Fig 6A**.

Figure 6. FNDC1 promotes skeletal muscle regeneration in CTX-induced muscle injury in mice. (A) Experimental procedures. CTX-treated mice (6-week-old C57BL/6J) were either given Control (dialysis buffer, DB) or mFNDC1 (2.5 mg/kg body weight) every two days via intramuscular injection (IM) for two weeks starting with the treatment of CTX.

**18. Fig 6D-E and Fig 7A, p.13: "By day 5 post-injury.... inflammatory cells were**
**almost cleared in mFNDC1-treated mice", unfortunately I cannot appreciate the**
**author's claim from the presented data. Please quantitate the infiltrating cells as in**
**Fig S12G.**

**Response:** Thanks for your suggestion. To quantify inflammatory cell infiltrating, we
performed F4/80 staining in skeletal muscle of CTX-injured mice treated with control or
mFNDC1. Immunofluorescence staining showed reduced proportion of F4/80⁺
macrophages in the muscle of CTX-injured mice treated with mFNDC1 compared with that
in the control group (**Appendix Fig S15A-B**). Furthermore, we test whether the anti-
inflammatory effects of mFNDC1 depends on the integrin receptor. mFNDC1 treatment
decreased the number of F4/80⁺ cells in AAV-scramble mice, while these changes were
abolished following *Itgb1* knockdown (**Fig R9A-B**).

**Appendix Figure S15. FNDC1 inhibits the inflammatory cell infiltration in skeletal muscle of**
**CTX-injured mice. (A-B)** Representative images of immunofluorescence staining (A) and
quantification (B) for F4/80-labeled macrophages (in red) from CTX-injured mice treated with
mFNDC1 or control (n = 6). Scale bars = 100µm. Two-tailed t-test was performed to compare all
listed conditions unless otherwise noted, and data are represented as mean ± SEM. * $p < 0.05$, ** p
< 0.01 , and *** $p < 0.001$.

Figure for reviewers removed

**19. Fig 8: I appreciate the beautiful data on mdx mice. Could the authors provide a**
**mechanistic explanation for the beneficial effects of FNDC1 on dystrophic muscles?**
**Given the ongoing absence of dystrophin in these muscles, it remains unclear how**
**muscle performance can be enhanced under such conditions. How would the**
**cytoskeletal desmin and the bound myofibrils stabilized in a system lacking intact**
**dystrophin-glycoprotein complex?**

**Response:** We sincerely appreciate your insightful comments. We have reviewed the
references and made supplementation to discuss the integrin upregulation and activation
in rescuing Duchenne muscular dystrophy (DMD) and muscle injury in the revised
manuscript. The details are as follows:

"FNDC1 ameliorates the muscle pathological phenotype of *mdx* mice possibly via
activation of the integrin $\alpha 5\beta 1$ pathway. DMD is a chronic muscle disease accompanied by
muscle damage and sustained regeneration (Mazala, D. A. et al., *JCI Insight*, 2020).
Enhancement of sarcolemma integrity by integrins is an effective way to alleviate DMD
(Han, R. et al., *PNAS*, 2009; Wang, H. V. et al., *J Cell Biol*, 2008). It has been shown that
moderate upregulation or activation of integrin $\alpha 7\beta 1$ increases sarcolemma integrity and

reduces mechanical stress-induced muscle fiber damage in *mdx* mice (Burkin et al., *J Cell*
 *Biol*, 2001; Rooney et al., *PNAS*, 2009; Rooney et al., *J Cell Sci*, 2006; Sarathy et al., *Mol*
 *Ther*, 2017). In line with integrin $\alpha7\beta1$, integrin $\alpha5\beta1$ is required for long-term sarcomere
 integrity (Taverna, D. et al., *J Cell Biol*, 1998; Wang, H. V. et al., *J Cell Biol*, 2008).
 Deficiency of integrin $\alpha5$ impairs sarcomere integrity and results in muscular dystrophy
 (Prause, M. et al., *Acta Biotechnologica*, 1984; Wang, H. V. et al., *J Cell Biol*, 2008). Both
 integrin $\alpha5\beta1$ and $\alpha7\beta1$ are present in fibers at sites where mechanical stress occurs,
 thereby maintaining the connection between muscle fibers and the matrix (Wang, H. V. et
 al., *J Cell Biol*, 2008). We therefore deduced that FNDC1 binding to integrin $\alpha5\beta1$
 increases sarcolemma integrity and attenuates the pathological phenotype in *mdx* mice.
 Furthermore, dystrophin-glycoprotein complex, which connects the extracellular matrix to
 the actin cortex, play a role in maintaining the mechanical structure of the cell membrane
 and in signal transduction (Eid Mutlak Y, et al. *Nat Commun*, 2020; Waite A, et al. *Trends*
 *Neurosci*, 2012; Ronald Worton, et al. *Science*, 1995). Like the dystrophin-glycoprotein
 complex, integrin complexes mediate interactions between the ECM and the membrane
 cytoskeleton (Pang, Xiaocong et al. *Signal Transduct Target Ther*, 2023; D Choquet, et al.
 *Cell*, 1997; Nishizaka, T et al. *PNAS*, 2000). Thus, FNDC1 may regulate cytoskeletal
 desmin and myofibril stabilization via integrin complex in systems lacking an intact
 dystrophin-glycoprotein complex. Additional studies are needed to understand the
 mechanisms by which FNDC1 attenuates DMD and to evaluate its role in the treatment of
 chronic muscle disease.”

**Minor comment:**

**Fig 5C-D: please specify in the figure which phosphorylated residue in AKT the**
 **antibody detects.**

**Response:** Thanks. The phosphorylated residues in AKT (Ser473) and mTOR (Ser2448)
 were determined in these experiments and the information has now been added in the
 figure and figure legend of **Fig 5C-E.**

**Figure 5. FNDC1 improves myogenesis through activating the**
 **Integrin/FAK/AKT/mTOR pathway. (C-D)** Representative immunoblotting and
 quantification of total and phosphorylated AKT (Ser 473) and mTOR (Ser 2448) in si-
 *Fndc1* or mFNDC1-treated C2C12 cells (n = 3). **(E)** Representative immunoblotting and

quantification of total and phosphorylated AKT (Ser473) and mTOR (Ser 2448) in si-
control or si-*Fak* C2C12 cells treated with Control or mFNDC1 (n = 3).

**Response to Reviewer #3**

**This study investigates the role of FNDC1 as a novel myogenic regulator and**
**potential therapeutic agent for muscle regeneration and dystrophy. The authors**
**conducted a series of experiments using in vitro and in vivo models to elucidate the**
**mechanisms underlying FNDC1-mediated myogenesis and its therapeutic potential.**

**Response:** We highly appreciate your careful and insightful review of our manuscript. We
now addressed all of your questions, either by providing clarification, presenting additional
results, or by performing new experiments. Point-by-point responses to your comments
are given below.

**1. Please include a table listing the abbreviations, some of which lack explanations.**

**Response:** Thanks. We provide the following table of the abbreviations in the manuscript.

**Abbreviations**

ABCA1	Atp binding cassette transporter a1	ITGB1	Integrin beta1
APOE	Apolipoprotein e	mdx	Muscular dystrophic
BDNF	Brain-derived neurotrophic factor	MuSC	Muscle stem cell
BGN	Biglycan	MYH3	Myosin heavy chain 3
CK	Creatine phosphate kinase	MYHC	Myosin heavy chain
COL1A1	Collagen type i alpha 1	MYL4	Myosin light chain 4
COL3A1	Collagen type iii alpha 1	MYOD	Myogenic differentiation antigen
COL5A3	Collagen type v alpha 3	MYOG	Myoglobin
CTSC	Cathepsin c	NFKB1	Nuclear factor kappa b subunit 1
CTX	Cardiotoxin	PAX7	Paired box 7
CXCR4	Cxc receptor 4	PLTP	Phospholipid transfer protein
DEGS	Differentially expressed genes	POSTN	Periostin
DIA	Diaphragm muscle	QUA	Quadriceps muscle
DMD	Duchenne muscular dystrophy	RGS2	Regulator of g protein signaling 2
FAK	Focal adhesion kinase	SPR	Surface plasmon resonance
FN1	Fibronectin 1	TA	Tibial mouse
FNDC1	Fibronectin type iii domain containing 1	TGFBI	Transforming growth factor beta induced
GNG2	G protein subunit gamma 2	TNFA	Tumor necrosis factor-α
IL1A	Interleukin-1 A	TNNT2	Troponin t2
ISLR	Eucine-rich repeat	WGA	Wheat Germ Agglutinin

**2. If applicable, it would be beneficial to include one gene whose role in myogenesis**
**has been documented to compare with the other genes lacking documentation in**
**Figures 1B and C.**

**Response:** Thanks. As suggested, we now include *Myh3*, which has been documented to
be involved in myogenesis (Zhang H F, et al. *Sci Adv.* 2020; Venuti, J M et al. *J Cell Biol.*
1995), as a positive control. Previous studies showed that *Myh3* expression gradually
increased during the differentiation of mouse primary myoblasts and C2C12 cells (Cong X
X, et al. *Cell Death Differ.* 2020; Tao J, et al. *Nat Commun.* 2023). Compared to day 0,

*Fndc1* mRNA was increased approximately 8-fold after 3 days of differentiation in mouse
primary myoblasts (**Fig 1B**). Meanwhile, *Myh3*, *Gng2*, and *Pltp* mRNA expression was
increased, and the expression of *Ctsc* and *Rgs2* was unchanged with cell differentiation
(**Fig 1B**).

During C2C12 cell differentiation, *Myh3* and *Fndc1* mRNA expression was up-
regulated on day 2 and remained high throughout the differentiation stage (**Fig 1C**). The
mRNA expression of *Gng2*, *Pltp*, and *Ctsc* was slightly elevated during the terminal
differentiation stage (**Fig 1C**). On the contrary, mRNA expression of *Rgs2* showed a
decreasing trend on day 2 after differentiation (**Fig 1C**).

**Figure 1. FNDC1 is closely associated with myogenesis. (B-C)** mRNA expression of *Myh3*,
*Fndc1*, *Gng2*, *Pltp*, *Ctsc*, and *Rgs2* in primary myoblasts (n = 5) (B) and C2C12 cells (n = 3) (C)
during myogenic differentiation. Two-tailed t-test (B) or one-way ANOVA (C) was performed unless
otherwise noted, and data are represented as mean \pm SEM. * $p < 0.05$, ** $p < 0.01$, and *** $p < 0.001$.

**3. In Figure 1F, at day 0, both *Fndc1* and *Myog* exhibit similar relative mRNA**
**expression levels. However, during days 3 and 5, both genes show an increase, yet**
**after 14 days, *Myog* expression falls below that of *Fndc1*, which was higher during**
**days 3 and 5. Can you provide an explanation for this trend?**

**Response:** Thanks for your question. During CTX-induced injury in mice, the expression
pattern of *Fndc1* and *Myog* was in general similar, showing a gradual increase and then a
final stage reduction (**Fig 1F**). The reason for a more abrupt reduction in the *Myog*
expression than in *Fndc1* at day 14 may be due to the fact that *Myog* is highly expressed
in differentiated myoblasts. At day 14, myoblast differentiation and fusion are reduced
(Siles, L., et al. *Nat Commun*, 2019; He S, et al. *J Clin Invest*, 2021), thus the expression
of *Myog* expression is significantly reduced at this time point. This expression pattern is
consistent with the results in several publications (He Shengqi et al, *J Clin Invest*, 2021;
Zhao, Yu et al. *Nat Commun*, 2019). In regard to *Fndc1*, we showed that it promotes
myogenesis during CTX-induced injury and is crucial for regeneration. At day 14, the
regeneration process is almost complete, so its expression is decreased. Meanwhile, new
myofibrils were formed, but the maturation and remodeling of new myofibrils were not yet
completed (Chargé SB, et al, *Physiol Rev*, 2004; Wang Xun, et al, *J Clin Invest*, 2022).
The observed *Fndc1* (at day 14), which is still higher than its baseline (at day 0), may play
an unknown role in the reconstruction of new myofibrils. This merits further exploration.

 **Figure 1. FNDC1 is closely associated with myogenesis. (F)** mRNA expression of *Fndc1* and
 *Myog* in the tibialis anterior (TA) muscle of cardiotoxin (CTX) injured WT mice (n = 6). One-way
 ANOVA was performed to compare all listed conditions unless otherwise noted, and data are
 represented as mean ± SEM. * $p < 0.05$, ** $p < 0.01$, and *** $p < 0.001$.

 **4. In Figure 1I, it appears that the scale bars for WT and mdx are not consistent and**
 **need to be verified. Additionally, WGA staining is incomplete in some parts of the**
 **fibers, and the image quality is inferior to that of the WT. It requires clarification.**

**Response:** Thanks for your careful examination of the Figure. We checked all scales to
 avoid such problems. WGA staining was incomplete in some parts of the fibers, which was
 a quality problem with the image staining, and we re-performed the immunofluorescence
 staining and provided the new images.

 **Figure 1. FNDC1 is closely associated with myogenesis. (I)** Representative
 immunofluorescence staining of FNDC1⁺ fibers (in red) in TA muscle of WT or *mdx* mice. The cell
 membrane was stained with WGA (in green) and nuclei were counterstained with DAPI (in blue) (n
 = 6). Scale bar = 20µm.

 **5. Authors concluded in line 124 that "immunofluorescence analysis showed**
 **significantly higher FNDC1 in the muscle fibers of mdx mice compared with WT**
 **mice." However, the images do not sufficiently support this statement. To**
 **strengthen this conclusion, higher-quality images should be provided.**

**Response:** Thanks. We now provide higher-quality images to support our conclusions.
Quantification of FNDC1 showed that FNDC1 was significantly higher in the muscle fibers
of *mdx* mice than in WT mice.

**Figure 1. FNDC1 is closely associated with myogenesis. (I)** Representative
immunofluorescence staining of FNDC1⁺ fibers (in red) in TA muscle of WT or *mdx* mice. The cell
membrane was stained with WGA (in green) and nuclei were counterstained with DAPI (in blue) (n
= 6). Scale bar = 20µm.

**6. The cells used as controls in Figures 2C and H are the same, yet the percentage**
**of nucleus differs between these two charts. Can you explain this discrepancy?**

**Response:** Thanks for your question. Although both **Figure 2D** (original Figure 2C) and
**Figure 2J** (original Figure 2H) have controls, they underwent different treatment. Cells
were transfected with control siRNA in Figure C, and with pcDNA3.1 vector in Figure 2H.
This might be one reason for the difference. Also, they are cells cultured in different
batches. Batch effects might be another reason.

To waive your concern, we reperformed the immunofluorescence staining and
quantification of MYHC in Figure 2A and G with same batch of cultured cells. There were
no significant differences in differentiation index, fusion index and the percentage of
nucleus between control cells transfected with control siRNA in **Figure 2A-D** and
pcDNA3.1 in **Figure 2G-J**. Furthermore, in **Figure 2A-D**, compared to control cells
transfected with si-control, knockdown of *Fndc1* significantly decreased multinuclear myotube
formation events. In contrast, in **Figure 2G-J**, compared to control cells transfected with
pcDNA3.1, FNDC1 overexpression increased the formation of myotubes. These results
indicate that FNDC1 promotes myogenic differentiation, which is consistent with our
original conclusion.

Figure 2. FNDC1 promotes myoblast differentiation-mediated myogenesis in vitro. (A)

Representative immunofluorescence staining of MYHC in si-Control or si-Fndc1 (*Fndc1* knockdown) C2C12 cells at day 4 post-differentiation (n = 6). Staining for MYHC (in green) marks differentiated cells, and nuclei were counterstained with DAPI (in blue). C2C12 cells were transfected with si-Control or si-Fndc1 for 12h before the initiation of differentiation. Scale bars = 100µm. **(B)** Quantification of differentiation index (n = 6). **(C-D)** Quantification of fusion index (a MYHC⁺ cell with at least three nucleus) and the nucleus distribution per myotube (n = 6). **(G)** Representative immunofluorescence staining of MYHC (in green) in Control or *Fndc1*-overexpressing (*Fndc1*-OE) C2C12 cells at day 4 post-differentiation. Scale bars = 100µm. **(H)** Quantification of differentiation index (n = 6). **(I-J)** Quantification of fusion index and the nucleus

distribution per myotube (n = 6). Two-tailed t-test was performed to compare all listed conditions
unless otherwise noted, and data are represented as mean ± SEM. * $p < 0.05$, ** $p < 0.01$, and *** p
< 0.001 .

**7. The sentence in line 205, "These results suggest that ITGB1 is the receptor for**
**FNDC1," may be overstated. It would be more appropriate to state: "These results**
**suggest that ITGB1 could be a receptor for FNDC1."**

**Response:** Thanks. We fully agree with your suggestion and revised the statement as
suggested in line 197-198.

**8. In Figure 6C, please provide an image with a larger tissue area.**

**Response:** As shown below, we provide an image with a larger tissue area (left panel).
Immunofluorescence staining of eMYHC, a marker for newly formed muscle fibers during
myogenesis and muscle regeneration, revealed significantly increased positive staining in
mFNDC1-treated mice at day 5 post-injury (**Fig 6C**).

**Figure 6. FNDC1 promotes skeletal muscle regeneration in CTX-induced muscle injury in**
**mice. (C)** Representative immunofluorescence staining of eMYHC⁺ fibers in Control or mFNDC1-
treated TA muscle at day 5 post-injury (n = 6). Staining for eMYHC (in red) marks the newborn
myofibrils. Cell membrane was stained with WGA (in green) and nuclei were counterstained with
DAPI (in blue). Scale bar = 50µm.

**9. In Figure S9A, the staining of PAX7 and MYOD is not sufficiently clear to**
**accurately determine the percentage of PAX7 and MYOD, making comparisons**
**between different groups unreliable.**

**Response:** We apologize for the unclear images provided in the **Appendix Figure S17A**
(original Figure S9A). we now re-stained and quantified PAX7, MYOD, and Wheat Germ
Agglutinin (WGA) as follows. The results showed that FNDC1 does not affect the
percentage of PAX7⁺MYOD⁻ and PAX7⁺MYOD⁺ cells from CTX-injured mice at day 3 post-
injury (**Appendix Fig S17B-C**).

Appendix Figure S17. FNDC1 promotes satellite cell differentiation during skeletal muscle regeneration *in vivo* (A) Immunofluorescence staining of TA muscle for WGA (in weight), PAX7 (in green) and MYOD (in red) in CTX-injured mice receiving mFNDC1 or AAV-shFndc1 at day 3 post-injury (n = 6). Nuclei were counterstained with DAPI (in blue). Scale bar = 150 μ m. (B-C) Quantification of different states of satellite cell (MuSC) in TA muscle from mice in Figure (A) (n = 6).

10. In Figure 6E, please set the y-axis origin to zero (it currently starts at 1200, thus overrepresenting the difference in fiber size).

Response: Thanks. As suggested, we set the y-axis origin to zero in **Figure 6E**. The results showed that the average myofiber size of mFNDC1-treated mice was significantly increased compared to control mice.

Figure 6. FNDC1 promotes skeletal muscle regeneration in CTX-induced muscle injury in mice. (E) Average CSA of eMYHC⁺ myofiber in TA muscle from CTX-injured mice treated with Control or mFNDC1 at day 5 post-injury (n = 6). Two-tailed t-test was performed to compare all

1195 listed conditions unless otherwise noted, and data are represented as mean \pm SEM. * p < 0.05, ** p
< 0.01, and *** p < 0.001.

**11. In Figure 6J, please provide an image with a larger area or lower magnification**
**for clarity.**

**Response:** Thanks. As suggested, we now provide an image with a larger area in **Figure**
**6J**. Immunofluorescence staining showed that mFNDC1 treatment increased, while *Fndc1*-
knockdown decreased the percentage of MYOG⁺ cells compared with their respective
controls, demonstrating that FNDC1 promotes MuSCs differentiation.

**Figure 6. FNDC1 promotes skeletal muscle regeneration in CTX-induced muscle injury in**
**mice. (J)** Immunofluorescence staining for MYOG (in red) in satellite cell-derived primary
myoblasts after differentiation for 36h. Isolated satellite cell from WT mice treated with recombinant
proteins (Control or mFNDC1) or lentivirus (shControl or sh*Fndc1*) were induced to differentiate for
36h. Scale bars = 50µm.

**12. In Figure 6A, the arrows are green, contrary to the mention of blue in line 978.**
**Please verify this discrepancy.**

**Response:** Sorry for the wrong description. This has been corrected now.

**Figure 6. FNDC1 promotes skeletal muscle regeneration in CTX-induced muscle injury in**
 **mice. (B)** Representative H&E staining of TA muscle sections at days 5 and 14 post-injury in
 Control or mFNDC1-treated mice (n = 6). **Yellow arrows** indicate inflammatory infiltrates and
 **green arrows** indicate newborn muscle fibers. Scale bar = 20µm.

**13. In Figure 6K, please explain how many microscopic fields did the authors count**
 **for evaluating the data, which the authors concluded promote the formation of**
 **multinucleates in line 295.**

**Response:** Thanks. In Figure 6K, the data were evaluated by counting 6 microscopic
 fields per group. We now provide all microscopic fields of view.

**Figure 6. FNDC1 promotes skeletal muscle regeneration in CTX-induced muscle injury in**
 **mice. (K)** Representative immunofluorescence staining of MYHC (in green) in satellite cell-derived
 primary myoblasts treated with mFNDC1 or sh*Fndc1* at day 3 post-differentiation (n = 6). Scale
 bars = 200µm.

Figure for reviewers removed

**14. For Figure 7B, a larger tissue area needs to be provided.**

**Response:** Thanks. As suggested, a larger area image is provided in **Figure 7B**.
Immunofluorescence staining of eMYHC showed that mFNDC1 treatment promoted
muscle regeneration, whereas this effect was abolished in AAV-sh*Itgb1* mice, suggesting
that the promotive effect of mFNDC1 on muscle regeneration was ITGB1-dependent.

Figure 7. ITGB1 is essential for FNDC1-promoted skeletal muscle regeneration in CTX-induced muscle injury in mice. (B) Representative immunofluorescence staining of eMYHC⁺ fibers in TA muscle at day 5 post-injury (n = 6). Staining for eMYHC (in red) marks the newborn myofibrils. Cell membrane was stained with WGA (in green) and nuclei were counterstained with DAPI (in blue). Scale bars = 50µm.

15. In Figure S12J, regarding the immunofluorescence staining of IgM, what are the green lines surrounding the fibers? Since this staining is for necrotic fibers, how does it appear strongly on the cell membrane in FNDC1 group not in fibers?

Response: Thanks. Immunostaining for the serum markers of membrane impermeability, immunoglobulin M (IgM) or immunoglobulin G (IgG) protein, was performed as a measure of muscle necrosis. As shown by immunohistochemical staining from the publication (Straub V, *J Cell Biol.* 1997), in the skeletal muscle of uninjured mice, IgM or IgG mainly localized in the perimysium and myocyte membranes but not in the intracellular fibers. However, in *mdx* mice, as shown by immunofluorescence staining from the publications (Leduc-Gaudet et al. *Nat Commun.* 2023; Boyer, Justin G et al. *Nat Commun.* 2022), positive staining of IgG and IgM was found in the cytoplasm and cell membrane of the skeletal muscle.

Figure (c-d) from The Journal of cell biology (PMID: 9334342). Immunohistochemical staining of 7µm femoral quadriceps cryosection from normal and *mdx* mice with antibodies against mouse IgM (c and d) showed accumulation of the serum markers in *mdx* skeletal muscle.

Figure C from *JCI insight* (PMID: 32493839). (C) Necrosis assessment by immunofluorescent staining of IgM and IgG (green) binding and its calculation indicating no differences between groups; n = 9-10/group. Scale bar: 100 µm.

These results from the previous publications showed positive staining of IgG and IgM in the cytoplasm and cell membrane of skeletal muscle from *mdx* mice and dKO mice. Thus, when cell integrity was disrupted, this marker could present in the different subcellular parts of muscle cells.

As shown in **Appendix Fig S20J** (Original Fig S12J), IgM accumulated predominantly in the myofiber cytoplasm and cell membrane of skeletal muscle from *mdx* mice, which was consistent with the above listed publications. However, in *mdx* mice treated with mFNDC1, IgM was significantly reduced in the cytoplasm and was mainly present in the myocyte membrane, suggesting that mFNDC1 significantly attenuated myofibril necrosis in *mdx* mice (**Appendix Fig S20J-K**).

Appendix Figure S20. Effects of FNDC1 on muscle weight and pathology indices in mdx mice. (J-K) Representative immunofluorescence staining for IgM (in green) of fibers (**J**) and quantification of the area of muscle fiber necrosis (**K**) in TA muscle from *mdx* mice treated with Control or mFNDC1 for 4 weeks (n = 6). Scale bars = 50μm. Two-tailed t-test was performed to compare all listed conditions unless otherwise noted, and data are represented as mean ± SEM. **p* < 0.05, ***p* < 0.01, and ****p* < 0.001.

16. What specific assays were used to confirm the activation of the FAK/PI3K/AKT/mTOR pathway by FNDC1 binding to integrin α5β1?

Response: Thanks. To confirm the activation of the FAK/PI3K/AKT/mTOR pathway by FNDC1 binding to integrin α5β1, we performed the following experiments: ① *In vitro*, we observed increased FAK phosphorylation in mFNDC1-treated C2C12 cells (**Appendix Fig S10D-E and Appendix Fig S13A-D**). Knockdown of *Itgb1* in myoblasts inhibited mFNDC1-induced elevations in p-FAK (**Appendix Fig S10D-E**). Furthermore, we found that knockdown of *Itgb1* or pharmacological inhibition of α5β1 by K34c completely blocked FNDC1-induced phosphorylation of AKT and mTOR in C2C12 cells (**Appendix Fig S13A-D**). ② *In vivo*, we generated *Itgb1* knockdown mice by injecting AAV-sh*Itgb1* into mouse skeletal muscle. Mice were subjected to a single CTX injury followed by mFNDC1 treatment. Protein levels of phosphorylation of FAK, AKT, and mTOR in skeletal muscle of mice were determined (**Fig 7E-F**). The results showed that mFNDC1 treatment increased protein levels of phosphorylation of FAK, AKT, and mTOR in AAV-scramble mice, while these changes were abolished in AAV-*Itgb1* knockdown mice (**Fig 7E-F**). These results suggest that FNDC1 activates FAK/PI3K/AKT/mTOR pathway in an α5β1-dependent manner.

**Appendix Figure S10. FNDC1 targeting integrin $\beta 1$ promotes myogenic differentiation. (D-E)**

Representative immunoblotting (D) and quantification (E) of indicated proteins in si-Control or si-

*Itgb1* C2C12 cells treated with control or mFNDC1 at day 4 post differentiation ($n = 3$). One-way

ANOVA were performed to compare all listed conditions unless otherwise noted, and data are

represented as mean \pm SEM. * $p < 0.05$, ** $p < 0.01$, and *** $p < 0.001$.

**Appendix Figure S13. FNDC1 improves myogenesis through activating the AKT/mTOR**
 **pathway (A-B)** Representative immunoblotting (A) and quantification (B) of indicated proteins in
 Control (PBS) or K34c treated C2C12 cells receiving Control or mFNDC1 (n = 3). **(C-D)**
 Representative immunoblotting (C) and quantification (D) of total and phosphorylated AKT and
 mTOR in si-Control or si-*Itgb1* C2C12 cells treated with Control or mFNDC1 at day 4 post
 differentiation (n = 3). One-way ANOVA was performed to compare all listed conditions unless
 otherwise noted, and data are represented as mean ± SEM. **p* < 0.05, ***p* < 0.01, and ****p* < 0.001.

 **Figure 7. ITGB1 is essential for FNDC1-promoted skeletal muscle regeneration in CTX-**
 **induced muscle injury in mice. (E-F)** Representative immunoblotting (E) and quantification (F) of
 indicated proteins in TA muscle at day 5 post-injury (n = 3). One-way ANOVA was performed to
 compare all listed conditions unless otherwise noted, and data are represented as mean ± SEM. **p*
 < 0.05, ***p* < 0.01, and ****p* < 0.001.

**17. How does mFNDC1 compare to other potential therapeutic agents for muscle**
 **regeneration, such as glucocorticoids, in terms of efficacy and safety?**

**Response:** Thanks for your insightful question. Intermittent use of glucocorticoids such as
 Prednisone enhances muscle repair without causing muscle atrophy and is the standard of
 care for the treatment of Duchenne muscular dystrophy (Salamone, et al. *J Clin Invest*,
 2022). To compare the efficacy of mFNDC1 with glucocorticoids in the treatment of *mdx*
 mice, we set up new experiments: ① Glucocorticoid-treated group: *mdx* mice (4 weeks old)
 were given intraperitoneal injections of prednisone (1 mg/kg once a week for 8 weeks, Ref.
 Quattrocelli, et al. *J Clin Invest*. 2017, PMID: 28481224). ② mFNDC1 treatment group:
 *mdx* mice (4 weeks old) were injected intraperitoneally with mFNDC1 (2.5 mg/kg every two
 1333 days for 8 weeks). Consistent with previous studies, *mdx* mice treated with weekly
 prednisone maintain their body weight compared with reduced weight in controls (**Fig.**
 **R11A**). In contrast, mFNDC1 treatment is able to increase the body weight of *mdx* mice
 (**Fig R11A**). Muscle damage was further assessed by serum CK levels, showing
 significantly reduced levels by either mFNDC1 or prednisone treatment compared with
 their respective controls (**Fig. R11B**). To further examine muscle performance, mice were
 tested by running on an inclined treadmill until exhaustion. mFNDC1 and Prednisone
 treatments both increased the running distance of *mdx* mice in a similar extent (**Fig R11C**).

In line with this, grip strength was significantly increased in mFNDC1 and prednisone-
treated *mdx* mice (**Fig R11D**). H&E staining showed that mFNDC1 and Prednisone
treatment reduced the number of muscle fibers with smaller cross-sectional areas and
increased the mean muscle fiber area in *mdx* mice (**Fig R11E-F**). In addition, the
expression of *Myog* and *Myh3*, markers of muscle regeneration, were significantly
increased in mFNDC1 or Prednisone-treated mice compared to their respective controls.
These results suggest that in terms of efficacy for muscle regeneration, mFNDC1 exhibits
similar effects compared to the widely used glucocorticoids.

In addition, we evaluated the safety of mFNDC1 by assaying markers of liver and
kidney functions in *mdx* mice. Circulating aspartate aminotransferase (AST) and alanine
aminotransferase (ALT) levels were significantly reduced in *mdx* mice treated with
mFNDC1 (**Fig R12A-B**), whereas there were no significant differences in uric acid (UA)
and blood urea nitrogen (BUN) (**Fig R12C-D**). Consistently, no morphological
abnormalities were found in the liver and kidney, indicating that mFNDC1 did not cause
toxicity (**Fig R12E**). Long-term systematic comparison of the safety and efficacy of
mFNDC1 with glucocorticoids would be an interesting and fertile area to explore.

Figure for reviewers removed

Figure for reviewers removed

18. Can the therapeutic effects of mFNDC1 be sustained over the long term, or are there concerns about tolerance or resistance development?

Response: Thanks for your question. We treated *mdx* mice with mFNDC1 for 4 weeks and found that it significantly improves the skeletal muscle function of *mdx* mice (Fig. 9A-J). To further investigate whether the therapeutic effects of mFNDC1 could be sustained over the long term, we treated *mdx* mice with mFNDC1 for 8 weeks and found a reduction in serum CK levels (**Fig R11B**), as well as increases in exercise distance and grip strength (**Fig R11C-D**). H&E staining showed that mFNDC1-treated *mdx* mice had larger myofiber than controls, suggesting that the muscle pathological phenotype of *mdx* mice was improved (**Fig R11E-F**). Furthermore, to assess whether there are concerns about mFNDC1 tolerance or resistance, we compared the muscle pathological phenotype of *mdx* mice treated with mFNDC1 for 4 weeks and 8 weeks. The results showed that mFNDC1 treatment significantly increased the average myofiber size for both 4 and 8 weeks (**Fig R13A-B**). We further examined the downstream signaling of FNDC1 and found a sustained increase in phosphorylated FAK in skeletal muscle of mFNDC1 treated mice for

4 or 8 weeks (**Fig R13C-D**). These results suggest that no obvious tolerance or resistance
developed during the current observation period. We will monitor the mice for a long period
and look forward to sharing new results with you in the future.

Figure for reviewers removed

**19. Given the diverse roles of integrins in various cellular processes, are there any**
**potential off-target effects of mFNDC1 via interactions with other integrins or**
**pathways?**

**Response:** Thanks, this is a very interesting question and warrants further study. As
shown in **Appendix Table S3**, in addition to ITGB1, other receptors were also identified to
potentially interact with FNDC1 such as ITGBA11. Previous studies have shown that

FNDC1 binds to VEGFR2 and enhances the proliferation and metastasis of gastric cancer
 cells (Lu Y, et al, *iScience*. 2023). This suggests that FNDC1 plays different roles through
 different pathways. Therefore, it is possible that FNDC1 may interact with other receptors
 to produce off-target effects. In addition, as a secreted protein, FNDC1 can reach the
 whole body through circulation. FNDC1 may play regulatory roles in other tissues or
 organs by acting in different receptors.

**Appendix Table S3. Mass spectrometry results**

Accession	Coverage [%]	# Peptides	# PSMs	# Unique Peptides	Sequest HT	Gene name
Q4ZHG4.4	8	6	6	6	35.25	FNDC1
P09055.1	6	4	4	4	24.36	ITGB1
P09813.2	3	3	3	2	14.36	APOA1
Q920R6.1	7	3	3	2	10.35	ATP6V0C
P01027.3	5	3	3	1	10.6	C3
P23927.2	5	4	3	1	8.32	CRYAB
Q9D952.3	5	3	3	2	5.36	EVPL
E9PV24.1	4	2	2	1	3.23	FGA
Q8K0E8.1	4	3	3	1	4.36	FGB
Q8VCM7.1	3	2	1	1	5.32	FGG
P68871.2	2	2	1	1	3.26	HBB
Q61646.1	2	1	1	1	2.31	HP
P01868.1	4	2	1	1	2.35	IGHG1
P01837.2	2	1	1	1	2.36	IGKC
P63328.1	3	1	1	1	4.32	IVL
Q60590.1	4	1	1	1	6.32	LGALS7B
Q8K4L4.3	3	2	1	1	3.25	ORM1
Q9R269.1	8	2	1	1	5.36	POF1B
P26595.1	3	1	1	1	2.83	AAT
P70124.1	2	1	1	1	3.68	SERPINB5
Q92111.1	3	1	1	1	15.32	TRF
Q9ESL8.2	2	1	1	1	14.3	FGF16
Q66K08.1	5	2	1	1	10.5	CILP1
Q8BV57.1	6	3	1	1	8.36	SSC5D
P37889.2	4	2	1	1	7.25	FBLN2
Q61554.2	6	3	1	1	3.25	FBLN1
Q9QZJ6.1	2	1	1	1	3.66	MFAP5
O08999.2	4	2	1	1	1.3	LTBP2
Q9QZZ6.1	2	1	1	1	5.36	DPT
P61622.1	3	2	1	1	8.691	ITGBA11
Q3TYX2.1	2	1	1	1	7.25	LRRN4CL
Q9R045.2	2	1	1	1	4.23	ANGPTL2
Q6PE55.1	2	1	1	1	2.01	PDGFRL
Q9ET66.2	2	1	1	1	1.25	PI16
Q99MQ4.1	2	1	1	1	1.58	ASPN
Q8CD91.1	2	1	1	1	2.98	SMOC2

**20. Can the findings regarding FNDC1's role in promoting myogenesis be**

**extrapolated to other muscle-related conditions beyond acute and chronic muscle**
**diseases, such as age-related muscle loss or sarcopenia?**

**Response:** Thanks for your question. To investigate whether the role of FNDC1 in
promoting myogenesis can be extrapolated to other muscle-related conditions, we first
examined serum levels of FNDC1 in old mice (18-month-old) and found that levels were
significantly decreased compared to young mice (2-month-old) (**Fig R14A**). We then
treated old mice with recombinant mFNDC1 protein (2.5mg/kg body weight) by
intraperitoneal injection every two days for 4 weeks and found increased grip strength and
improved inverted grid hanging time (**Fig R14B-C**). Similarly, running distance was
increased in mFNDC1-treated old mice (**Fig R14D**). Furthermore, mFNDC1 treatment
increased muscle weight of tibialis anterior (TA), gastrocnemius (GAS), and quadriceps
(QUA) in old mice (**Fig R14E-H**). In summary, these results suggest that mFNDC1
improves age-related muscle loss and function performance. Systematic studies of the role
of FNDC1 in age-related muscle loss merits further investigation.

Figure for reviewers removed

**21. Has the safety profile of mFNDC1 been assessed, particularly in terms of**
**potential immunogenicity, toxicity, or adverse effects on other tissues or organs?**

**Response:** Thanks. To assess the safety profile of mFNDC1, we determined indices for
spleen, renal, and liver, as well as serum inflammatory factors in C57BL/6J mice treated
with mFNDC1 for four weeks. mFNDC1-treated mice and their controls showed: ① similar
spleen weight/body weight ratio (**Fig R15A**) and blood lymphocyte count (**Table R1**); ②
comparable levels of renal function indicators including blood creatinine and blood urea
nitrogen (**Table R1**); ③ indistinguishable levels of liver function indicators alanine
transaminase (ALT), aspartate transaminase (AST), and alkaline phosphatase (ALP)
(Table R1); ④ similar levels of serum inflammatory factors including TNF- α , IL-1 β and IL-6
(**Fig R15B-D**). In *mdx* mice, four weeks treatment with mFNDC1 significantly reduced
serum levels of inflammatory factors TNF- α , IL-1 β , and IL-6, providing evidence for the
anti-inflammatory effect of FNDC1 (**Fig 8E-G**). In conclusion, these results suggest that
mFNDC1 does not induce obvious immunogenicity, toxicity and adverse effects on other
tissues or organs in mice.

Figure for reviewers removed

Table for reviewers removed

Figure 8. FNDC1 treatment ameliorates systemic inflammation in *mdx* mice. (E-G) Serum concentrations of TNF α , IL1 β , and IL6 in WT and *mdx* mice treated with mFNDC1 or control for 4 weeks (n = 6). Serum was taken four weeks after mFNDC1 treatment from the indicated mouse models. One-way ANOVA (E, F, G) was performed to compare all listed conditions unless otherwise noted, and data are represented as mean \pm SEM. * $p < 0.05$, ** $p < 0.01$, and *** $p < 0.001$.

Dear Prof. WU,

Congratulations on a great revision! Overall, the referees have been positive and in support of moving forward with your article. However, there remain several editorial items that need to be addressed in another revised version. When you submit your revised version, please add these to your final point-by-point response:

1. Please ensure that all fields are completed in the author checklist.
2. Please upload figures 1-9 as individual, high resolution figure files.
3. Please add the funding information into the Acknowledgement section.
4. Please rename the conflict of interest statement to "Disclosure and competing interests statement" and move to after the Acknowledgement section.
5. Please correct the reference format to fit our journal style, alphabetical order, 10 author names before et al.
6. Supplemental information should be renamed "Appendix" and uploaded in PDF format. The list of abbreviations should be removed and the nomenclature corrected to "Appendix Table S1" etc and "Appendix Figure S1" etc. A table of contents with page numbers should be added to the first page.
7. For the source data, the image files for Fig 3j, and 3l are missing and we were unable to open Fig 9.
8. Please provide a reagent table.
9. We include a synopsis of the paper (see <http://emboj.embopress.org/>). Please provide me with a general summary statement and 3-5 bullet points that capture the key findings of the paper.
10. We also need a summary figure for the synopsis. The size should be 550 wide by 200-440 high (pixels). You can also use something from the figures if that is easier.
11. Please ensure that the tables are called out in sequential order in the main manuscript.
12. Please note that the exact p values are not provided in the legends of figures 1b-c, f, h; 2b-e, h-k; 3b-d, f-g, i, n-o; 4i; 5c-e, g, i; 6e-i; 7c, f; 8b-g, i; 9a-j.
13. Please note that in figures 2b-e, h-k; 3b-d, f-g, i, n-o; 5c-e, g, i; 6e-i; 7c, f; 9a-j; there is a mismatch between the annotated p values in the figure legend and the annotated p values in the figure file that should be corrected.
14. Although 'n' is provided, please describe the nature of entity for 'n' in the legends of figures 1b-c; 2b-e, h-k; 3b, f-g, i, n-o; 4i; 5c-e, g, i; 6g; 7f; 8c, i.

Thank you for the opportunity to consider your work for publication, I look forward to your revision.

Yours sincerely,

Kelly M Anderson, PhD
Editor, The EMBO Journal
k.anderson@embojournal.org

Referee #1:

In this revised version, the authors have addressed most of my comments, and I find the revisions satisfactory. I only have one

final minor comment: there is an error in the title of Figure S4. I suggest changing it to "Fndc1 knockdown interferes with myoblast differentiation."

Referee #2:

The authors have successfully addressed all of my comments, and the revised article is now satisfactory. I recommend the publication of this article in EMBO J

Referee #3:

The authors have addressed every comment, and have actually done a large amount of additional work. The concerns I had have all been addressed, and I have no further comments.

Responses to the comments for the manuscript “The novel myokine FNDC1 promotes muscle regeneration and alleviates dystrophic muscle” (ID: EMBOJ-2024-117378R1).

Response to Reviewer #1

In this revised version, the authors have addressed most of my comments, and I find the revisions satisfactory. I only have one final minor comment: there is an error in the title of Figure S4. I suggest changing it to "Fndc1 knockdown interferes with myoblast differentiation."

Response: Thank you for your valuable feedback and for acknowledging our revisions. We have made the suggested correction to the title of Figure S4, which now reads "Fndc1 knockdown interferes with myoblast differentiation." We appreciate your attention to detail, and we believe this adjustment enhances the clarity of our manuscript.

Response to Reviewer #2

The authors have successfully addressed all of my comments, and the revised article is now satisfactory. I recommend the publication of this article in EMBO J.

Response: We greatly appreciate your recommendation for publication in EMBO J and are grateful for your support throughout the review process.

Response to Reviewer #3

The authors have addressed every comment, and have actually done a large amount of additional work. The concerns I had have all been addressed, and I have no further comments.

Response: Thank you for your positive feedback and for recognizing our efforts to address your comments.

Dear Dr. Wu,

Thank you for addressing the final editorial points. I sincerely apologise for the delay in communicating the decision due to the high number of submissions we receive at the moment. I am now pleased to inform you that your manuscript has been accepted for publication.

Before we forward your manuscript to our publishers, I would like to propose some edits in the manuscript title, abstract and synopsis (please see below and the attached file). I have also written a short blurb that will accompany the title of your manuscript in our online system. Please let me know if any corrections or adjustments are needed.

Title:

FNDC1 is a myokine that promotes muscle regeneration and alleviates muscular dystrophy

Blurb:

The fibronectin domain-containing protein FNDC1 promotes myogenesis via direct binding to integrin $\alpha 5\beta 1$.

Synopsis

Muscle injury induces expression of myokines that promote muscle regeneration. This study identifies the fibronectin domain-containing protein FNDC1 as a myokine that promotes myogenesis and muscle regeneration with potential therapeutic use for treating muscle injuries and related disorders.

- FNDC1 is a previously uncharacterized myokine that promotes myogenesis.
- FNDC1 enhances myoblast differentiation and facilitates myofiber formation.
- Recombinant FNDC1 promotes muscle regeneration and ameliorates muscular dystrophy.
- FNDC1 interacts with integrin $\alpha 5\beta 1$ to activate the FAK/PI3K/AKT/mTOR signaling pathway for muscle growth and repair.

If you have any questions, please do not hesitate to contact the Editorial Office. Thank you for this contribution to The EMBO Journal and congratulations on a nice study!

With best wishes,

Ieva
